# Efficient Diffusion Models under Nonconvex Equality and Inequality Constraints via Landing

**Kijung Jeon** [1]  **Michael Muehlebach** [2]  **Molei Tao** [1]

## Abstract

Generative modeling within constrained sets is essential for scientific and engineering applications involving physical, geometric, or safety requirements (e.g., molecular generation, robotics). We present a unified framework for constrained diffusion models on generic nonconvex feasible sets $\Sigma$ that simultaneously enforces equality and inequality constraints throughout the diffusion process. Our framework incorporates both overdamped and underdamped dynamics for forward and backward sampling. A key algorithmic innovation is a computationally efficient landing mechanism that replaces costly and often ill-defined projections onto $\Sigma$, ensuring feasibility without iterative Newton solves or projection failures. By leveraging underdamped dynamics, we accelerate mixing toward the prior distribution, effectively alleviating the high simulation costs typically associated with constrained diffusion. Empirically, this approach reduces function evaluations and memory usage during both training and inference while preserving sample quality. On benchmarks featuring equality and mixed constraints, our method achieves comparable sample quality to state-of-the-art baselines while significantly reducing computational cost, providing a practical and scalable solution for diffusion on nonconvex feasible sets.

## 1. Introduction

Generative modeling is a fundamental machine learning task. In recent years, denoising diffusion models (Ho et al., 2020; Song et al., 2021) have become the state-of-the-art in image, audio, and video generation, matching and surpassing earlier approaches such as GAN (Goodfellow et al., 2014; Dhariwal & Nichol, 2021). Key advantages include ease of training, high-fidelity sampling, and their flexibility in conditional generation.

The majority of these impressive results have been achieved for data in an unconstrained space, particularly $\mathbb{R}^d$. While this suffices for digital content creation, there are many emerging applications in science and engineering, where the generation of high-fidelity samples that adhere to constraints is crucial. Examples include molecular generation (Jing et al., 2022; Watson et al., 2023; Wu et al., 2024), where atoms must satisfy distance or chirality constraints and respect physical laws, robotics (Chi et al., 2024; Römer et al., 2025; Ma et al., 2025), where trajectories are subject to dynamics, actuation limits, safety margins, and collision-avoidance, and shape optimization for engineering design (Wagenaar et al., 2024; Kyaw et al., 2025; Regenwetter et al., 2022), where specifications, symmetries, and manufacturing impose constraints.

In these settings, a valid sample needs to remain inside a non-trivial feasible set $\Sigma \subset \mathbb{R}^d$, as constraint violations would render the generation physically meaningless or unsafe. Unfortunately, enforcing constraints in diffusion models is challenging: (i) Projection-based methods demand repeated Newton-iterations, which scales poorly with dimension and may fail when $\Sigma$ is nonconvex and projections are undefined (Christopher et al., 2024); (ii) Reparametrization encodes the constraints implicitly, which requires domain knowledge, may alter the conditioning of the score matching, and the fidelity of the sampling. Reparametrizations might be hard to find in applications with complex mixed equality-inequality constraints (Lou & Ermon, 2023); (iii) Penalty and barrier methods (Nocedal & Wright, 2006; Fishman et al., 2023a) incorporate constraints through additional terms in the objective function, which may introduce bias (barrier functions), lead to constraint violations (penalty), and result in additional hyperparameters that are difficult to tune.

Consequently, constrained diffusion is computationally demanding as both training and inference rely on the simulation of diffusion processes. For example, this poses a

---

[1]Georgia Institute of Technology [2]Max Planck Institute for Int. Sys. Correspondence to: Molei Tao <mtao@gatech.edu>.

*Proceedings of the 43rd International Conference on Machine Learning*, Seoul, South Korea. PMLR 306, 2026. Copyright 2026 by the author(s).

All code is available at: github.com/KraitGit/efficient-constrained-diffusion.

significant bottleneck for applications including robotics - where trajectories have to be generated in real time on resource-constrained hardware.

This article addresses the need for computationally-efficient constrained diffusion. Our contributions are threefold:

- **(Landing-based diffusion process)** We extend the recently developed constrained optimization technique of landing (Muehlebach & Jordan, 2022; Ablin & Peyré, 2022; Schechtman et al., 2023; Muehlebach & Jordan, 2026) to generative modeling, enabling inexpensive first-order updates that steer the diffusion process toward $\Sigma$ without requiring per-step exact projections or Newton iterations. The continuous-time dynamics preserve feasibility when initialized on the manifold and exponentially decay constraint violations; the discretized sampler uses this self-correction to avoid repeated Newton projections, with only an optional terminal projection.

- **(Unified framework for constraints)** Our method works for general nonconvex feasible sets as we develop an SDE-based framework that unifies both equality and inequality constraints.

- **(Fast constrained diffusion via underdamped dynamics)** The framework encompasses constrained versions of both underdamped and overdamped Langevin dynamics. The fast mixing of the underdamped version is leveraged to substantially reduce the length of forward trajectory $N$ to reach the prior distribution, thereby cutting the dominant sampling (up to $47\times$) and training cost (up to $5\times$) in constrained diffusion models.

## 2. Related Works

Recent works have extended score-based diffusion models from Euclidean spaces to non-Euclidean domains. Our setup considers a domain specified by equality and inequality constraints, and closely related is a rich collection of successful generative models for manifold data (Mathieu & Nickel, 2020; Rozen et al., 2021; De Bortoli et al., 2022; Huang et al., 2022; Chen & Lipman, 2024; Zhu et al., 2025). However, while many manifolds considered (such as $S^d$ and $SO(n)$) can also be specified using equality constraints, the geometry of the constrained set can easily become too complicated to handle when there are a large number of constraints, or when the constraints introduce manifolds with boundaries or even lower dimensional structures.

Approaches directly targeting constrained generation also exist, particularly when data lie in a bounded subset $\Sigma \subset \mathbb{R}^d$. Several strategies have been explored. For example, classical reflected Brownian motion (Williams, 1987; Pilipenko, 2014) was recently leveraged to create constrained diffusion models (Fishman et al., 2023a; Lou & Ermon, 2023; Fishman et al., 2023b), and the recent development of mirror

Langevin dynamics and algorithms (Zhang et al., 2020; Li et al., 2022) was also employed for constrained generation (Liu et al., 2023). However, the former approach is difficult to be made simulation-free and/or tractable in the conditional score, which is essential for efficiency, while the latter only works for convex constraints. Even more recently, Riemannian Denoising Diffusion Probabilistic Models (RD-DPM) (Liu et al., 2026) extends score-based models to general manifolds via per-step Newton's projections, ensuring feasibility but incurring sizable computational cost and occasional projection failures on nonconvex sets. In parallel, Riemannian Flow Matching (RFM) (Chen & Lipman, 2024) learns manifold flows without projections for simple manifolds, yet typically needs a long integration horizon or still projections on nontrivial geometries.

In the light of these advances, we construct an efficient diffusion process that remains on feasible sets described by equality and inequality constraints. The key is to incorporate landing, a technique developed in constrained optimization (Muehlebach & Jordan, 2022; Ablin & Peyré, 2022; Schechtman et al., 2023; Muehlebach & Jordan, 2026), which handles non-convex constraints and guarantees feasibility while avoiding expensive projections, retractions, or evaluations of the exponential map.

## 3. Preliminaries & Notations

**Constrained set and geometry.** We implement the diffusion model on a constrained set

$$\Sigma := \left\{ x \in \mathbb{R}^d \mid h(x) = 0, g(x) \leq 0 \right\}$$

defined by smooth equality constraints $h : \mathbb{R}^d \to \mathbb{R}^m$ and inequality constraints $g : \mathbb{R}^d \to \mathbb{R}^l$. For theoretical analysis, we assume that $\Sigma$ is a stratified manifold and constraints $h, g$ satisfy the relaxed Constant Rank Constraint Qualification (rCRCQ) (Minchenko & Stakhovski, 2011) on a neighborhood of $\Sigma$.

Specifically, for each $x \in \mathbb{R}^d$ with the index set of active inequalities $I_x := \{j \in [l] \mid g_j(x) \geq 0\}$, we say that rCRCQ holds if, for every index set $J \subset I_x$, the set $\{\nabla h_i(y)\}_{i=1}^m \cup \{\nabla g_j(y)\}_{j \in J}$ has the same rank for any $y$ in the neighborhood of $x$ (see Remark 3 for further discussion on rCRCQ).

We note that the rCRCQ implies that the stacked Jacobian $\nabla J(x) \in \mathbb{R}^{(m+|I_x|)\times d}$ (with $J(x) := [h(x), g_{I_x}(x) + \epsilon]^T$) has a constant rank in the neighborhood of $x$, where the boundary repulsion rate $\epsilon > 0$ is a hyperparameter to be introduced later. Due to this result, the tangent space of $\Sigma$ is characterized by the kernel of the Jacobian, given by

$$T_x\Sigma := \left\{ p \in \mathbb{R}^d \mid \nabla J(x)p = 0 \right\}.$$

Accordingly, the orthogonal projector $\Pi(x) := I -$

$\nabla J(x)^T G(x)^\dagger \nabla J(x)$ onto $T_x\Sigma$ is well-defined, where $G(x)^\dagger$ is the Moore-Penrose pseudo-inverse of $G(x) := \nabla J(x)\nabla J(x)^T$.

For the reference measure on $\Sigma$, we use the induced surface (Hausdorff) measure on $\Sigma$, denoted as $d\sigma_\Sigma$.

In the underdamped setting, the natural phase space is the cotangent bundle given by

$$T^*\Sigma := \left\{ (x,p) \in \mathbb{R}^d \times \mathbb{R}^d \mid x \in \Sigma, \nabla J(x)p = 0 \right\}$$

In this manifold, the natural reference measure is the Liouville measure $d\sigma_{T*\Sigma}(x,p) = d\sigma_\Sigma(x) \otimes dp(x)$ where $dp(x)$ is Lebesgue measure on $T_x^*\Sigma := \left\{ p \in \mathbb{R}^d \mid \nabla J(x)p = 0 \right\}$. For the detailed background and notations, see subsection B.1 for the overdamped, subsection B.2 for the underdamped, and the table of key notations (Table 8).

**Time grid and schedule.** In our paper, we use continuous time $t \in [0, T]$ and a uniform grid $t_k := k\Delta t$ for $k \in \{0, ..., N\}$ with $T = N\Delta t$ for the implementation of the diffusion model with step size $\Delta t > 0$.

Also, the noise magnitudes used in implemented diffusion models are specified by a scheduler $\sigma : [0, T] \to \mathbb{R}_+$ and, in our case, we use the linear scheduler given by $\sigma(t) := \sigma_{\min} + \frac{t}{T}(\sigma_{\max} - \sigma_{\min})$ with $\sigma_k := \sigma(t_k)$.

## 4. Main Results

### 4.1. Constrained Langevin Dynamics via Landing

Classical constrained samplers take an unconstrained (in $\mathbb{R}^d$) or tangential step (in $T_x\Sigma$) and then project back to $\Sigma$. This can be problematic for several reasons: (i) on nonconvex manifolds, nearest-point projection can be multi-valued or not globally defined; (ii) per-step projection solves are costly and may fail (e.g. Newton's method failure); and (iii) behavior near $\partial\Sigma$ is delicate since the active set $I_x$ changes frequently.

**From projections to landing mechanisms.** We therefore seek a projection-free scheme that remains well-posed even when local projections are unreliable. Our approach builds robustness directly into the SDE via a landing term that enforces exponential decay of constraint violation $J$:

$$dJ(X_t) = -\alpha\sigma(t)^2 J(X_t)dt \quad \text{(Target landing property)}$$

so discretization-induced infeasibility self-corrects without explicit projection. The landing modifies only the normal component of the drift, while leaving the tangential drift and diffusion unchanged, so trajectories evolve intrinsically on $\Sigma$ (or $T^*\Sigma$ in the underdamped case). Formally, any diffusion process $X_t$ with this property enjoys the guarantees stated in Lemma 1. For detailed proofs of the mathematical claims below, see Appendix B.

**Lemma 1** (Exponential decay of constraint functions). *Under the target property $dJ(X_t) = -\alpha\sigma(t)^2 J(X_t)dt$, the diffusion process $X_t$ satisfies the following constraint satisfaction property almost surely:*

$$h_i(X_t) = h_i(X_0)e^{-\alpha S(t)}, \quad t \geq 0$$

*and*

$$\begin{cases} g_j(X_t) = -\epsilon + (g_j(X_0) + \epsilon)e^{-\alpha S(t)}, & t \leq \tau_{j,\epsilon} \\ g_j(X_t) \leq 0, & t \geq \tau_{j,\epsilon}, \end{cases}$$

*where $S(t) := \int_0^t \sigma(s)^2 ds$ and $\tau_{j,\epsilon}$ are defined to be*

$$\tau_{j,\epsilon} := \inf \left\{ t \geq 0 \mid S(t) \geq \frac{1}{\alpha} \ln\left(\frac{g_j(X_0) + \epsilon}{\epsilon}\right) \right\}$$

*for all $j \in I_{X_0}$.*

**Constrained Overdamped Langevin dynamics via Landing (OLLA).** Following the framework proposed in Jeon et al. (2025), we first derive such landing-based constrained Langevin dynamics for the overdamped case. By viewing constrained Langevin dynamics in Lagrangian form (Lelièvre et al., 2010), we pick a Lagrangian process $d\lambda_t$ so that it can impose the target property $dJ(X_t) = -\alpha\sigma(t)^2 J(X_t)dt$ and have a closed-form SDE as follows:

**Proposition 1** (Construction, stationarity and backward process of OLLA). *Consider the following Lagrangian form constrained overdamped Langevin dynamics of $X_t \sim q_t$:*

$$dX_t = -\frac{1}{2}\sigma(t)^2 \nabla f(X_t)dt + \sigma(t)\circ dW_t + \nabla J(X_t)^T d\lambda_t \tag{1}$$

*where $d\lambda_t$ is the adapted process such that $dJ(X_t) = -\alpha\sigma(t)^2 J(X_t)$. The explicit solution of $d\lambda_t$ provides the closed form SDE of (1) as follows:*

$$dX_t = -\frac{\sigma(t)^2}{2}\Pi(X_t)\nabla f(X_t)\,dt + \sigma(t)\Pi(X_t)\,dW_t$$
$$+ \left[ \underbrace{-\alpha\sigma(t)^2\nabla J(X_t)^T G^\dagger(X_t)J(X_t)}_{\text{Landing term}} + \frac{\sigma(t)^2}{2}\mathcal{H}(X_t) \right]dt.$$

*Furthermore, the backward process $\overleftarrow{X}_t$ of OLLA is:*

$$d\overleftarrow{X}_t = \frac{1}{2}\sigma(T-t)^2\Pi(\overleftarrow{X}_t)\left[\nabla f(\overleftarrow{X}_t) + 2\nabla\ln q_{T-t}(\overleftarrow{X}_t)\right]dt$$
$$+ \frac{1}{2}\sigma(T-t)^2\mathcal{H}(\overleftarrow{X}_t)dt + \sigma(T-t)\Pi(\overleftarrow{X}_t)\circ d\bar{W}_t$$
$$- \underbrace{\alpha\sigma(T-t)^2\nabla J(\overleftarrow{X}_t)^T G^\dagger(\overleftarrow{X}_t)J(\overleftarrow{X}_t)}_{\text{Landing term}}dt.$$

*where $\mathcal{H}$ is the mean curvature correction term defined as*

$$\mathcal{H}(x) := -\nabla J(x)^T G^\dagger(x) \begin{bmatrix} \text{Tr}\left(\nabla^2 J_1(x)\Pi(x)\right) \\ \vdots \\ \text{Tr}\left(\nabla^2 J_{m+|I_x|}(x)\Pi(x)\right) \end{bmatrix}.$$

*By assuming $\sigma(t)$ constant and $X_0 \in \Sigma$, OLLA has the stationary distribution $q_\Sigma \propto \exp(-f(x))$ with respect to $d\sigma_\Sigma$.*

**Constrained Underdamped Langevin Dynamics via Landing (ULLA).** In the underdamped case, we are required to satisfy both the target property $dJ(X_t) = -\alpha\sigma(t)^2 J(X_t)dt$ and also the momentum tangency constraint $\nabla J(X_t)P_t = 0$ so that $(X_t, P_t) \in T^*\Sigma$ for $t \geq 0$. Therefore, we control the two Lagrangian processes $d\lambda_t, d\mu_t$ to impose such constraints, and the resulting solution of Lagrangian processes produces the following closed SDE for ULLA:

**Proposition 2** (Construction, stationarity, and backward process of ULLA). *Consider the following Lagrangian form constrained underdamped Langevin of $(X_t, P_t) \sim q_t$:*

$$\begin{cases} dX_t = \sigma(t)^2 P_t dt + \nabla J(X_t)^T d\lambda_t, \\ dP_t = -\sigma(t)^2 \nabla f(X_t)dt - \sigma(t)^2 \gamma P_t dt + \nabla J(X_t)^T d\mu_t \\ \qquad + \sigma(t)\sqrt{2\gamma} \circ dW_t, \end{cases} \tag{2}$$

*where $d\lambda_t, d\mu_t$ are the adapted processes such that $dJ(X_t) = -\alpha\sigma(t)^2 J(X_t)$ (position constraint) and $\nabla J(X_t)P_t = 0$ (momentum tangency constraint), respectively.*

*Assuming $\nabla J(X_0)P_0 = 0$, the explicit solution of $d\lambda_t, d\mu_t$ provides the closed form SDE of (2) as follows:*

$$\begin{cases} dX_t = \sigma(t)^2 P_t dt - \alpha\sigma(t)^2 \nabla J(X_t)^T G^\dagger(X_t) J(X_t)dt, \\ dP_t = \Pi(X_t)\left[-\sigma(t)^2 \nabla f(X_t) - \sigma(t)^2 \gamma P_t\right]dt \\ \qquad - \sigma(t)^2 \nabla J(X_t)^T G^\dagger(X_t)\mathcal{H}_1(X_t, P_t)dt \\ \qquad + \alpha\sigma(t)^2 \nabla J(X_t)^T G^\dagger(X_t)\mathcal{H}_2(X_t, P_t)dt \\ \qquad + \sigma(t)\sqrt{2\gamma}\Pi(X_t)dW_t, \end{cases}$$

*where $\mathcal{H}_1, \mathcal{H}_2 \in \mathbb{R}^{m+|I_x|}$ are the curvature correction terms defined as*

$$[\mathcal{H}_1(x,p)]_i := p^T \nabla^2 J_i(x)p$$
$$[\mathcal{H}_2(x,p)]_i := p^T \nabla^2 J_i(x)(\nabla J(x)^T G^\dagger(x)J(x))$$

*with $[\mathcal{H}_1(x,p)]_i$ and $[\mathcal{H}_2(x,p)]_i$ denoting the $i$-th entries of $\mathcal{H}_1(x,p)$ and $\mathcal{H}_2(x,p)$, respectively. Furthermore, the*

*backward process $\overleftarrow{X}_t$ of ULLA is given as:*

$$\begin{cases} d\overleftarrow{X}_t = -\sigma(T-t)^2 \overleftarrow{P}_t dt \\ \qquad - \alpha\sigma(T-t)^2 \nabla J(\overleftarrow{X}_t)^T G^\dagger(\overleftarrow{X}_t)J(\overleftarrow{X}_t)dt, \\ d\overleftarrow{P}_t = \sigma(T-t)^2 \Pi(\overleftarrow{X}_t)\left[\nabla f(\overleftarrow{X}_t) + \gamma\overleftarrow{P}_t\right]dt \\ \qquad + 2\gamma\sigma(T-t)^2 \Pi(\overleftarrow{X}_t)\nabla_p \ln q_{T-t}(\overleftarrow{X}_t, \overleftarrow{P}_t)dt \\ \qquad + \sigma(T-t)^2 \nabla J(\overleftarrow{X}_t)^T G^\dagger(\overleftarrow{X}_t)\mathcal{H}_1(\overleftarrow{X}_t, \overleftarrow{P}_t)dt \\ \qquad + \alpha\sigma(T-t)^2 \nabla J(\overleftarrow{X}_t)^T G^\dagger(\overleftarrow{X}_t)\mathcal{H}_2(\overleftarrow{X}_t, \overleftarrow{P}_t)dt \\ \qquad + \sigma(T-t)\sqrt{2\gamma}\Pi(\overleftarrow{X}_t)d\bar{W}_t. \end{cases}$$

*By assuming $\sigma(t)$ constant and $X_0 \in \Sigma, \nabla J(X_0)P_0 = 0$, ULLA has the stationary distribution $q_{T^*\Sigma} \propto \exp(-f(x) - \frac{1}{2}\|p\|^2)$ with respect to $d\sigma_{T^*\Sigma}$.*

### 4.2. Transition Kernels for Forward and Backward Processes

In this section, we outline the discretization of the OLLA and ULLA processes. For the notations, we let $x_k$ for $k \in \{1, ..., N\}$ to be the position vector at $k$-th discrete step of the diffusion process, and set $q_k, p_k, p_k^\theta$ to be the marginal probability densities for forward and backward processes, and the parametrized backward process of $x_k$. Also, we set $p_N, \rho(\cdot|x)$ to be the prior of position and momentum where the momentum prior is given by $\Pi(x)\zeta \sim \rho(\cdot|x)$ with $\zeta \sim \mathcal{N}(0, I_d)$. For detailed derivations of the discretization schemes below, we refer to subsection B.4.

**Discretization of OLLA.** For OLLA, the discretization is straightforward. We employ a standard Euler-Maruyama scheme to integrate the corresponding SDE as follows:

$$\begin{cases} x_{k+1} = x_k - \frac{\sigma_k^2 \Delta t}{2}\Pi(x_k)\nabla f(x_k) + \sigma_k\sqrt{\Delta t}\Pi(x_k)\zeta_k \\ \qquad - \alpha\sigma_k^2 \Delta t(\nabla J^T G^\dagger J)(x_k) + \kappa_k^O \\ x_k = x_{k+1} + \frac{\sigma_{k+1}^2 \Delta t}{2}\Pi(x_{k+1})\left[\nabla f + s_{k+1}\right](x_{k+1}) \\ \qquad + \sigma_{k+1}\sqrt{\Delta t}\Pi(x_{k+1})\zeta_{k+1} \\ \qquad - \alpha\sigma_{k+1}^2 \Delta t(\nabla J^T G^\dagger J)(x_{k+1}) + \kappa_{k+1}^O \end{cases}$$

where $\kappa_k^O := \frac{\sigma_k^2}{2}\mathcal{H}(x_k)\Delta t$ is the mean curvature term and $s_k(x_k) := 2\nabla \ln q_k(x_k)$, which can be learned by a neural network $s_{k+1}^\theta$.

**Discretization of ULLA.** For ULLA, we adopt a specialized scheme to achieve **2× memory efficiency**, which becomes critical for storing long forward trajectories during training.

The method is based on the 1st order non-symmetric OBA

splitting integrator. To eliminate the need to explicitly store the momentum trajectory, we first use an approximated $\tilde{B}$ step, which relies on the before-O step momentum on correction terms $\mathcal{H}_1, \mathcal{H}_2$, rather than after, and secondly, we leverage the recursive nature of the update rule to express the momentum at step $k$ as a function of positions at previous steps, using an approximated momentum vector $\tilde{p}_k$.

This *collapses* the dynamics into the 2nd order Markov chain solely depending on position variables $x_k$ as follows:

$$\begin{cases} x_{k+1} = x_k + \sigma_k^2 \Delta t \Pi(x_k) \left[ a_k \tilde{p}_k^{\text{fwd}} - \sigma_k^2 \Delta t \nabla f(x_k) \right] \\ \qquad + \sigma_k^2 \Delta t \sqrt{1 - a_k^2} \Pi(x_k) \zeta_k \\ \qquad - \alpha \sigma_k^2 \Delta t (\nabla J^T G^\dagger J)(x_k) + \kappa_{k,\text{fwd}}^U \\ x_k = x_{k+1} - \sigma_{k+1}^2 \Delta t \Pi(x_{k+1}) a_{k+1} \tilde{p}_{k+1}^{\text{bwd}} \\ \qquad - \sigma_{k+1}^4 \Delta t^2 \Pi(x_{k+1}) \left[ \nabla f + s_{k+1} \right](x_{k+1}, \tilde{p}_{k+1}^{\text{bwd}}) \\ \qquad + \sigma_{k+1}^2 \Delta t \sqrt{1 - a_{k+1}^2} \Pi(x_{k+1}) \zeta_{k+1}' \\ \qquad - \alpha \sigma_{k+1}^2 \Delta t (\nabla J^T G^\dagger J)(x_{k+1}) + \kappa_{k+1,\text{bwd}}^U \end{cases}$$

where $a_k = e^{-\gamma \sigma_k^2 \Delta t} \in [0, 1]$ is a decaying factor induced by friction $\gamma$ and the approximated momentum are defined by $\tilde{p}_k^{\text{fwd}} := \Pi(x_k) \left( (x_k - x_{k-1})/(\sigma_{k-1}^2 \Delta t) \right)$ and $\tilde{p}_{k+1}^{\text{bwd}} := \Pi(x_{k+1}) \left( (x_{k+2} - x_{k+1})/(\sigma_{k+2}^2 \Delta t) \right)$. Also, the curvature correction terms are provided as:

$$\begin{cases} \kappa_{k,\text{fwd}}^U := -\sigma_k^4 \Delta t^2 \nabla J(x_k)^T G^\dagger(x_k) \\ \qquad \cdot \left[ \mathcal{H}_1(x_k, \tilde{p}_k^{\text{fwd}}) - \alpha \mathcal{H}_2(x_k, \tilde{p}_k^{\text{fwd}}) \right] \\ \kappa_{k+1,\text{bwd}}^U := -\sigma_{k+1}^4 \Delta t^2 \nabla J(x_{k+1})^T G^\dagger(x_{k+1}) \\ \qquad \cdot \left[ \mathcal{H}_1(x_{k+1}, \tilde{p}_{k+1}^{\text{bwd}}) + \alpha \mathcal{H}_2(x_{k+1}, \tilde{p}_{k+1}^{\text{bwd}}) \right] \end{cases}$$

Similarly, we define approximated score $s_{k+1}^\theta$ and train a neural network to approximate this:

$$s_{k+1}(x_{k+1}, \tilde{p}_{k+1}^{\text{bwd}}) := 2\gamma \left( \nabla_p \ln q_{k+1}(x_{k+1}, \tilde{p}_{k+1}^{\text{bwd}}) + \tilde{p}_{k+1}^{\text{bwd}} \right)$$

**Remark 1** (Discretization by Newton solver). Projection-based variants of proposed methods, denoted OLLA-P and ULLA-P, can be obtained by dropping all normal landing and correction terms, and instead solving a Lagrangian multiplier system at each step so that $h(x_k) = 0$. For the detailed derivation, we refer to subsection B.4.

**Remark 2** (Error decomposition and benefits of ULLA). Informally, sample generation error via backward process decomposes into mixing $\mathcal{E}_{\text{mix}}$, discretization $\mathcal{E}_{\text{disc}}$, and score estimation $\mathcal{E}_{\text{score}}$ terms:

$$W_2(q_0, p_0^\theta) \leq \mathcal{E}_{\text{mix}} + \mathcal{E}_{\text{disc}} + \mathcal{E}_{\text{score}}.$$

In this point of view, ULLA significantly reduces $\mathcal{E}_{\text{mix}}$ via ballistic dynamics, accelerating convergence and enabling

smaller trajectory lengths $N$ compared to OLLA. Additionally, the momentum variable mitigates score singularities near $t = 0$, yielding a potentially smoother training objective for $\mathcal{E}_{\text{score}}$. We refer to Remark 4 for a detailed discussion.

**4.3. Conditional Wasserstein Path Matching (CWPM)**

Since the proposed OLLA and ULLA do not use per-step projections, intermediate samples $x_k$ may exhibit minor constraint violations and lie off $\Sigma$. This renders previously proposed training loss, such as DT-ELBO (Liu et al., 2026) or score matching (De Bortoli et al., 2022; Huang et al., 2022), theoretically unstable, as they rely on the assumption that $x_k \in \Sigma$, which can lead to singularity issues.

This problem is particularly acute when measuring the NLL loss, where small violations can introduce substantial bias and undermine its reliability as a sample quality metric. To resolve these theoretical issues, we propose the CWPM framework as below, which is based on the Wasserstein distance rather than KL-divergence, eliminating such theoretical singularities. The derivation involves the relationship between Gelbrich distance and 2-Wasserstein distance (Borelle et al., 2024; Gelbrich, 1990), and refer to Appendix D for detailed proofs and assumptions.

**Theorem 1** (CWPM variational bound – overdamped, informal). *Let $T_{k+1}^\theta = p^\theta(x_k|x_{k+1})$ be the backward transition kernel of the discretized OLLA and define the circuitous density at step $k$ as*

$$\sigma_k := q_k T_k^\theta T_{k-1}^\theta, ..., T_1^\theta, \quad \sigma_0 := q_0.$$

*Assuming existence of $\Lambda_{k+1} > 0$ such that*

$$W_2(\sigma_k, \sigma_{k+1}) \leq \Lambda_{k+1} W_2(q_k, q_{k+1} T_{k+1}^\theta) + \mathcal{O}(\sqrt{\Delta t})$$

*for $k \in \{0, ..., N-1\}$, assuming the sufficient regularity conditions in Lemma D.1, we have $W_2(q_0, p_0^\theta) \lesssim \mathcal{L}^o(\theta) + C^o$, where*

$$\mathcal{L}^o(\theta) := \mathbb{E}\left[ \sum_{k=0}^{N-1} \underbrace{\|\Pi(x_{k+1})(x_k - \mu_{k+1}^o(x_{k+1}))\|^2}_{:= \ell_t^o(\theta)} \right]$$

*and $\mu_{k+1}^o(x_{k+1})$ is the tangential part mean of the parametrized backward process of OLLA defined by*

$$\mu_{k+1}^o(x_{k+1}) := x_{k+1} + \frac{\sigma_{k+1}^2 \Delta t}{2} \Pi(x_{k+1}) \nabla f(x_{k+1})$$

$$+ \frac{\sigma_{k+1}^2 \Delta t}{2} \Pi(x_{k+1}) s_\theta^{k+1}(x_{k+1})$$

*with $C^o$ being a constant independent of $\theta$.*

Similarly, the following results hold for the underdamped:

**Theorem 2** (CWPM variational bound – underdamped, informal). *Let* $y_k = (x_k, x_{k+1}) \in \mathbb{R}^{2d}$ *where* $x_k \sim q_k, x_{k+1} \sim q_{k+1}$. *Define* $\bar{q}_k$ *to be the law of* $y_k$ *and set* $\bar{T}^\theta_{k+1} = p^\theta(y_k|y_{k+1})$ *to be the associated backward transition kernel to* $y_k$. *We set the circuitous density at step* $k$ *as*

$$\bar{\sigma}_k := \bar{q}_k \bar{T}^\theta_k \bar{T}^\theta_{k-1} ... \bar{T}^\theta_1, \quad \bar{\sigma}_0 := \bar{q}_0.$$

*Assuming existence of* $\bar{\Lambda}_{k+1} > 0$ *such that*

$$W_2(\bar{\sigma}_k, \bar{\sigma}_{k+1}) \leq \bar{\Lambda}_{k+1} W_2(\bar{q}_k, \bar{q}_{k+1}\bar{T}^\theta_{k+1}) + \mathcal{O}(\Delta t),$$

*for* $k \in \{0, ..., N-1\}$, *assuming the sufficient regularity conditions in Lemma D.2, we have* $W_2(q_0, p^\theta_0) \lesssim \mathcal{L}^u(\theta) + C^u,$

$$\mathcal{L}^u(\theta) := \mathbb{E}\left[\sum_{k=0}^{N-1} \underbrace{\|\Pi(x_{k+1})(x_k - \mu^u_{k+1}(x_{k+1}, x_{k+2}))\|^2}_{:=\ell^u_t(\theta)}\right]$$

*where* $\mu^u_{k+1}(x_{k+1})$ *is the tangential part mean of the parametrized backward process of ULLA defined by*

$$\mu^u_{k+1} := x_{k+1} - a_{k+1}\sigma^2_{k+1}\Delta t \Pi(x_{k+1})\tilde{p}^{bwd}_{k+1}$$
$$- \sigma^4_{k+1}\Delta t^2 \Pi(x_{k+1})\left[\nabla f(x_{k+1}) + s^{k+1}_\theta(x_{k+1}, \tilde{p}^{bwd}_{k+1})\right]$$

*with* $C^u$ *being a constant independent of* $\theta$.

**Choice of Training Loss.** We remark that other works on diffusion models (Ho et al., 2020; Wang et al., 2024; Karras et al., 2022) demonstrated that choosing the training loss weight $\lambda(k)$ proportional to the inverse of the variance up to a proportionality constant of $1/2$ (in our case, $\lambda(k) = 1/(2\sigma^2_{k+1}\Delta t)$ for overdamped and $\lambda(k) = 1/(2\sigma^4_{k+1}\Delta t^2(1 - a^2_{k+1}))$ for underdamped) is helpful for training the score network. Notably, the resulting training losses

$$\begin{cases} L^{over}_{CWPM}(\theta) = \mathbb{E}_{x_{0:N}}\left[\sum_{k=0}^{N-1} \frac{\ell^o_t(\theta)}{2\sigma^2_{k+1}\Delta t}\right] \\ L^{under}_{CWPM}(\theta) = \mathbb{E}_{x_{0:N}, p_N|x_N}\left[\sum_{k=0}^{N-1} \frac{\ell^u_t(\theta)}{2\sigma^4_{k+1}\Delta t^2(1 - a^2_{k+1})}\right] \end{cases}$$

lead to exactly the same training loss provided in DT-ELBO (Lemma C.1 and Lemma C.2) without the requirement $x_k \in \Sigma$. Summarizing the proposed frameworks, we leave the complete algorithms to Algorithm 1 (OLLA) and Algorithm 2 (ULLA) for detailed implementation.

## 5. Experiments

Our evaluation largely follows the benchmarks of RDDPM (Liu et al., 2026)—Earth/climate datasets, mesh data, the $SO(10)$ manifold, and Alanine dipeptide—and additionally

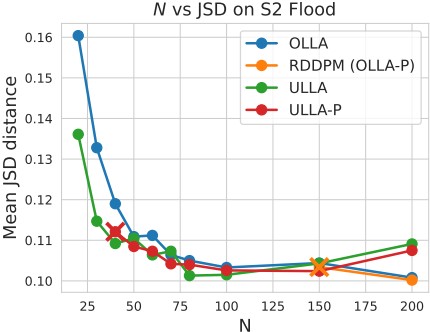

*Figure 1.* Mean JSD on $S^2$ flood versus trajectory length $N$ under the fixed $T$. Cross mark ($\times$) indicates the smallest $N$ values after which projection failures no longer occur during the forward process.

includes a 7-Degree of Freedom (DOF) robot arm trajectory task. The comparison covers state-of-the-art (SOTA) constrained generative model algorithms such as RFM (Chen & Lipman, 2024) and RDDPM, together with Euclidean forward-backward variant baselines that highlight the importance of handling intrinsic geometry and learning the score function on $\Sigma$.

Experimental setup, baseline descriptions, and hyperparameters are deferred to Appendix E. Following the practical landing-based sampling scheme demonstrated in Zhang et al. (2022); Jeon et al. (2025), where landing-based constrained sampling performs robustly even without explicit correction terms, we set the correction terms $\kappa = 0$ to circumvent the high computational cost of Hessian-related calculations.

### 5.1. Equality-only Scenario Tasks

**Earth and climate science datasets.** This benchmark lives on the 2-sphere $S^2$, where nearest-point projection is globally available, so landing-based dynamics are not strictly required. Nevertheless, we use this dataset to (i) assess the intrinsic benefits of underdamped dynamics and (ii) quantify the sampling quality–computational cost trade-off under landing.

From Figure 1, due to faster mixing of the underdamped dynamics, underdamped algorithms (ULLA, ULLA-P) markedly reduce the needed forward length $N$: ULLA-P is stable without projection failure even at $N = 40$, whereas RDDPM (OLLA-P) requires at least $N \approx 150$ to avoid failures. Thus, smaller $N$ yields large training-time savings while preserving comparable sample quality. Although exact projections are available here, Table 1 indicate that ULLA incurs comparable sampling quality under negligible constraint violations; visual comparisons of ULLA (subsection E.3) show similarly generated distribution to projection-based methods, supporting the practical value of landing.

**3D Mesh data on learned manifold.** Unlike the 2-sphere, meshes lie on manifolds where nearest-point projection

*Table 1.* **Generative performance comparison on Earth & Climate, and Mesh datasets.** We report the Jensen-Shannon Distance (JSD) calculated using 2D spherical histograms $(\theta, \phi)$ for Earth & Climate data and face histograms for Mesh data. The average $|h|$ measures the equality constraint violation of generated samples. Results represent the mean $\pm$ standard error over five independent runs. For the trajectory length $N$, $A/B$ denotes the values used for Earth/Climate $(A)$ and Mesh datasets $(B)$, respectively. **Bold** and underline indicate the best and second-best performing methods.

| Method | $N$ | Earth & Climate (JSD) | | | | Mesh Data (JSD) | | | | Avg. $|h|$ |
| | | Volcano | Earthquake | Flood | Fire | Bunny-50 | Bunny-100 | Cow-50 | Cow-100 | (Earth / Mesh) |
|---|---|---|---|---|---|---|---|---|---|---|
| *Riemannian-based* | | | | | | | | | | |
| RFM | 1000 | **0.116**±.002 | **0.089**±.001 | 0.108±.002 | 0.058±.001 | 0.035±.001 | 0.047±.001 | 0.043±.002 | 0.050±.002 | 5.2e-8/1.5e-4 |
| RDDPM | 400 | 0.123±.004 | 0.093±.002 | 0.106±.002 | **0.051**±.001 | 0.032±.001 | 0.034±.000 | 0.046±.001 | **0.034**±.001 | 1.7e-8/1.6e-5 |
| *Euclidean fwd. + bwd. variant* | | | | | | | | | | |
| Euclidean | 50/30 | 0.158±.005 | 0.163±.004 | 0.135±.016 | 0.140±.001 | 0.040±.001 | 0.047±.001 | 0.048±.001 | 0.063±.001 | 4.1e-2/4.2e-2 |
| Projected | 50/30 | 0.156±.005 | 0.152±.003 | 0.133±.012 | 0.133±.001 | 0.049±.000 | 0.051±.001 | 0.057±.001 | 0.068±.001 | 2.1e-8/6.4e-8 |
| Lagrangian | 50/30 | 0.156±.004 | 0.152±.003 | 0.137±.011 | 0.133±.001 | 0.047±.001 | 0.050±.001 | 0.056±.001 | 0.067±.001 | 8.5e-10/1.1e-4 |
| Guided | 50/30 | 0.160±.006 | 0.174±.003 | 0.146±.021 | 0.158±.002 | 0.068±.005 | 0.050±.002 | 0.051±.001 | 0.065±.001 | 2.1e-2/8.2e-1 |
| *Ours* | | | | | | | | | | |
| OLLA | 100 | 0.128±.005 | 0.096±.002 | **0.103**±.002 | 0.060±.001 | 0.030±.000 | **0.032**±.001 | 0.047±.001 | 0.035±.001 | 3.9e-9/3.4e-6 |
| ULLA-P | 100/50 | 0.122±.007 | 0.092±.001 | **0.103**±.002 | 0.053±.001 | 0.040±.001 | 0.038±.001 | **0.040**±.001 | 0.035±.001 | 8.4e-9/4.2e-5 |
| ULLA | 50/30 | 0.125±.005 | 0.099±.002 | 0.110±.001 | 0.069±.002 | **0.029**±.001 | 0.033±.001 | 0.044±.001 | 0.036±.001 | 2.1e-9/1.5e-7 |

*Table 2.* **Comparison of computational efficiency (Wall-clock time).** Total training time and simulation times (Sim.) are reported in seconds, where Sim. denotes the time spent on forward trajectory simulation during training. Our landing-based methods (OLLA, ULLA) exhibit significantly lower simulation cost than Riemannian baselines.

| Method | Earth (s) | Mesh (s) |
| *(Traj. length $N$)* | (Train / Sim.) | (Train / Sim.) |
|---|---|---|
| *Riemannian-based* | | |
| RFM (1000) | 4019 (0) | 145424 (112244) |
| RDDPM (400) | 12388 (3302) | 1916 (126) |
| *Ours* | | |
| OLLA (100) | 1686 (749) | 642 (4.0) |
| ULLA-P (100/50) | 1631 (1154) | 387 (10.5) |
| ULLA (50/30) | **1021** (**530**) | **360** (**2.0**) |

is not globally defined and, in our benchmark, must be approximated by a Newton solver because the constraint $h(x) = 0$ is represented by a learned neural network - making projection-based sampling computationally expensive. In this regime, landing becomes particularly effective: as shown in Table 1, ULLA and ULLA-P show comparable JSD of RDDPM and RFM with far fewer steps $(N = 30, 50)$, yielding $5\times$ faster training and up to $47\times$ faster sampling than RDDPM.

The gains stem from the combinations of the following facts: (i) the underdamped dynamics permits much smaller $N$, and (ii) landing (particularly without curvature corrections) requires only a single constraint-gradient evaluation per step, avoiding iterative projections. These improvements indicate

that, for complex learned manifolds where projection is expensive, ULLA provides a scalable and efficient alternative.

**High-dimensional special orthogonal group: SO(10).** This experiment evaluates scalability on the high-dimensional Lie group $SO(10) \subset \mathbb{R}^{100}$, defined by 55 equality constraints $(X^T X = I)$; $\det(X) = 1$ condition is checked based on rejection. The synthetic distribution is multimodal with $m$ modes, and sampling quality is assessed by power-trace statistics. As shown in Figure 2a and subsection E.3, landing-based methods (ULLA/ULLA-P/OLLA) remain efficient on this complex manifold, producing high-quality samples with a forward trajectory length of $N = 50$, whereas RDDPM requires at least $N \approx 150$ to avoid projection failures.

### 5.2. Mixed Scenario Tasks

**Alanine dipeptide and 7-DOF robot arm.**

We further evaluate our landing algorithms under mixed-constraints setups.

The feasible set $\Sigma$ is defined by complex equality and inequality constraints. In these settings, exact projections are often numerically unstable

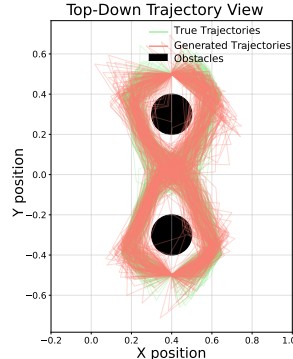

*Figure 3.* Generated Robot arm trajectories (red) by ULLA.

or computationally prohibitive. As summarized in Table 3, standard baselines encounter significant difficulties: the *Pro-*

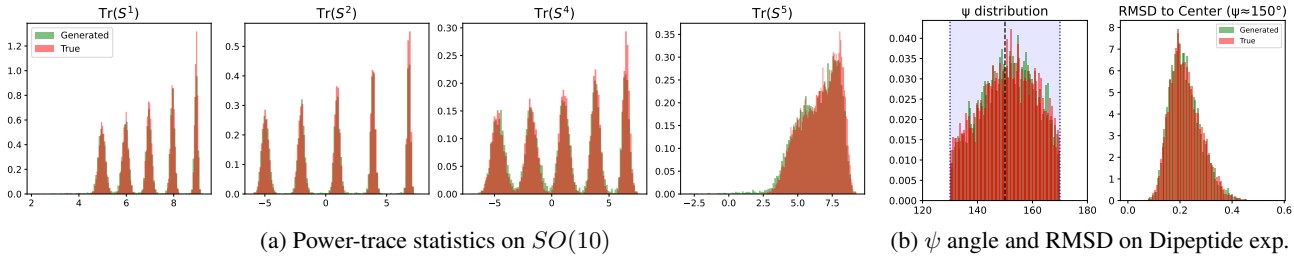

(a) Power-trace statistics on $SO(10)$      (b) $\psi$ angle and RMSD on Dipeptide exp.

*Figure 2.* **Generative performance on complex geometric tasks.** (a) Histograms of the generated power-trace statistics $\mathsf{Tr}\left(S^k\right)$ for $k \in \{1, 2, 4, 5\}$ on $SO(10)$ ($m = 5$), where ULLA (green) accurately recovers the ground-truth (red) distributions. (b) Joint distribution of $\psi$ angle and Root Mean Square Deviation (RMSD) for the Alanine Dipeptide task; the blue shaded area represents the feasible region defined by inequality constraints $\psi \in [130°, 170°]$.

*Table 3.* **Unified generative performance on mixed constraint tasks.** We report JSD (lower is better) with standard errors. Left: Dimension scalability on 7-DOF robot arm ($N = 100$) across dimensions $d$. Middle: Alanine Dipeptide conformation ($N = 100$). Right: Average constraint violations for Robot (Rob) and Alanine (Ala). NaN indicates method failure (divergence or projection failure). **Bold** and underline indicate the best and second-best JSD results.

| Method | 7-DOF Robot Arm (JSD) | | | | Alanine (JSD) | | Violations (Avg.) | | | |
|---|---|---|---|---|---|---|---|---|---|---|
| | $d = 140$ | $d = 280$ | $d = 420$ | $d = 560$ | $\psi$ Angle | RMSD | Rob-$|h|$ | Rob-$|g^+|$ | Ala-$|h|$ | Ala-$|g^+|$ |
| *Euclidean fwd. + bwd. variant* | | | | | | | | | | |
| Euclidean | $\underline{0.498}_{\pm.028}$ | $0.656_{\pm.034}$ | $\underline{0.647}_{\pm.022}$ | $0.750_{\pm.025}$ | $0.150_{\pm.003}$ | $\underline{0.057}_{\pm.001}$ | 8.3e-1 | 5.3e-3 | 5.7e-2 | 2.4e-3 |
| Lagrangian | $0.769_{\pm.043}$ | $0.816_{\pm.005}$ | $0.831_{\pm.001}$ | $0.831_{\pm.002}$ | NaN | | 4.9e-2 | 1.6e-3 | NaN | |
| Projected | NaN | | | | $\underline{0.145}_{\pm.002}$ | $0.073_{\pm.005}$ | NaN | | 5.4e-8 | 3.3e-3 |
| Guided | $0.499_{\pm.028}$ | $\underline{0.655}_{\pm.041}$ | $0.665_{\pm.013}$ | $\underline{0.740}_{\pm.033}$ | $0.152_{\pm.003}$ | $\underline{0.057}_{\pm.002}$ | 8.0e-1 | 4.9e-3 | 5.7e-2 | 2.4e-3 |
| *Ours* | | | | | | | | | | |
| ULLA | $\mathbf{0.275}_{\pm.011}$ | $\mathbf{0.295}_{\pm.006}$ | $\mathbf{0.366}_{\pm.005}$ | $\mathbf{0.391}_{\pm.012}$ | $\mathbf{0.031}_{\pm.002}$ | $\mathbf{0.035}_{\pm.002}$ | 1.8e-5 | 1.5e-9 | 1.4e-7 | 6.0e-11 |

*jected* Euclidean variant failed in the high-dimensional 7-DOF robot arm task due to severe projection failures, while the *Lagrangian* Euclidean variant failed to converge to a high-quality distribution in the Dipeptide task. For similar reasons, OLLA, OLLA-P, and ULLA-P are omitted from the full mixed-task benchmark in Table 3: OLLA suffered from landing instability, while OLLA-P and ULLA-P encountered frequent projection failures, preventing competitive performance.

In contrast, our proposed ULLA method demonstrates superior performance compared to the valid Euclidean forward-backward variants. ULLA not only achieves significantly lower JSDs, consistently outperforming Euclidean baselines even as the dimension scales, but also maintains extremely low constraint violations (e.g., avg. $|h| \approx 10^{-5}$ and $|g^+| \approx 10^{-9}$), effectively respecting the complex geometry without the need for expensive multiple projection steps. In the 7-DOF robot arm experiments, ULLA is run without the optional terminal projection; nevertheless, it still achieves low constraint violation, showing that the landing drift provides effective numerical self-correction.

### 5.3. Component Ablations

We organize the ablations around 5 practical questions: (i) how much landing reduces projection overhead compared

with projection-based algorithms, (ii) why ULLA is preferable to OLLA on complicated mixed-constraint tasks, (iii) whether the correction terms materially affect performance, (iv) what happens when the optional terminal projection is omitted, and (v) how sensitive ULLA is to the landing and repulsion rates.

**Projection overhead.** Projection is the dominant cost in projection-based variants. ULLA avoids repeated per-step projection and uses only an optional terminal cleanup projection. As reported in Table 4, projection cost accounts for more than $90\%$ of inference time in ULLA-P. Since ULLA-P performs projections at intermediate sampling steps, it can also fail before termination; ULLA avoids these failure modes through landing and terminal projection. This supports the main computational motivation for replacing per-step projections with landing.

*Table 4.* **Projection overhead on Alanine.** Inference time and projection time are reported in ms/sample; overhead is projection time divided by inference time, and failure is the fraction of failed trajectories among all generated trajectories.

| Method | Inf. time | Proj. time | Overhead | Failure |
|---|---|---|---|---|
| ULLA | 0.13 | 0.01 | 7.7% | 0.0% |
| ULLA-P | 1.13 | 1.03 | 91.2% | 20.0% |

*Table 5.* **OLLA vs. ULLA on Alanine.** AUC eq. and AUC ineq. are the areas under the per-step equality- and inequality-violation curves along inference trajectories; larger values indicate greater cumulative violation.

| Method | JSD ($\psi$) | JSD (RMSD) | AUC eq. | AUC ineq. |
|--------|------|------|---------|-----------|
| OLLA | 0.0804 | 0.7830 | 3.93e-3 | 2.96e-3 |
| ULLA | 0.0372 | 0.0381 | 1.52e-4 | 2.14e-5 |

**Gain of ULLA on mixed constraints.** On complex mixed-constraint tasks, ULLA benefits from the inertial position update induced by momentum. In OLLA, tangential noise directly perturbs the position at each step, whereas in ULLA the noise acts through momentum and the position evolves through an integrated velocity. This makes trajectories more persistent and can make the landing correction less oscillatory in practice. For equality constraints, this reduces repeated small deviations around $\Sigma$; for active inequalities, once repulsion moves a sample away from the boundary, momentum tends to carry it toward the feasible interior rather than immediately diffusing back to violation.

As shown in Table 5, ULLA improves trajectory-wise feasibility on Alanine, reducing both equality- and inequality-violation AUC by more than an order of magnitude compared with OLLA, which also improves sampling quality.

**Correction terms.** We also ablate the correction terms. These terms are normal-direction, second-order geometric corrections: they improve the fidelity of the discretized dynamics to the ideal continuous-time SDE, but they are not the primary tangential drift, driving sample transport on $\Sigma$. In our experiments, the first-order landing term already suppresses constraint residuals effectively.

As shown in Table 6, adding correction terms gives negligible improvement in sample quality or constraint violation in these tested settings, while increasing computational cost and potential numerical sensitivity.

*Table 6.* **Correction-term ablation.** The on/off setting indicates whether correction terms are included.

| Task | Method | Corr. | JSD ($\psi$) | Eq. viol. | Ineq. viol. |
|------|--------|-------|------|-----------|-------------|
| Bunny-100 | ULLA | off | 0.034 | 1.59e-7 | NA |
| Bunny-100 | ULLA | on | 0.040 | 2.24e-5 | NA |
| Alanine | ULLA | off | 0.034 | 1.45e-6 | 1.20e-10 |
| Alanine | ULLA | on | 0.035 | 1.48e-6 | 3.90e-10 |

**Terminal projection.** The terminal projection mainly improves final numerical precision, rather than sample quality. During backward sampling, the landing drift is already applied at every step, so constraint residuals do not accumulate or amplify toward the final iterate; instead, they remain controlled up to discretization error. As shown in Table 7, removing this optional cleanup step on Alanine leaves sample quality nearly unchanged while increasing only the

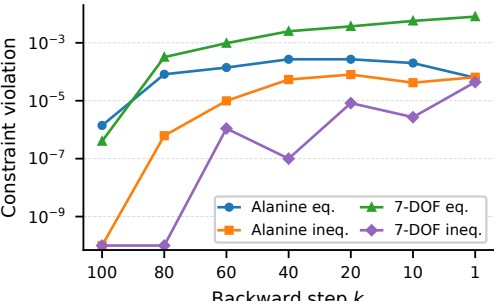

*Figure 4.* **Trajectory-wise constraint violations.** Backward sampling starts from an on-manifold prior at $k = 100$ and ends at $k = 1$.

final numerical residuals. We further report trajectory-wise diagnostics in Figure 4, which show that the practical discretized sampler is not exactly pathwise feasible but stays near-feasible throughout inference due to landing mechanism; therefore, omitting the terminal projection does not lead to large constraint violations.

*Table 7.* **No-terminal-projection ablation on Alanine.** The terminal projection mainly improves final residuals.

| Task | Terminal | JSD ($\psi$) | Eq. viol. | Ineq. viol. |
|------|----------|------|-----------|-------------|
| Alanine | off | 0.033 | 7.2e-5 | 8.0e-5 |
| Alanine | on | 0.031 | 1.4e-7 | 6.0e-10 |

**Landing and repulsion rates.** We study the effects of landing rate $\alpha$ and boundary repulsion rate $\epsilon$ on Alanine. Table 11 shows that increasing $\alpha$ improves equality constraint contraction up to a point, but overly large values can introduce discretization error induced by the landing. Similarly, too small $\epsilon$ causes boundary stickiness, while too large $\epsilon$ pushes samples too aggressively into the interior. We give a practical tuning guideline in subsection E.4, with visual examples of the boundary-repulsion effect in Figure 9.

## 6. Conclusion, Limitations, and Future Work

We introduced a landing-based constrained diffusion framework built from overdamped and underdamped Langevin dynamics for equality and inequality constraints. By replacing repeated per-step projections with inexpensive normal corrections and using faster-mixing underdamped dynamics, our models achieve competitive sample quality with substantially lower projection overhead and computational cost. As limitations, training still stores forward trajectories, so memory scales as $O(dN)$ with ambient dimension $d$ and trajectory length $N$. Also, when $\Sigma$ is disconnected, the terminal prior must cover its components appropriately; otherwise, the learned backward process may generate a skewed component distribution. A natural future direction is to develop landing-based latent constrained diffusion models with encoder-decoder architectures, where landing acts through decoder-composed constraints by the chain rule to improve high-dimensional scalability.

## Acknowledgements

Michael Muehlebach thanks the German Research Foundation for the support. Kijung Jeon and Molei Tao are grateful for partial supports by NSF Grant DMS-2513699 (KJ & MT), DOE Grants NA0004261 (MT), SC0026274 (KJ & MT), Richard Duke Fellowship (KJ & MT), and Simons Institute for the Theory of Computing at UC Berkeley (MT).

## Impact Statement

This work develops more efficient diffusion models for constrained generation. It may be useful in scientific modeling, robotics, and engineering design, where generated samples need to satisfy physical, geometric, or safety constraints. The broader impact will depend on the specific application, and we do not identify risks beyond those generally associated with generative machine learning methods.

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

TABLE OF CONTENTS

## A. Table of Key Notation, Additional Remarks, and Algorithms

*Table 8.* Table of Key Notations

| Symbol | Definition | Descriptions |
|---|---|---|
| $h$ | $h(x) = [h_1(x), \ldots, h_m(x)]^T$ | Equality constraints |
| $g$ | $g(x) = [g_1(x), \ldots, g_l(x)]^T$ | Inequality constraints |
| $\Sigma$ | $\{x \in \mathbb{R}^d \mid h(x) = 0, g(x) \leq 0\}$ | Constraint manifold |
| $I_x$ | $\{i \in [l] \mid g_i(x) \geq 0\} = \{i_1, .., i_{|I_x|}\}$ | Active index set of inequalities |
| $g_{I_x}$ | $g_{I_x}(x) = [g_{i_1}(x), ...g_{i_{|I_x|}}(x)]^T$ | Active inequality constraints |
| $J(x)$ | $\{h(x)^T, g_{i_1}(x) + \epsilon, ..., g_{i_{|I_x|}}(x) + \epsilon\}^T$ | Constraint-correction vector |
| $\Pi(x)$ | $I - \nabla J(x)^T G(x)^\dagger \nabla J(x)$ | Orthogonal projector onto $T_x\Sigma$ |
| $T_x\Sigma$ | $\{p \in \mathbb{R}^d \mid \nabla h(x)v = 0, \nabla g_{I_x}(x)v = 0\}$ | Tangent space of $\Sigma$ at $x$ |
| $T^*\Sigma$ | $\{(x, p) \in \mathbb{R}^{2d} \mid x \in \Sigma, p \in T_x\Sigma\} \simeq T\Sigma$ | Cotangent bundle of $\Sigma$ |
| $\nabla_\Sigma f$ | $\Pi(x)\nabla f(x)$ | Intrinsic gradient on $\Sigma$ |
| $\mathsf{div}_\Sigma X$ | $\mathsf{Tr}\left(\Pi(x)\nabla X(x)\right)$ | Intrinsic divergence on $\Sigma$ |
| $d\sigma_\Sigma$ | Surface (Hausdorff) measure of $\Sigma$ | Natural measure on $\Sigma$ |
| $d\sigma_{T^*\Sigma}$ | Liouville measure of $T^*\Sigma$ | Natural measure on $T^*\Sigma$ |
| $G(x)$ | $\nabla J(x)\nabla J(x)^T$ | Gram matrix |
| $\epsilon$ | Boundary repulsion rate | Controls effect of repulsion. |
| $\alpha$ | Landing rate | Controls constraint decay |
| $\gamma$ | Friction coefficient | Used in ULLA, ULLA-P |
| $\rho_\Sigma$ | Target (stationary) density on $\Sigma$ | Proportional to $\exp(-f)d\sigma_\Sigma$ |
| $\mathsf{KL}^\Sigma(\rho\|\pi)$ | $\int_\Sigma \rho \ln \frac{\rho}{\pi} d\sigma_\Sigma$ | KL-divergence on $\Sigma$ |
| $\Delta t$ | Discrete time step size | Relationship: $\Delta t = T/N$ |
| $T, N$ | Continuous and discrete terminal time | Relationship: $T = N\Delta t$ |
| $\sigma(t), \sigma_k$ | $\sigma_{\min} + \frac{t}{T}(\sigma_{\max} - \sigma_{\min}), \quad \sigma_k = \sigma(k\Delta t)$ | Noise schedule function |
| $q_t, p_t^\theta$ | Continuous time marginal densities at $t$ | Forward $q_t$, Backward $p_t^\theta$ |
| $q_k, p_k^\theta$ | Discrete time marginal densities at $k$ | Forward $q_k$, Backward $p_k^\theta$ |
| $p_N, \rho(\cdot|x)$ | Prior distribution of $x$ and $p$ ($p_N$ varies) | $\rho(\cdot|x) \sim \Pi(x)\zeta, \ \zeta \sim \mathcal{N}(0, I)$ |
| $\tilde{p}_k^{\mathrm{fwd}}$ | $\Pi(x_k)\left(\frac{x_k - x_{k-1}}{\sigma_{k-1}^2\Delta t}\right)$ | Forward approximated momentum |
| $\tilde{p}_k^{\mathrm{bwd}}$ | $\Pi(x_{k+1})\left(\frac{x_{k+2} - x_{k+1}}{\sigma_{k+2}^2\Delta t}\right)$ | Backward approximated momentum |

**Remark 3** (Comments on relaxed Constant Rank Constraint Qualification (rCRCQ)). In this remark, we further clarify the definition of the relaxed Constant Rank Constraint Qualification (rCRCQ) and its relationship with other constraint qualifications.

We first recall the definitions of the Linear Independence Constraint Qualification (LICQ), Constant Rank Constraint Qualification (CRCQ) (Janin, 1984), and its relaxed version (rCRCQ) (Minchenko & Stakhovski, 2011).

**Definition A.1** (LICQ, CRCQ, and rCRCQ; (Solodov, 2010)). Let $\Sigma := \left\{ x \in \mathbb{R}^d \mid h(x) = 0, g(x) \leq 0 \right\}$ be the feasible set, and denote $I_x = \{i \in [l] \mid g_i(x) \geq 0\}$ to be the active index set of inequalities.

- **LICQ** (Rockafellar & Wets, 1998): LICQ holds at $x \in \Sigma$ if the set $\{\nabla h_i(x)\}_{i=1}^m \cup \{\nabla g_j(x)\}_{j \in I_x}$ is linearly independent.

- **CRCQ** (Janin, 1984): CRCQ holds at $x \in \Sigma$ if there exists a neighborhood $U \subset \mathbb{R}^d$ of $x$ such that for *any* subsets of indices $I \subset [m]$ and $J \subset I_x$, the family of gradients $\{\nabla h_i(y)\}_{i \in I} \cup \{\nabla g_j(y)\}_{j \in J}$ has a constant rank for all $y \in U$.

- **rCRCQ** (Minchenko & Stakhovski, 2011): rCRCQ holds at $x \in \Sigma$ if there exists a neighborhood $U \subset \mathbb{R}^d$ of $x$ such that for any subset of active inequalities $J \subset I_x$, the family of gradients $\{\nabla h_i(y)\}_{i=1}^m \cup \{\nabla g_j(y)\}_{j \in J}$ has a constant rank for all $y \in U$.

The core reason for assuming rCRCQ lies in the stability of the SDE coefficients. It is a fundamental result in matrix analysis (Stewart, 1969) that the Moore-Penrose pseudo-inverse $A(x)^\dagger$ is continuous at a point $x_0$ if and only if the rank of $A(x)$ is constant in a neighborhood of $x_0$. By assuming rCRCQ, we guarantee that the Jacobian $\nabla J(x)$ maintains locally constant rank (even as the active set changes across strata), which ensures that the pseudo-inverse $G(x)^\dagger$ and the resulting projection operator $\Pi(x)$ are continuous and well-defined. Therefore, this guarantees the drift vector and diffusion matrix of the OLLA and ULLA dynamics to be well defined.

**Hierarchy of Constraint Qualifications.** We remark that, from the variational analysis and optimization literature (e.g., (Solodov, 2010)), rCRCQ is a strictly weaker condition than CRCQ, and CRCQ is strictly weaker than LICQ, therefore, their logical implication is as follows:

$$\text{LICQ} \implies \text{CRCQ} \implies \text{rCRCQ}.$$

In particular, rCRCQ can relax the gradient degeneracy problem appearing in LICQ.

To illustrate a case where LICQ fails due to gradient degeneracy while rCRCQ holds, consider a feasible set $\Sigma \subset \mathbb{R}^3$ representing the $z$-axis. It is defined by two equality constraints and one redundant inequality constraint with a nonlinear term:

$$h_1(x) = x_1 = 0, \quad h_2(x) = x_2 = 0,$$
$$g(x) = x_1 + x_2 + x_1^2 \leq 0.$$

On the manifold $\Sigma$ (where $x_1 = x_2 = 0$), the inequality is active since $g(0) = 0$.

- **LICQ fails:** The gradients at the origin $x = 0$ are $\nabla h_1 = (1,0,0)^T$, $\nabla h_2 = (0,1,0)^T$, and $\nabla g = (1,1,0)^T$. We observe that $\nabla g = \nabla h_1 + \nabla h_2$, meaning the gradients are linearly dependent. Thus, the Gram matrix is singular, and LICQ is violated.

- **rCRCQ holds:** Now consider the Jacobian matrix of the active constraints for an arbitrary point $x \in \mathbb{R}^3$:

$$J(x) = \begin{bmatrix} \nabla h_1(x)^T \\ \nabla h_2(x)^T \\ \nabla g(x)^T \end{bmatrix} = \begin{bmatrix} 1 & 0 & 0 \\ 0 & 1 & 0 \\ 1 + 2x_1 & 1 & 0 \end{bmatrix}.$$

Regardless of the location $x$, the rank of $J(x)$ is constant and equal to two in the entire neighborhood, satisfying rCRCQ and ensuring that the projection operator $\Pi(x)$ via the pseudo-inverse $G(x)^\dagger$ remains well-defined and continuous.

**Extended Usage in Our Framework.** While the standard definition of rCRCQ is checking the condition at a point "$x \in \Sigma$", we appropriately extend this usage in our diffusion model context.

In particular, since our landing-based discretized sampling algorithms (OLLA, ULLA) involves noise that may push particles slightly off the manifold, we implicitly assume that this constant rank property extends to an sufficiently large neighborhood of $\Sigma$ which contains all discretized samples $\{X_k\}_{k=0}^{N}$, or to the entire ambient space $\mathbb{R}^d$. This ensures that the projection operator $\Pi(x)$ and the drift terms are well-defined not just on $\Sigma$, but in the surrounding ambient space $\mathbb{R}^d$ where the landing mechanism operates.

**Remark 4** (Error decomposition and probable benefit of ULLA). Recent theoretical progress on diffusion models (e.g., (Chen et al., 2023; Strasman et al., 2025)) suggests that the total generation error can be naturally decomposed into three distinct components. In the 2-Wasserstein distance, this can be viewed as:

$$W_2(q_0, p_0^\theta) \leq \underbrace{\mathcal{E}_{\mathrm{mix}}}_{\text{Mixing}} + \underbrace{\mathcal{E}_{\mathrm{disc}}}_{\text{Discretization}} + \underbrace{\mathcal{E}_{\mathrm{score}}}_{\text{Score estimation}}$$

1. **Discretization error** ($\mathcal{E}_{\mathrm{disc}}$) **& Mixing error** ($\mathcal{E}_{\mathrm{mix}}$): Regarding discretization, our ULLA implementation employs a memory-efficient first-order splitting scheme; thus, both ULLA and the baseline OLLA share the same convergence order with respect to the step size. However, ULLA gains a significant advantage in the *mixing error* due to the ballistic behavior of underdamped dynamics, which theoretically accelerates convergence to $\mathcal{O}(\sqrt{d}/\epsilon)$ compared to the diffusive $\mathcal{O}(d/\epsilon^2)$ of overdamped dynamics (Cheng et al., 2018; Ma et al., 2021). This allows for a significantly smaller trajectory length $N$ to reach the stationary prior, thereby reducing the computational cost for training and storage.

2. **Score estimation error** ($\mathcal{E}_{\mathrm{score}}$): Employing a constrained forward process with the proposed landing mechanism allows the model to faithfully capture the intrinsic geometry of $\Sigma$. Crucially, because the landing mechanism analytically handles the ill-conditioned normal component, the score network $s_t^\theta$ is only required to learn the smoother tangential component $\Pi(x)s_t^{\mathrm{true}}$ (Liu et al., 2025). Adopting underdamped dynamics introduces a trade-off: learning on the extended phase space potentially increases regression complexity compared to position-only models. However, since empirical data distributions are usually supported on some data manifold $\Sigma_{\mathrm{data}} \subset \Sigma$, standard overdamped models suffer from score singularities where $\|s_t^{\mathrm{true}}\|_2 \propto \mathcal{O}(1/t)$ near $t = 0$ (Liu et al., 2025). In contrast, as highlighted in Dockhorn et al. (2022), underdamped dynamics yield a smoother training objective that bypasses this singularity problem due to the existence of momentum variable.

Figure 5 provides empirical evidence of this effect on the volcano experiment. The underdamped model exhibits Jacobian norms that are several orders of magnitude smaller across all times and, in particular, does *not* show the sharp blow-up near $t \approx 0$ that appears in the overdamped case. This suggests that ULLA provides a numerically better-conditioned score regression problem, which can potentially reduce $\mathcal{E}_{\mathrm{score}}$ in practice.

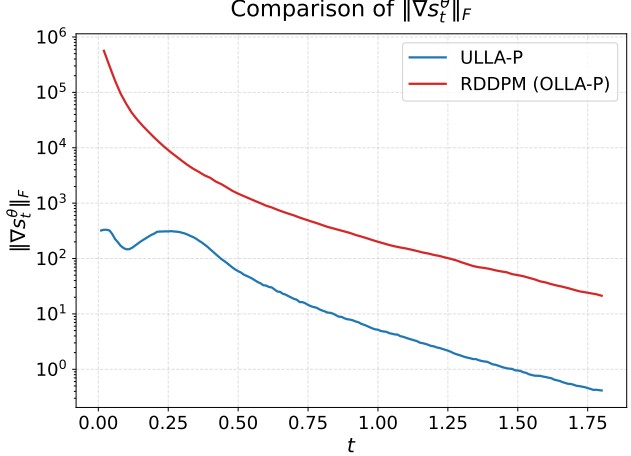

*Figure 5.* Comparison of the Frobenius norm of the score Jacobian $\|\nabla s_t^\theta\|_F$ over time on the volcano experiment. The overdamped RDDPM (OLLA-P) baseline (red) exhibits very large Jacobian norms and a pronounced singular behavior as $t \to 0$, while the underdamped ULLA-P sampler (blue) remains several orders of magnitude smaller and shows no blow-up near $t \approx 0$. Jacobian is taken over to position $x$ for the overdamped and to momentum $p$ for the underdamped.

**Remark 5** (Explicit versus implicit landing). The main algorithms and analysis in this paper focus on explicit landing. Nevertheless, one may also use an implicit landing variant in practice, where the tangential proposal is formed first and the landing correction is evaluated at the proposed point. We explain the distinction in the equality-only case $\Sigma = \{x \in \mathbb{R}^d : h(x) = 0\}$ with $\kappa = 0$. Assume $h$ is $C^2$ and $\nabla h$ has full row rank near $\Sigma$.

For OLLA, the tangential part of the forward update can be written as

$$\tilde{x}_{k+1} = x_k + u_k^O, \qquad u_k^O = -\frac{\sigma_k^2 \Delta t}{2} \Pi(x_k) \nabla f(x_k) + \sigma_k \sqrt{\Delta t} \Pi(x_k) \zeta_k.$$

For ULLA, the corresponding tangential/inertial proposal is

$$\tilde{x}_{k+1} = x_k + u_k^U, \qquad u_k^U = \sigma_k^2 \Delta t \Pi(x_k) \left( a_k \tilde{p}_k^{\text{fwd}} - \sigma_k^2 \Delta t \nabla f(x_k) \right) + \sigma_k^2 \Delta t \sqrt{1 - a_k^2} \Pi(x_k) \zeta_k.$$

In both cases, the proposal increment is tangent to the current level set, i.e., $\nabla h(x_k) u_k^O = 0$ and $\nabla h(x_k) u_k^U = 0$. Let $u_k$ denote either proposal increment, and let $\beta_k$ be the scalar prefactor multiplying $(\nabla h^\top G^\dagger h)$ in the corresponding landing update. The explicit landing update evaluates the normal correction at $x_k$, giving $x_{k+1}^{\text{exp}} = \tilde{x}_{k+1} - \beta_k (\nabla h^\top G^\dagger h)(x_k)$, whereas the implicit landing update evaluates the correction at the proposal, giving $x_{k+1}^{\text{imp}} = \tilde{x}_{k+1} - \beta_k (\nabla h^\top G^\dagger h)(\tilde{x}_{k+1})$.

---

**Lemma A.1** (Local residual comparison). *For a bounded $\beta_k$ and a sufficiently small tangential proposal $u_k$, the explicit update satisfies*

$$h(x_{k+1}^{\text{exp}}) = (1 - \beta_k) h(x_k) + O\left( \|h(x_k)\|^2 + \|u_k\|^2 \right),$$

*whereas the implicit update satisfies*

$$h(x_{k+1}^{\text{imp}}) = (1 - \beta_k) h(x_k) + O\left( \|h(x_k)\|^2 + |1 - \beta_k| \, \|u_k\|^2 + \|u_k\|^4 \right).$$

*In particular, when $\beta_k = 1$,*

$$\|h(x_{k+1}^{\text{exp}})\| = O\left( \|h(x_k)\|^2 + \|u_k\|^2 \right), \qquad \|h(x_{k+1}^{\text{imp}})\| = O\left( \|h(x_k)\|^2 + \|u_k\|^4 \right).$$

---

*Proof.* Set $L(x) = (\nabla h^\top G^\dagger h)(x)$. Since $\nabla h$ has full row rank, we have

$$\nabla h(x) L(x) = h(x), \qquad \|L(x)\| \le C \|h(x)\|$$

for some constant $C$ near $\Sigma$. For explicit landing, Taylor expansion around $x_k$ gives

$$\begin{aligned} h(x_{k+1}^{\text{exp}}) &= h(x_k + u_k - \beta_k L(x_k)) \\ &= h(x_k) + \nabla h(x_k) u_k - \beta_k \nabla h(x_k) L(x_k) + O(\|u_k\|^2 + \|u_k\| \|L(x_k)\| + \|L(x_k)\|^2) \\ &= (1 - \beta_k) h(x_k) + O(\|h(x_k)\|^2 + \|u_k\|^2), \end{aligned}$$

where we used $\nabla h(x_k) u_k = 0$. For implicit landing, Taylor expansion around $\tilde{x}_{k+1} = x_k + u_k$ gives

$$h(x_{k+1}^{\text{imp}}) = h(\tilde{x}_{k+1}) - \beta_k \nabla h(\tilde{x}_{k+1}) L(\tilde{x}_{k+1}) + O(\|L(\tilde{x}_{k+1})\|^2) = (1 - \beta_k) h(\tilde{x}_{k+1}) + O(\|h(\tilde{x}_{k+1})\|^2).$$

Moreover,

$$h(\tilde{x}_{k+1}) = h(x_k + u_k) = h(x_k) + O(\|u_k\|^2),$$

again using $\nabla h(x_k) u_k = 0$. Substituting this into the previous equation yields

$$h(x_{k+1}^{\text{imp}}) = (1 - \beta_k) h(x_k) + O(|1 - \beta_k| \, \|u_k\|^2) + O\left( \|h(x_k)\|^2 + \|u_k\|^4 \right),$$

which proves the claim. $\qquad \square$

---

This shows that explicit landing contracts the residual evaluated at the current base point $x_k$, while implicit landing contracts the residual after the proposal has been formed. Thus, implicit landing directly controls the infeasibility created by the current tangential proposal. Also, implicit landing can obtain a lower residual order in practice at the cost of one additional proposal-point evaluation of $h$, $\nabla h$, and $G^\dagger$. When using implicit landing, we adapt $\alpha$ across steps so that $\beta_k = 1$. We use this implicit variant for all OLLA/ULLA experiments on $S^2$, and for OLLA on the mesh datasets.

---

**Algorithm 1** Full Diffusion Pipeline for OLLA / OLLA-P (=RDDPM (Liu et al., 2026))

---

1: **Input:** Data distribution $q_{\text{data}}$, initial score network $s_k^\theta(x)$, number of steps $N$, terminal step $N$, landing rate $\alpha$, boundary repulsion rate $\epsilon$, constraints $h, g$.
2: **Options:** mode $\in \{\text{OLLA}, \text{OLLA-P}\}$, use_curvature $\in \{\text{True}, \text{False}\}$
3: **Output:** Trained score network $s_k^\theta(x)$, generated sample $x_0$

---

**Part 1: Forward Process (Noising)**        ▷ Run Forward Process per $l_f$ iterations

4: Sample $x_0 \sim q_{\text{data}} = q_0$
5: **for** $k \in \{0, \dots, N-1\}$ **do**
6:     Compute $\nabla f(x_k), J(x_k), \nabla J(x_k), G(x_k)^\dagger, \Pi(x_k)$
7:     $\bar{\mu}_k^o(x_k) \leftarrow x_k - \frac{1}{2}\sigma_k^2 \Delta t \Pi(x_k)\nabla f(x_k)$        ▷ Prior drift term
8:     **if** mode = OLLA-P **then**        ▷ Projection-based noising
9:        $x_{k+1} \leftarrow \text{Proj}_\Sigma(\bar{\mu}_k^o(x_k) + \sigma_k\sqrt{\Delta t}\Pi(x_k)\zeta_k), \quad \zeta_k \sim \mathcal{N}(0, I_d)$
10:     **else**        ▷ Landing-based noising (OLLA)
11:        $\mathcal{H}(x_k) \leftarrow 0$
12:        **if** use_curvature **then**
13:          $\text{Tr} \leftarrow [\text{Tr}(\Pi\nabla^2 J_1), \dots, \text{Tr}(\Pi\nabla^2 J_{m+|I_{x_k}|})]^T$
14:          $\mathcal{H}(x_k) \leftarrow -\nabla J(x_k)^T G(x_k)^\dagger \text{Tr}$
15:        **end if**
16:        $L_k(x_k) \leftarrow -\alpha\sigma_k^2\Delta t\nabla J(x_k)^T G(x_k)^\dagger J(x_k)$        ▷ Landing term
17:        $\kappa_k^O(x_k) \leftarrow \frac{1}{2}\sigma_k^2\Delta t\mathcal{H}(x_k)$        ▷ Curvature term
18:        $x_{k+1} \leftarrow \bar{\mu}_k^O(x_k) + L_k(x_k) + \kappa_k^O(x_k) + \sigma_k\sqrt{\Delta t}\Pi(x_k)\zeta_k, \quad \zeta_k \sim \mathcal{N}(0, I_d)$
19:     **end if**
20: **end for**
21: $x_N \leftarrow \text{Proj}_\Sigma(x_N)$        ▷ Terminal projection by Newton's method
22: Store trajectory $\{x_k\}_{k=0}^N$

---

**Part 2: Score Network Training**

23: $L_{\text{CWPM}}^{\text{over}}(\theta) \leftarrow \sum_{k=0}^{N-1} \frac{\|\Pi(x_{k+1})(x_k - \mu_{k+1}^o(x_{k+1}))\|^2}{2\sigma_{k+1}^2\Delta t}$
24: Update network parameters: $\theta \leftarrow \theta - \eta\nabla_\theta L_{\text{CWPM}}^{\text{over}}(\theta)$        ▷ $\eta$ is the learning rate

---

**Part 3: Backward Process (Sampling)**

25: Sample $x_N \sim p_N$ (prior)
26: **for** $k \in \{N, \dots, 1\}$ **do**
27:     Compute $\nabla f(x_k), J(x_k), \nabla J(x_k), G(x_k)^\dagger, \Pi(x_k)$
28:     $\mu_k^o(x_k) \leftarrow x_k + \frac{1}{2}\sigma_k^2\Delta t\Pi(x_k)[\nabla f(x_k) + s_k^\theta(x_k)]$
29:     **if** mode = OLLA-P **then**        ▷ Projection-based variant
30:        $x_{k-1} \leftarrow \text{Proj}_\Sigma\left(\mu_k^o(x_k) + \sigma_k\sqrt{\Delta t}\Pi(x_k)\zeta_k\right)$
31:     **else**        ▷ Landing-based variant (OLLA)
32:        $\mathcal{H}(x_k) \leftarrow 0$
33:        **if** use_curvature **then**
34:          $\text{Tr} \leftarrow [\text{Tr}(\Pi\nabla^2 J_1), \dots, \text{Tr}(\Pi\nabla^2 J_{m+|I_{x_k}|})]^T$
35:          $\mathcal{H}(x_k) \leftarrow -\nabla J(x_k)^T G(x_k)^\dagger \text{Tr}$
36:        **end if**
37:        $L_k(x_k) \leftarrow -\alpha\sigma_k^2\Delta t\nabla J(x_k)^T G(x_k)^\dagger J(x_k)$
38:        $\kappa_k^o(x_k) \leftarrow \frac{1}{2}\sigma_k^2\Delta t\mathcal{H}(x_k)$
39:        $x_{k-1} \leftarrow \mu_k^O(x_k) + L_k(x_k) + \kappa_k^o(x_k) + \sigma_k\sqrt{\Delta t}\Pi(x_k)\zeta_k$
40:     **end if**
41: **end for**
42: $x_0 \leftarrow \text{Proj}_\Sigma(x_0)$        ▷ Terminal projection by Newton's method
43: **return** $x_0$

---

---

**Algorithm 2** Full Diffusion Pipeline for ULLA / ULLA-P

---

1: **Input:** Data distribution $q_{\text{data}}$, initial score network $s_k^\theta(x,p)$, number of steps $N$, terminal time $T$, landing rate $\alpha$, boundary repulsion rate $\epsilon$, friction $\gamma$, constraints $h, g$.
2: **Options:** $\texttt{mode} \in \{\text{ULLA}, \text{ULLA-P}\}$, $\texttt{use\_curvature} \in \{\text{True}, \text{False}\}$
3: **Output:** Trained score network $s_k^\theta(x,p)$, generated sample $x_0$

---

    **Part 1: Forward Process (Noising)**                  ▷ Run Forward Process per $l_f$ iterations
4: Sample $x_0 \sim q_{\text{data}}$, $p_0 \sim \mathcal{N}(0, I_d)$ and set $\tilde{p}_0 \leftarrow \Pi(x_0)p_0$
5: $x_{-1} \leftarrow x_0 - \sigma_{-1}^2 \Delta t \tilde{p}_0 \quad (\sigma_{-1} := \sigma_0)$            ▷ Create pseudo-point for first momentum
6: **for** $k \in \{0, \ldots, N-1\}$ **do**
7:      Compute $\nabla f(x_k), J(x_k), \nabla J(x_k), G(x_k)^\dagger, \Pi(x_k)$
8:      $\tilde{p}_k^{\text{fwd}} \leftarrow \Pi(x_k)\left(\frac{x_k - x_{k-1}}{\sigma_{k-1}^2 \Delta t}\right)$            ▷ Approximate momentum from positions
9:      $a_k \leftarrow e^{-\gamma \sigma_k^2 \Delta t}$
10:      $\bar{\mu}_k^u(x_k, \tilde{p}_k^{\text{fwd}}) \leftarrow x_k + \sigma_k^2 \Delta t \Pi(x_k)[a_k \tilde{p}_k^{\text{fwd}} - \sigma_k^2 \Delta t \nabla f(x_k)]$      ▷ Prior drift term
11:      **if** $\texttt{mode} = \text{ULLA-P}$ **then**            ▷ Projection-based noising
12:          $x_{k+1} \leftarrow \text{Proj}_\Sigma(\bar{\mu}_k^u(x_k, \tilde{p}_k^{\text{fwd}}) + \sigma_k^2 \Delta t \sqrt{1 - a_k^2} \Pi(x_k)\zeta_k), \quad \zeta_k \sim \mathcal{N}(0, I_d)$
13:      **else**            ▷ Landing-based noising (ULLA)
14:          $\mathcal{H}_1(x_k, \tilde{p}_k^{\text{fwd}}), \mathcal{H}_2(x_k, \tilde{p}_k^{\text{fwd}}) \leftarrow 0, 0$
15:          **if** $\texttt{use\_curvature}$ **then**
16:              Compute $\mathcal{H}_1, \mathcal{H}_2$ using $x_k, \tilde{p}_k^{\text{fwd}}$
17:          **end if**
18:          $L_k(x_k) \leftarrow -\alpha \sigma_k^2 \Delta t \nabla J(x_k)^T G(x_k)^\dagger J(x_k)$      ▷ Landing term
19:          $\kappa_{k,\text{fwd}}^U(x_k, \tilde{p}_k^{\text{fwd}}) \leftarrow -\sigma_k^4 \Delta t^2 \nabla J(x_k)^T G^\dagger(x_k)[\mathcal{H}_1 - \alpha \mathcal{H}_2]$      ▷ Curvature term
20:          $x_{k+1} \leftarrow \bar{\mu}_k^u(x_k, \tilde{p}_k^{\text{fwd}}) + L_k(x_k) + \kappa_{k,\text{fwd}}^U(x_k, \tilde{p}_k^{\text{fwd}}) + \sigma_k^2 \Delta t \sqrt{1 - a_k^2} \Pi(x_k)\zeta_k$
21:      **end if**
22: **end for**
23: $x_N \leftarrow \text{Proj}_\Sigma(x_N)$            ▷ Terminal projection by Newton's method
24: Store trajectory $\{x_k\}_{k=0}^N$

---

    **Part 2: Score Network Training**
25: $L_{\text{CWPM}}^{\text{under}}(\theta) \leftarrow \sum_{k=0}^{N-1} \frac{\|\Pi(x_{k+1})(x_k - \mu_{k+1}^u(x_{k+1}, x_{k+2}))\|^2}{2\sigma_{k+1}^4 \Delta t^2 (1 - a_{k+1}^2)}$
26: Update network parameters: $\theta \leftarrow \theta - \eta \nabla_\theta L_{\text{CWPM}}^{\text{under}}(\theta)$            ▷ $\eta$ is the learning rate

---

    **Part 3: Backward Process (Sampling)**
27: Sample $x_N \sim p_N$ (prior), $p_N \sim \mathcal{N}(0, I_d)$. Set $\tilde{p}_N \leftarrow \Pi(x_N)p_N$.
28: $x_{N+1} \leftarrow x_N + \sigma_N^2 \Delta t \tilde{p}_N$            ▷ Create pseudo-point for terminal momentum
29: **for** $k \in \{N, \ldots, 1\}$ **do**
30:      Compute $\nabla f(x_k), J(x_k), \nabla J(x_k), G(x_k)^\dagger, \Pi(x_k)$
31:      $\tilde{p}_k \leftarrow \Pi(x_k)\left(\frac{x_{k+1} - x_k}{\sigma_{k+1}^2 \Delta t}\right)$
32:      $a_k \leftarrow e^{-\gamma \sigma_k^2 \Delta t}$
33:      $\mu_k^u(x_k, \tilde{p}_k^{\text{bwd}}) \leftarrow x_k - \sigma_k^2 \Delta t \Pi(x_k)[a_k \tilde{p}_k^{\text{bwd}} + \sigma_k^2 \Delta t (\nabla f(x_k) + s_k^\theta(x_k, \tilde{p}_k^{\text{bwd}}))]$
34:      **if** $\texttt{mode} = \text{ULLA-P}$ **then**            ▷ Projection-based variant
35:          $x_{k-1} \leftarrow \text{Proj}_\Sigma\left(\mu_k^u(x_k, \tilde{p}_k^{\text{bwd}}) + \sigma_k^2 \Delta t \sqrt{1 - a_k^2} \Pi(x_k)\zeta_k\right)$
36:      **else**            ▷ Landing-based variant (ULLA)
37:          $\mathcal{H}_1(x_k, \tilde{p}_k^{\text{bwd}}), \mathcal{H}_2(x_k, \tilde{p}_k^{\text{bwd}}) \leftarrow 0, 0$
38:          **if** $\texttt{use\_curvature}$ **then**
39:              Compute $\mathcal{H}_1, \mathcal{H}_2$ using $x_k, \tilde{p}_k^{\text{bwd}}$
40:          **end if**
41:          $L_k(x_k) \leftarrow -\alpha \sigma_k^2 \Delta t \nabla J(x_k)^T G(x_k)^\dagger J(x_k)$
42:          $\kappa_k^U(x_k, \tilde{p}_k^{\text{bwd}}) \leftarrow -\sigma_k^4 \Delta t^2 \nabla J(x_k)^T G^\dagger(x_k)[\mathcal{H}_1 + \alpha \mathcal{H}_2]$
43:          $x_{k-1} \leftarrow \mu_k^u(x_k, \tilde{p}_k^{\text{bwd}}) + L_k(x_k) + \kappa_k^U(x_k, \tilde{p}_k^{\text{bwd}}) + \sigma_k^2 \Delta t \sqrt{1 - a_k^2} \Pi(x_k)\zeta_k$
44:      **end if**
45: **end for**
46: $x_0 \leftarrow \text{Proj}_\Sigma(x_0)$            ▷ Terminal projection by Newton's method
47: **return** $x_0$

---

# B. Constrained Langevin Dynamics

In this section, we review the constrained Langevin dynamics and introduce their landing versions.

## B.1. Construction of OLLA

**Notations and Background for overdamped setup.** We consider the constrained set

$$\Sigma := \left\{ x \in \mathbb{R}^d \mid h(x) = 0, g(x) \leq 0 \right\},$$

assumed to be a stratified manifold $\mathbb{R}^d$ with rCRCQ satisfied. We define the stacked active constraint map and its Jacobian as

$$J(x) := [h(x), g_{I_x}(x) + \epsilon] \in \mathbb{R}^{m+|I_x|}, \quad \nabla J(x) \in \mathbb{R}^{(m+|I_x|) \times d}$$

where $I_x$ denotes the set of active inequality constraints, i.e., $I_x := \{i \in [l] \mid g_i(x) \geq 0\}$. Denote the Gram matrix $G(x) := \nabla J(x) \nabla J(x)^T \in \mathbb{R}^{(m+|I_x|) \times (m+|I_x|)}$. The orthogonal projector onto the tangent space of $T_x \Sigma := \left\{ p \in \mathbb{R}^d \mid \nabla J(x) p = 0 \right\}$ is given by $\Pi(x) = I - \nabla J(x)^T G(x)^\dagger \nabla J(x)$. On this manifold $\Sigma$, all intrinsic differential operators are defined via the projector $\Pi$. For a smooth scalar function $\phi$ and smooth vector field $X$ on $\Sigma$, we have

$$\nabla_\Sigma \phi := \Pi \nabla \phi, \quad \mathsf{div}_\Sigma(X) = \mathsf{Tr}\left(\Pi \nabla X\right)$$

and the Laplace-Betrami operator is $\Delta_\Sigma \phi := \mathsf{div}_\Sigma(\nabla_\Sigma \phi)$, where $\Delta$ denotes ambient Euclidean gradient or Jacobian. For comprehensive backgrounds on constrained overdamped Langevin dynamics, see Chapter 3.2 in Lelièvre et al. (2010).

---

**Proposition B.1** (Construction of OLLA). *Consider the following Lagrangian-form constrained overdamped Langevin dynamics:*

$$dX_t = -\frac{1}{2}\sigma(t)^2 \nabla f(X_t)dt + \sigma(t) \circ dW_t + \nabla J(X_t)^T d\lambda_t \tag{3}$$

*where $d\lambda_t$ is the adapted process such that $dJ(X_t) = -\alpha\sigma(t)^2 J(X_t)$. The explicit minimum norm solution of $d\lambda_t$ is given by*

$$d\lambda_t = G^\dagger(X_t)\left[-\alpha\sigma(t)^2 J(X_t)dt + \frac{1}{2}\sigma(t)^2 \nabla J(X_t)\nabla f(X_t)dt - \sigma(t)\nabla J(X_t) \circ dW_t\right],$$

*with $G(X_t) := \nabla J(X_t)\nabla J(X_t)^T$ defined as the Gram matrix. Therefore, the closed form SDE of (3) is as follows:*

$$dX_t = -\left[\frac{\sigma(t)^2}{2}\Pi(X_t)\nabla f(X_t) + \alpha\sigma(t)^2\nabla J(X_t)^T G^\dagger(X_t)J(X_t)\right]dt + \frac{\sigma(t)^2}{2}\mathcal{H}(X_t)dt$$
$$+ \sigma(t)\Pi(X_t)dW_t,$$

*where $\mathcal{H}$ is the mean curvature correction term defined as*

$$\mathcal{H}(x) := -\nabla J(x)^T G^\dagger(x)\left[\mathsf{Tr}\left(\nabla^2 J_1(x)\Pi(x)\right), ..., \mathsf{Tr}\left(\nabla^2 J_{m+|I_x|}(x)\Pi(x)\right)\right]^T.$$

---

*Proof.* From the Stratonovich chain rule, it holds that

$$-\alpha\sigma(t)^2 J(X_t)dt = \nabla J(X_t) \circ dX_t = -\frac{1}{2}\sigma(t)^2\nabla J(X_t)\nabla f(X_t)dt + \sigma(t)\nabla J(X_t) \circ dW_t + G(X_t)d\lambda_t$$

Among the many solutions $d\lambda_t$ satisfying the above equation, we choose the (unique) minimum norm solution of $d\lambda_t$ process:

$$d\lambda_t = G^\dagger(X_t)\left[-\alpha\sigma(t)^2 J(X_t)dt + \frac{1}{2}\sigma(t)^2 \nabla J(X_t)\nabla f(X_t)dt - \sigma(t)\nabla J(X_t) \circ dW_t\right].$$

We remark that $\nabla J(X_t)^T d\lambda_t$ is unique regardless of the choice of solution $d\lambda_t$. Substituting back to the SDE (3) gives the following Stratonovich version of the unique closed-form SDE:

$$dX_t = -\left[\frac{1}{2}\sigma(t)^2\Pi(X_t)\nabla f(X_t) + \alpha\sigma(t)^2\nabla J(X_t)^T G^\dagger(X_t)J(X_t)\right]dt + \sigma(t)\Pi(X_t) \circ dW_t.$$

To recover the Itô version of the closed-form SDE, we observe that the Itô-Stratonovich correction term coincides with the mean curvature term of a stratum $\Sigma_{I_x} := \{x \in \mathbb{R}^d \mid J(x) = 0\}$ and its representation is given by

$$\frac{1}{2}\nabla\left(\sigma(t)\Pi\right)\left(\sigma(t)\Pi\right) = \frac{\sigma(t)^2}{2}\nabla(\Pi)\Pi = \frac{\sigma(t)^2}{2}\sum_{k=1}^{d}\nabla(\Pi_k)\Pi_k.$$

From the same tensor-calculus technique of Equation 3.46 in Lelièvre et al. (2010), we observe that $(\nabla\Pi)\Pi$ is given by

$$\nabla\Pi(x)\Pi(x) = -\nabla J(x)^T G^\dagger(x)\left[\mathsf{Tr}\left(\nabla^2 J_1(x)\Pi(x)\right), ..., \mathsf{Tr}\left(\nabla^2 J_{m+|I_x|}(x)\Pi(x)\right)\right]^T.$$

Therefore, this gives the following Itô version of the closed-form SDE:

$$dX_t = -\left[\frac{\sigma(t)^2}{2}\Pi(X_t)\nabla f(X_t) + \alpha\sigma(t)^2\nabla J(X_t)^T G^\dagger(X_t)J(X_t)\right]dt + \frac{\sigma(t)^2}{2}\mathcal{H}(X_t)dt + \sigma(t)\Pi(X_t)dW_t.$$

$\square$

**Theorem B.1** (Fokker-Planck equation (Chirikjian, 2009; Huang et al., 2022) and the generator (Ikeda & Watanabe, 2014) on Riemannian manifold). *Let $Z_t \in \Sigma$ be a stochastic process following the SDE:*

$$dZ_t = V_0 dt + \sum_{k=1}^{d}V_k \circ dB_t^k,$$

*where $V_0, V_k$ are smooth vector fields on $\Sigma$ for each $k \in [d]$ and $B_t^k$ are $k$th components of Brownian motion $B_t$. Then, the law $\rho_t$ of the stochastic process $Z_t$ satisfies the following Fokker-Planck equation:*

$$\partial_t \rho_t = -\mathsf{div}_\Sigma(\rho_t V_0) + \frac{1}{2}\sum_{k=1}^{d}\mathsf{div}_\Sigma(\mathsf{div}_\Sigma(\rho_t V_k)V_k).$$

*Also, the generator of $\mathcal{L}$ of the corresponding SDE is provided as*

$$\mathcal{L}\phi = V_0\phi + \frac{1}{2}\sum_{k=1}^{d}V_k(V_k\phi)$$

*for any smooth function $\phi$ on $\Sigma$.*

**Lemma B.1** (Boundary condition of OLLA). *Assuming $X_0 \in \Sigma$, OLLA (3) satisfies the following boundary condition on $\partial\Sigma$ and property for $t \geq 0$:*

$$(1)\ \ \langle J_t(x), n(x)\rangle = 0 \quad \text{a.e. on } \partial\Sigma, \qquad (2)\ \ X_t \in \Sigma \quad \text{a.s}$$

*where $J_t(x)$ is the probability current density defined by $\partial_t \rho_t = -\mathsf{div}_\Sigma(J_t)$ and $n(x)$ is the outward unit normal vector on $\partial\Sigma$.*

*Proof.* First, we show that $\mathbb{P}(g_k(X_t) \leq 0) = 1$ for $t \geq 0$ and $k \in [l]$. To show this, we define a convex smooth violation penalty function $\Psi_\delta^k(x) : \mathbb{R}^d \to \mathbb{R}$ as follows:

$$\Psi_\delta^k(x) := \phi_\delta(g_k(x)), \qquad \phi_\delta(r) := \begin{cases} \frac{r^2}{2\delta}, & 0 \leq r \leq \delta \\ r - \delta/2, & r \geq \delta \\ 0 & r < 0. \end{cases}$$

Then, $\phi_\delta$ is convex, $C^1$, and satisfies

$$\phi_\delta \downarrow (r)_+, \quad \phi_\delta'(r) \to \mathbb{1}_{\{r>0\}}, \quad \text{as } \delta \downarrow 0$$

with $(r)^+ := \max\{r, 0\}$. Now, we observe that, on $\{g_k \geq 0\}$, the Stratonovich chain rule (as in Lemma B.4) gives

$$dg_k(X_t) = -\alpha\sigma(t)^2\left(g_k(X_t) + \epsilon\right)dt.$$

Therefore, applying Itô's lemma on $\phi_\delta(g_i(X_t))$ gives

$$d\phi_\delta(g_k(X_t)) = \phi_\delta'(g_k(X_t))dg_k(X_t) + \frac{1}{2}\phi_\delta''(g_k(X_t))\underbrace{d\langle g_k(X_t), g_k(X_t)\rangle_t}_{\text{Quadratic variation=0}}$$

$$= -\alpha\sigma(t)^2(g_k(X_t) + \epsilon)\phi_\delta'(g_k(X_t))dt.$$

For the case $\{g_k < 0\}$, it trivially holds that $\phi_\delta(g_k(X_t)) = 0$ with $\phi_\delta'(g_k(X_t)) = 0$. Therefore, the above observations lead to the following relation for $t \geq 0$:

$$\frac{d}{dt}\mathbb{E}[\Psi_\delta^k(X_t)] = \frac{d}{dt}\mathbb{E}[\phi_\delta(g_k(X_t))] = -\alpha\sigma(t)^2\mathbb{E}\left[(g_k(X_t) + \epsilon)\phi_\delta'(g_k(X_t))\right].$$

At this moment, we note that the non-decreasing property of $\phi_\delta'(r)$ implies, for $\forall r \geq 0$,

$$\phi_\delta(r) = \int_0^r \phi_\delta'(s)ds \leq \int_0^r \phi_\delta'(r)ds \leq (r + \epsilon)\phi_\delta'(r) \quad \Rightarrow \quad \Psi_\delta^k(x) \leq (g_k(x) + \epsilon)\phi_\delta'(g_k(x))$$

where the inequality $\phi_\delta(r) \leq (r + \epsilon)\phi_\delta'(r)$ also holds trivially for $r < 0$. Hence, we finally have

$$\frac{d}{dt}\mathbb{E}[\Psi_\delta^k(X_t)] = -\alpha\sigma(t)^2\mathbb{E}\left[(g_k(X_t) + \epsilon)\phi_\delta'(g_k(X_t))\right] \leq -\alpha\sigma(t)^2\mathbb{E}[\Psi_\delta(X_t)]$$

and the Grönwall inequality gives

$$0 \leq \mathbb{E}[\Psi_\delta^k(X_t)] \leq e^{-\alpha\int_0^t \sigma(s)^2 ds}\mathbb{E}[\Psi_\delta^k(X_0)] = 0 \quad (\because X_0 \in \Sigma)$$

which leads to $(g_k(X_t))_+ = 0 \Rightarrow g_k(X_t) \leq 0$ for $k \in [l]$ by letting $\delta \downarrow 0$ and applying the monotone convergence theorem. This proves $\mathbb{P}(g(X_t) \leq 0) = 1$ for $t \geq 0$ and $X_t \in \Sigma$ a.s.

Next, we prove $\langle J_t(x), n(x)\rangle = 0$ for $x \in \partial\Sigma$. We first observe that Theorem B.1 gives the following Fokker-Planck equation for $g(x) > 0$:

$$\partial_t\rho_t = -\text{div}_\Sigma\left(\rho_t\left[-\frac{\sigma^2}{2}\nabla_\Sigma f - \alpha\sigma^2\nabla J^T G^\dagger J\right]\right) + \frac{\sigma^2}{2}\sum_{k=1}^d \text{div}_\Sigma(\text{div}_\Sigma(\rho_t f_k)f_k)$$

$$= -\text{div}_\Sigma\left(\rho_t\left[-\frac{\sigma^2}{2}\nabla_\Sigma f - \alpha\sigma^2\nabla J^T G^\dagger J\right] - \frac{\sigma^2}{2}\nabla_\Sigma\rho_t\right),$$

where $f_k := \Pi e_k$ and $e_k$ being the $k$-th standard basis of $\mathbb{R}^d$. Also, we remark that the last equality holds using the property:

$$\sum_{k=1}^d \text{div}_\Sigma(\rho_t f_k)f_k = \sum_{k=1}^d \langle\nabla_\Sigma\rho_t, f_k\rangle f_k + \rho_t\underbrace{\sum_{k=1}^d \text{div}_\Sigma(f_k)f_k}_{=0} = \nabla_\Sigma\rho_t.$$

Therefore, the probability current density $J_t$ is given as follows

$$J_t = -\rho_t\left[\frac{\sigma^2}{2}\nabla_\Sigma f + \alpha\sigma^2\nabla J^T G^\dagger J\right] - \frac{\sigma^2}{2}\nabla_\Sigma\rho_t,$$

and we have

$$0 = \frac{d}{dt}\int_\Sigma \rho_t(x)d\sigma_\Sigma = -\int_\Sigma \text{div}_\Sigma(J_t(x))d\sigma_\Sigma = -\int_{\partial\Sigma} \langle J_t(x), n(x)\rangle d\sigma_{\partial\Sigma}$$

$$= \int_{\partial\Sigma} \alpha\rho_t\sigma^2\underbrace{\langle\nabla J^T G^\dagger J, n\rangle}_{>0}d\sigma_{\partial\Sigma} \geq 0.$$

This implies $\rho_t = 0$ a.e on $\partial\Sigma$ and the following boundary condition holds almost everywhere on $\partial\Sigma$

$$\langle J_t, n \rangle = \langle \rho_t \left[ -\frac{\sigma^2}{2} \nabla_\Sigma f - \alpha\sigma^2 \nabla J^T G^\dagger J \right] - \frac{\sigma^2}{2} \nabla_\Sigma \rho_t, n \rangle = -\alpha\sigma^2 \rho_t \langle \nabla J^T G^\dagger J, n \rangle = 0.$$

$\square$

**Lemma B.2** (Huang et al. (2022)). *Let $\{f_k\}_{k=1}^d$ be a set of vectors defined by $f_k = \Pi(x)e_k$, where $\Pi(x)$ is the orthogonal projector onto $T_x\Sigma$ and $e_k$ is the kth standard basis vector of $\mathbb{R}^d$. Then, it holds that*

$$\sum_{k=1}^d (div_\Sigma f_k) f_k = 0.$$

*Proof.* Let $r$ be the rank of the $\nabla J(x)$ and $\{n_1(x), \ldots, n_r(x)\}$ be an orthonormal basis of $\text{Im}(\nabla J(x)^T)$. Since $\Pi(x)$ is the orthogonal projector onto the tangent space, it can be written using the projector onto the normal space as:

$$\Pi(x) = I - \sum_{l=1}^r n_l(x)n_l(x)^T.$$

Note that by definition, $n_l(x) \in \text{Im}(\nabla J(x)^T)$ implies $n_l(x)^T \Pi(x) = 0$ for all $l \in [r]$.

Next, we define a vector field $F(x)$ by $F(x) = \Pi(x)\text{div}_\Sigma(\Pi(x))$ where $(\text{div}_\Sigma\Pi(x))_k := \text{div}_\Sigma(f_k(x))$ for the vector field $f_k(x) = \Pi(x)^T e_k = \Pi(x)e_k$. With this definition, we have $\text{div}_\Sigma\Pi = -\sum_{l=1}^r \text{div}_\Sigma(n_l n_l^T)$ and observe that for any component index $k \in [d]$,

$$(\text{div}_\Sigma(n_l n_l^T))_k = \text{Tr}\left(\Pi\nabla(n_l n_l^T e_k)\right) = \sum_{i,j=1}^d \Pi_{ij}\partial_i(n_{lj}n_{lk}) = \sum_{i,j=1}^d \left[\Pi_{ij}(\partial_i n_{lj})n_{lk} + \Pi_{ij}n_{lj}(\partial_i n_{lk})\right]$$

$$= (\text{div}_\Sigma n_l)n_{lk} + \sum_{i=1}^d \underbrace{(n_l^T\Pi)_i}_{=0} \partial_i n_{lk} = (\text{div}_\Sigma n_l)n_{lk},$$

where we used the property that $n_l$ is orthogonal to the tangent space ($n_l^T\Pi = 0$). From this fact, we have the following result:

$$\text{div}_\Sigma\Pi = -\sum_{l=1}^r \text{div}_\Sigma(n_l n_l^T) = -\sum_{l=1}^r (\text{div}_\Sigma n_l)n_l \;\Rightarrow\; F = \Pi\text{div}_\Sigma(\Pi) = -\sum_{l=1}^r (\text{div}_\Sigma n_l)\underbrace{\Pi n_l}_{=0} = 0.$$

Finally, the definition of $F$ gives $\sum_{k=1}^d (\text{div}_\Sigma f_k) f_k = F$, which is zero by the argument above. $\square$

**Theorem B.2** (Stationarity of OLLA). *Assume $\sigma(t)$ is constant for $\forall t \geq 0$ and $X_0 \in \Sigma$. Then, OLLA (3) has the following stationary distribution $\rho_\Sigma$ with respect to measure $d\sigma_\Sigma$:*

$$\rho_\Sigma(x) = \frac{1}{Z_\Sigma} e^{-f(x)}, \quad x \in \Sigma$$

*where $d\sigma_\Sigma$ is the surface (or Hausdorff) measure on $\Sigma$ and $Z_\Sigma := \int_\Sigma e^{-f(x)} d\sigma_\Sigma$ is the normalization constant.*

*Proof.* To prove stationarity, we observe that

$$\int_\Sigma \mathcal{L}\phi\rho_t d\sigma_\Sigma = \frac{d}{dt}\int_\Sigma \phi\rho_t d\sigma_\Sigma = \int_\Sigma \phi\partial_t\rho_t d\sigma_\Sigma = -\int_\Sigma \phi\text{div}_\Sigma(J_t)d\sigma_\Sigma = \int_\Sigma \langle J_t, \nabla_\Sigma\phi\rangle d\sigma_\Sigma$$

where the last equality comes from the boundary condition in Lemma B.1. Since $J_t$ is given by

$$J_t = -\frac{\sigma^2}{2}\left(\rho_t\nabla_\Sigma f + \nabla_\Sigma\rho_t\right) = -\frac{\sigma^2}{2}e^{-f}\nabla_\Sigma(\rho e^f),$$

on the interior of $\Sigma$, we conclude that

$$\int_\Sigma \mathcal{L}\phi\rho_\Sigma d\sigma_\Sigma = \int_\Sigma \langle J_t, \nabla_\Sigma\phi\rangle d\sigma_\Sigma = -\frac{\sigma^2}{2}\int_\Sigma \langle e^{-f}0, \nabla_\Sigma\phi\rangle d\sigma_\Sigma = 0,$$

where $J_t = 0$ due to the fact that $\rho_\Sigma \propto e^{-f}$. This proves that $\rho_\Sigma$ is the stationary distribution of the OLLA. □

### B.2. Construction of ULLA

**Notations and Background for underdamped setup.** In the constrained underdamped Langevin case, assuming $X_0 \in \Sigma$, the natural space of $(X_t, P_t)$ is the cotangent bundle $T^*\Sigma := \{(x,p) \in \mathbb{R}^d \times \mathbb{R}^d \mid x \in \Sigma, \langle \nabla J(x), p\rangle = 0\}$, where $\Sigma := \{x \in \mathbb{R}^d \mid h(x) = 0, g(x) \le 0\}$ is a stratified manifold. Because there is no boundary for the cotangent space $T_x^*\Sigma := \{p \in \mathbb{R}^d \mid \langle \nabla J(x), p\rangle = 0\}$, the boundary of $T^*\Sigma$ is given as $\partial T^*\Sigma = \partial\Sigma \times T_x^*\Sigma$. In this cotangent bundle $T^*\Sigma$, the canonical reference measure is the Liouville (symplectic) measure $\sigma_{T^*\Sigma}$ defined as $d\sigma_{T^*\Sigma}(x,p) := d\sigma_\Sigma(x) \otimes dp(x)$ with $d\sigma_\Sigma$ the surface measure on $\Sigma$ and $dp(x)$ the Lebesgue measure on the cotangent space $T_x^*\Sigma$ induced by the inner product $\langle u, v\rangle = u^T v$.

For the notations, we write $\nabla_\Sigma$ and $\mathsf{div}_\Sigma$ for the intrinsic gradient and divergence in $x$, and $\nabla_{T_x^*\Sigma}, \mathsf{div}_{T_x^*\Sigma}$ for the intrinsic gradient and divergence in $p$. Under these notations, for a smooth function $\phi$ and smooth vector field $X$ on $\Sigma$, we have

$$\nabla_\Sigma\phi = \Pi\nabla_x\phi, \qquad \mathsf{div}_\Sigma(X) = \mathsf{Tr}\left(\Pi\nabla_x X\right)$$

Similarly, for a smooth function $\psi$ and smooth vector $Y$ on $T_x^*\Sigma$, we have

$$\nabla_{T_x^*\Sigma}\psi = \Pi\nabla_p\psi, \qquad \mathsf{div}_{T_x^*\Sigma}(Y) = \mathsf{Tr}\left(\Pi\nabla_p Y\right),$$

where $\nabla_x$ and $\nabla_p$ represent ambient Euclidean partial gradient or Jacobian operators with respect to $x$ or $p$. Also, the global gradient with respect to $\sigma_{T^*\Sigma}$ is given by $\nabla_{T^*\Sigma}\phi = [\nabla_\Sigma\phi, \nabla_{T_x^*\Sigma}\phi]^T$ for any smooth function $\phi$ on $T^*\Sigma$ and the global divergence with respect to $\sigma_{T^*\Sigma}$ can be represented by $\mathsf{div}_{T^*\Sigma}([V^x, V^p]^T) = \mathsf{div}_\Sigma(V^x) + \mathsf{div}_{T_x^*\Sigma}(V^p)$ for any smooth tangent field $[V^x, V^p]^T \in T^*\Sigma$. For comprehensive backgrounds on constrained underdamped Langevin dynamics, see Chapter 3.3 in Lelièvre et al. (2010).

---

**Proposition B.2** (Construction of ULLA)**.** *Consider the following Lagrangian-form constrained underdamped Langevin dynamics:*

$$\begin{cases} dX_t = \sigma(t)^2 P_t dt + \nabla J(X_t)^T d\lambda_t \\ dP_t = -\sigma(t)^2 \nabla f(X_t)dt - \sigma(t)^2\gamma P_t dt + \sigma(t)\sqrt{2\gamma}\circ dW_t + \nabla J(X_t)^T d\mu_t, \end{cases} \quad (4)$$

*where $d\lambda_t, d\mu_t$ are the adapted processes such that $dJ(X_t) = -\alpha\sigma(t)^2 J(X_t)$ (position constraint) and $\nabla J(X_t)P_t = 0$ (momentum tangency constraint), respectively.*

*Assuming $\nabla J(X_0)P_0 = 0$, the explicit minimum norm solution of $d\lambda_t, d\mu_t$ are given by*

$$\begin{cases} d\lambda_t = -\alpha\sigma(t)^2 G^\dagger(X_t)J(X_t)dt \\ d\mu_t = G^\dagger(X_t)\nabla J(X_t)\left[\sigma(t)^2\nabla f(X_t) + \sigma(t)^2\gamma P_t dt + \sigma(t)\sqrt{2\gamma}\circ dW_t\right] + \\ \qquad G^\dagger(X_t)\left[-\sigma(t)^2\mathcal{H}_1(X_t, P_t) + \alpha\sigma(t)^2\mathcal{H}_2(X_t, P_t)\right]dt, \end{cases}$$

*where $G(x) := \nabla J(x)\nabla J(x)^T$ is the Gram matrix and $\Pi(x) = I - \nabla J(x)^T G^\dagger(x)\nabla J(x)$ is the tangential projection map.*

*Therefore, the closed form SDE of (4) is given as follows:*

$$\begin{cases} dX_t = \sigma(t)^2 P_t dt - \alpha\sigma(t)^2\nabla J(X_t)^T G^\dagger(X_t)J(X_t)dt \\ dP_t = \Pi(X_t)\left[-\sigma(t)^2\nabla f(X_t) - \sigma(t)^2\gamma P_t dt + \sigma(t)\sqrt{2\gamma}\circ dW_t\right] \\ \qquad - \sigma(t)^2\nabla J(X_t)^T G^\dagger(X_t)\left[\mathcal{H}_1(X_t, P_t) - \alpha\mathcal{H}_2(X_t, P_t)\right]dt \end{cases}$$

---

where $\mathcal{H}_1, \mathcal{H}_2 \in \mathbb{R}^{m+|I_x|}$ are the curvature correction terms defined as

$$[\mathcal{H}_1(x,p)]_i := p^T \nabla^2 J_i(x)p,$$
$$[\mathcal{H}_2(x,p)]_i := p^T \nabla^2 J_i(x)(\nabla J(x)^T G^\dagger(x)J(x))$$

with $[\mathcal{H}_1(x,p)]_i$ and $[\mathcal{H}_2(x,p)]_i$ denoting the i-th entries of $\mathcal{H}_1(x,p)$ and $\mathcal{H}_2(x,p)$, respectively.

*Proof.* From the Stratonovich chain rule, we observe that

$$-\alpha\sigma(t)^2 J(X_t)dt = dJ(X_t) = \nabla J(X_t) \circ dX_t = \sigma(t)^2 \underbrace{\nabla J(X_t)P_t}_{=0} dt + G(X_t)d\lambda_t.$$

Because the initial condition gives $\nabla J(X_0)P_0 = 0$ and $d\mu_t$ imposes $d(\nabla J(X_t)P_t) = 0$, the first term becomes zero and the minimum norm solution of $d\lambda_t$ simplifies to

$$d\lambda_t = -\alpha\sigma(t)^2 G^\dagger(X_t)J(X_t)dt.$$

To find the explicit minimum norm solution for the process $d\mu_t$, we consider the momentum tangency constraint $\nabla J(X_t)^T P_t = 0$. Using the Stratonovich chain rule again, we have

$$0 = d(\nabla J(X_t)P_t) = P_t^T \nabla^2 J(X_t)dX_t + \nabla J(X_t)dP_t.$$

By substituting $dX_t$ and $dP_t$, the previous equation simplifies to

$$\begin{aligned} 0 =& P_t^T \nabla^2 J(X_t) \left[\sigma(t)^2 P_t dt + \nabla J(X_t)^T d\lambda_t\right] \\ &+ \nabla J(X_t)\left[-\sigma(t)^2 \nabla f(X_t) - \sigma(t)^2 \gamma P_t dt + \sigma(t)\sqrt{2\gamma} \circ dW_t + \nabla J(X_t)^T d\mu_t\right] \\ =& \left[\sigma^2 \mathcal{H}_1 - \alpha\sigma^2 \mathcal{H}_2 - \sigma^2 \nabla J \nabla f - \sigma^2 \gamma \nabla J P_t\right]dt + \sigma\nabla J\sqrt{2\gamma} \circ dW_t + Gd\mu_t. \end{aligned}$$

This gives the following minimum norm solution of $d\mu_t$:

$$d\mu_t = G^\dagger \left[\sigma^2 \nabla J \nabla f + \sigma^2 \gamma \nabla J P_t - \sigma^2 \mathcal{H}_1 + \alpha\sigma^2 \mathcal{H}_2\right]dt - \sigma G^\dagger \nabla J\sqrt{2\gamma} \circ dW_t.$$

Therefore, we recover the following $dX_t, dP_t$ by plugging the adapted process $d\lambda_t, d\mu_t$ into the previous equations:

$$\begin{cases} dX_t =& \sigma(t)^2 P_t dt - \alpha\sigma(t)^2 \nabla J(X_t)^T G^\dagger(X_t)J(X_t)dt \\ dP_t =& \Pi(X_t)\left[-\sigma(t)^2 \nabla f(X_t)dt - \sigma(t)^2 \gamma P_t dt + \sigma(t)\sqrt{2\gamma} \circ dW_t\right] \\ & - \sigma(t)^2 \nabla J(X_t)^T G^\dagger(X_t)\left[\mathcal{H}_1(X_t, P_t) - \alpha\mathcal{H}_2(X_t, P_t)\right]dt. \end{cases}$$

We note that this is the unique closed form SDE because $\nabla J(X_t)^T d\lambda_t$ and $\nabla J(X_t)^T d\mu_t$ are unique among many solutions $d\lambda_t, d\mu_t$ satisfying the properties.

Finally, we observe that the Itô-Stratonovich correction term $\frac{1}{2}(\nabla B)B = 0$ where $B = [0, \sigma\sqrt{2\gamma}\Pi]^T \in \mathbb{R}^{2d \times d}$ is the diffusion matrix. This is because the position entries of $B$ are zero and the momentum entries of $B$ depend only on position. Therefore, we have the same formula on the Itô version of the above Stratonovich SDE. □

**Lemma B.3** (Boundary condition of ULLA). *Assuming $X_0 \in \Sigma$ and $\nabla J(X_0)P_0 = 0$, ULLA (4) satisfies the following boundary condition on $\partial\Sigma$ and property for $t \geq 0$:*

$$(1)\ \langle J_t(x,p), n(x)\rangle = 0 \quad \text{a.e on } \partial T^*\Sigma, \qquad (2)\ (X_t, P_t) \in T^*\Sigma,$$

*where $J_t(x)$ is the probability current density defined by $\partial_t \rho_t = -\mathsf{div}_\Sigma(J_t)$ and $n(x)$ is the outward unit normal vector on $\partial T^*\Sigma$.*

*Proof.* First, we prove $\mathbb{P}(g_k(X_t) > 0) = 0$ for $t \geq 0, k \in [l]$. In particular, this implies $(X_t, P_t) \in T^*\Sigma$ a.s. for all $t \geq 0$. To show this, we observe that Lemma B.4 gives

$$dg_k(X_t) = -\alpha\sigma(t)^2(g_k(X_t) + \epsilon)dt$$

for $\{g_k \geq 0\}$. Thus, the same proof introduced in Lemma B.1 gives $\mathbb{P}(g(X_t) \leq 0) = 1$ for $t \geq 0$ and, therefore, we have $(X_t, P_t) \in T^*\Sigma$ a.s for $t \geq 0$.

Next, we show the boundary condition of this SDE. To demonstrate this, we first formulate the SDE of ULLA in the following form:

$$dZ_t = V_0 dt + \sum_{k=1}^{2d} V_k \circ dB_t^k$$

with $Z_t := [X_t, P_t]^T \in \mathbb{R}^{2d}$, $V_0(x,p) := \left[\sigma^2 p - \alpha\sigma^2 \nabla J^T G^\dagger J, -\sigma^2 \Pi \nabla f - \sigma^2 \gamma \Pi p + \kappa\right]^T \in \mathbb{R}^{2d}$, and $V_k(x,p) := [0, Be_k] \in \mathbb{R}^{2d}$, where $\kappa := -\sigma^2 \nabla J^T G^\dagger \left[\mathcal{H}_1 - \alpha\mathcal{H}_2\right]$ is the curvature correction related term and $B := \sigma\Pi\sqrt{2\gamma}$. Now, we use the Fokker-Planck equation on the interior of $T^*\Sigma$ (Theorem B.1) applied with $\mathsf{div}_{T^*\Sigma}$, and this gives the following equation:

$$\partial_t \rho_t = -\mathsf{div}_{T^*\Sigma}(\rho_t V_0) + \frac{1}{2}\sum_{k=1}^{2d}\mathsf{div}_{T^*\Sigma}\left(\mathsf{div}_{T^*\Sigma}(\rho_t V_k)V_k\right),$$

where

$$-\mathsf{div}_{T^*\Sigma}(\rho_t V_0) = -\mathsf{div}_\Sigma(\rho_t\left(\sigma^2 p - \alpha\sigma^2 \nabla J^T G^\dagger J\right)) - \mathsf{div}_{T_x^*\Sigma}\left(\rho_t\left(-\sigma^2 \Pi \nabla f - \sigma^2 \gamma \Pi p + \kappa\right)\right)$$

and

$$\frac{1}{2}\sum_{k=1}^{2d}\mathsf{div}_{T^*\Sigma}\left(\mathsf{div}_{T^*\Sigma}(\rho_t V_k)V_k\right) = \frac{1}{2}\sum_{k=1}^{d}\mathsf{div}_{T_x^*\Sigma}\left(\langle Be_k, \nabla_{T_x^*\Sigma}\rho_t\rangle Be_k\right) = \frac{1}{2}\mathsf{div}_{T_x^*\Sigma}\left((BB^T)\nabla_{T_x^*\Sigma}\rho_t\right).$$

Therefore, we recover the following equation:

$$\partial_t \rho_t = -\mathsf{div}_\Sigma(\underbrace{\rho_t\left(\sigma^2 p - \alpha\sigma^2 \nabla J^T G^\dagger J\right)}_{:=J_t^x(x,p)})$$

$$- \mathsf{div}_{T_x^*\Sigma}\left(\underbrace{\rho_t\left(-\sigma^2 \Pi \nabla f - \sigma^2 \gamma \Pi p + \kappa - \sigma^2 \gamma \Pi \Pi^T \nabla_{T_x^*\Sigma}\rho_t\right)}_{:=J_t^p(x,p)}\right)$$

$$= -\mathsf{div}_{T^*\Sigma}\left([J_t^x, J_t^p]^T\right) = -\mathsf{div}_{T^*\Sigma}(J_t)$$

Finally, we observe that, on the boundary $\partial T^*\Sigma = \partial\Sigma \times T_x^*\Sigma$, the outward normal is given by $n = [n_x, 0]^T \in \mathbb{R}^{2d}$ with $n_x$ being the outward unit normal vector on $\partial\Sigma$. Therefore, we have

$$0 = \frac{d}{dt}\int_{T^*\Sigma}\rho_t(x,p)d\sigma_{T^*\Sigma} = -\int_{T^*\Sigma}\mathsf{div}_{T^*\Sigma}(J_t(x,p))d\sigma_{T^*\Sigma} = -\int_{\partial T^*\Sigma}\langle J_t(x,p), n\rangle d\sigma_{\partial T^*\Sigma}$$

$$= -\int_{\partial T^*\Sigma}\langle J_t^x(x,p), n_x\rangle d\sigma_{\partial T^*\Sigma} = -\int_{\partial T^*\Sigma}\langle \rho_t\left(\sigma^2 p - \alpha\sigma^2 \nabla J^T G^\dagger J\right), n_x\rangle d\sigma_{\partial T^*\Sigma}$$

$$= \alpha\sigma^2\int_{\partial T^*\Sigma}\rho_t\underbrace{\langle\nabla J^T G^\dagger J, n_x\rangle}_{>0}d\sigma_{\partial T^*\Sigma}$$

and it implies $\rho_t(x,p) = 0$ a.e. on $\partial T^*\Sigma$. Lastly, we conclude the proof by observing that the following holds a.e on $\partial T^*\Sigma$:

$$\langle J_t, n\rangle = \langle J_t^x, n_x\rangle = \langle\rho_t\left(\sigma^2 p - \alpha\sigma^2 \nabla J^T G^\dagger J\right), n_x\rangle = -\alpha\sigma^2 \rho_t\langle\nabla J^T G^\dagger J, n_x\rangle = 0.$$

$\square$

**Theorem B.3** (Stationarity of ULLA). *Assume $\sigma(t)$ is constant for $\forall t \geq 0$, $X_0 \in \Sigma$, and the tangency constraint $\nabla J(X_0)P_0 = 0$ holds. Then, the ULLA (4) has the following stationary distribution $\rho_{T^*\Sigma}$ with respect to the measure $d\sigma_{T^*\Sigma}$:*

$$\rho_{T^*\Sigma}(x, p) = \frac{1}{Z_{T^*\Sigma}} \exp\left(-f(x) - \frac{1}{2}\|p\|_2^2\right), \quad (x, p) \in T^*\Sigma$$

*where $d\sigma_{T^*\Sigma}$ is the Liouville measure on $T^*\Sigma$ and $Z_{T^*\Sigma} := \int_{T^*\Sigma} e^{\left(-f(x) - \frac{1}{2}\|p\|_2^2\right)} d\sigma_{T^*\Sigma}$ is the normalization constant.*

*Proof.* First, from Lemma B.3, we know that the SDE of ULLA can be rewritten as follows on the interior of $T^*\Sigma$ :

$$dZ_t = V_0 dt + \sum_{k=1}^{2d} V_k \circ dB_t^k$$

with $Z_t := [X_t, P_t]^T \in \mathbb{R}^{2d}$, $V_0(x, p) := \left[\sigma^2 p, -\sigma^2 \Pi \nabla f - \sigma^2 \gamma \Pi p + \kappa\right]^T \in \mathbb{R}^{2d}$, and $V_k(x, p) := [0, Be_k] \in \mathbb{R}^{2d}$, where $\kappa := -\sigma^2 \nabla J^T G^\dagger \mathcal{H}_1$ is the curvature correction related term and $B := \sigma\Pi\sqrt{2\gamma}$. Therefore, Theorem B.1 gives the following generator $\mathcal{L}$ for any smooth function $\phi$ on $T^*\Sigma$:

$$\mathcal{L}\phi = V_0\phi + \frac{1}{2}\sum_{k=1}^{2d} V_k(V_k\phi).$$

Because we have

$$V_0\phi = \langle \nabla_\Sigma \phi, \sigma^2 p \rangle + \langle \nabla_{T^*_x\Sigma}\phi, -\sigma^2\Pi\nabla f - \sigma^2\gamma\Pi p + \kappa \rangle$$

and

$$\frac{1}{2}\sum_{k=1}^{2d} V_k(V_k\phi) = \frac{1}{2}\sum_{k=1}^{d} \langle Be_k, \nabla_p\left(\langle Be_k, \nabla_p\phi \rangle\right)\rangle = \frac{1}{2}\sum_{k=1}^{d} \langle \nabla_p^2\phi Be_k, Be_k \rangle = \sigma^2\gamma\mathsf{Tr}\left(\Pi\nabla_p^2\phi\right)$$

on the interior of $T^*\Sigma$, we can simplify $\mathcal{L}\phi$ as follows:

$$\mathcal{L}\phi = \sigma^2\left[\underbrace{\langle\nabla_\Sigma\phi, p\rangle - \langle\Pi\nabla f, \nabla_{T^*_x\Sigma}\phi\rangle + \langle\kappa, \nabla_{T^*_x\Sigma}\phi\rangle}_{\mathcal{L}_H\phi} + \underbrace{\gamma\left(\Delta_{T^*_x\Sigma}\phi - \langle\nabla_{T^*_x\Sigma}\phi, p\rangle\right)}_{\mathcal{L}_{OU}\phi}\right]$$

where we used $\Pi p = p$ (tangency constraint) and $\Delta_{T^*_x\Sigma}\phi = \mathsf{div}_{T^*_x\Sigma}(\Pi\nabla_p\phi) = \mathsf{Tr}\left(\Pi\nabla_p^2\phi\right)$. Next, we note the following identity:

$$\mathsf{div}_{T^*\Sigma}\left(e^{-\|p\|^2/2}\nabla_{T^*\Sigma}\phi\right) = e^{-\|p\|^2/2}\left(\Delta_{T^*\Sigma}\phi - \langle\nabla_{T^*\Sigma}\phi, p\rangle\right).$$

Under this identity, we observe that

$$\int_{T^*\Sigma} \mathcal{L}_{OU}\phi \rho_{T^*\Sigma} d\sigma_{T^*\Sigma} = \gamma \int_{T^*\Sigma} \left(\Delta_{T^*_x\Sigma}\phi - \langle\nabla_{T^*_x\Sigma}\phi, p\rangle\right) e^{-H} d\sigma_{T^*\Sigma} = 0$$

where $H := f(x) + \frac{1}{2}\|p\|^2$ and the last equality holds because $T^*_x\Sigma$ does not have boundary and $d\sigma_{T^*\Sigma}(x, p) := d\sigma_\Sigma(x) \otimes dp(x)$ holds. Lastly, we define $X_H := \left[p - \alpha\nabla J^T G^\dagger J, -\Pi\nabla f + \kappa\right]^T \in \mathbb{R}^{2d}$ so that $X_H = [p, -\Pi\nabla f + \kappa]^T$ and $\mathcal{L}_H\phi = \langle X_H, \nabla_{T^*\Sigma}\phi\rangle$ at the interior of $T^*\Sigma$. Then, we have

$$\langle X_H, \nabla_{T^*\Sigma}H\rangle = \langle p, \nabla_\Sigma f\rangle + (-\Pi\nabla f + \kappa)\Pi p = \underbrace{\langle(\nabla_\Sigma f - \Pi\nabla f), \Pi p\rangle}_{=0} + \underbrace{\langle\kappa, \Pi p\rangle}_{=0} = 0,$$

where the last equality holds due to tangency constraint of $p$. In addition to this, the following identity holds on the interior of $T^*\Sigma$ due to Liouville's theorem, which is the preservation property of Liouville measure on $T^*\Sigma$ under the constrained Hamiltonian field (see Chapter 1.2.2 and Proposition 3.46 in Lelièvre et al. (2010)):

$$\mathsf{div}_{T^*\Sigma}X_H = \mathsf{div}_\Sigma(p) + \mathsf{div}_{T^*_x\Sigma}\left(-\Pi\nabla f + \kappa\right) = 0.$$

Hence, using these properties and $\rho_{T^*\Sigma} \propto e^{-H}$, we have

$$\int_{T^*\Sigma} \mathcal{L}_{OU}\phi\rho_{T^*\Sigma}d\sigma_{T^*\Sigma} = \int_{T^*\Sigma}\langle X_H, \nabla_{T^*\Sigma}\phi\rangle\rho_{T^*\Sigma}d\sigma_{T^*\Sigma} \overset{(1)}{=} -\int \phi \text{div}_{T^*\Sigma}\left(\rho_{T^*\Sigma}X_H\right)d\sigma_{T^*\Sigma}$$

$$= -\int_{T^*\Sigma} \phi\left(\langle X_H, \nabla_{T^*\Sigma}\rho_{T^*\Sigma}\rangle + \underbrace{\text{div}_{T^*\Sigma}(X_H)}_{=0}\rho_{T^*\Sigma}\right)d\sigma_{T^*\Sigma}$$

$$\overset{(2)}{=} \int_{T^*\Sigma} \phi\rho_{T^*\Sigma}\langle X_H, \nabla_{T^*\Sigma}H\rangle d\sigma_{T^*\Sigma} = 0,$$

where (1) comes from the boundary condition in Lemma B.3 and (2) comes from the fact that $-\nabla_{T^*\Sigma}H = \nabla_{T^*\Sigma}\ln\rho_{T^*\Sigma} = \nabla_{T^*\Sigma}\rho_{T^*\Sigma}/\rho_{T^*\Sigma}$.

By combining the observations above, we have

$$\int_{T^*\Sigma} \mathcal{L}\phi\rho_{T^*\Sigma}d\sigma_{T^*\Sigma} = \int_{T^*\Sigma} \mathcal{L}_{OU}\phi\rho_{T^*\Sigma}d\sigma_{T^*\Sigma} + \int_{T^*\Sigma} \mathcal{L}_H\phi\rho_{T^*\Sigma}d\sigma_{T^*\Sigma} = 0,$$

which proves the theorem. $\qquad\square$

### B.3. Properties and Backward Processes of Constrained Langevin Dynamics with Landing

**Exponential decaying properties of constrained Langevin dynamics with landing**     Due to our previous construction, the landing term $-\alpha\nabla J(X_t)^T G^\dagger(X_t)J(X_t)$ appears on both constrained overdamped or underdamped Langevin dynamics so that the processes always satisfy $dJ(X_t) = -\alpha\sigma(t)^2 J(X_t)$ deterministically $\forall t \geq 0$. Therefore, even if $X_t \notin \Sigma$, the process can approach $\Sigma$ exponentially fast as illustrated in the following Lemma B.4.

**Lemma B.4** (Exponential decay of constraint functions). *Let $J(x)$ be the constraint function vector defined as*

$$J(x) = \left[h_1(x), ..., h_m(x), g_{i_1}(x) + \epsilon, ..., g_{i_{|I_x|}}(x) + \epsilon\right]^T \in \mathbb{R}^{m+|I_x|} \tag{5}$$

*where $h : \mathbb{R}^d \to \mathbb{R}^m, g : \mathbb{R}^d \to \mathbb{R}^l$ are equality and inequality constraint functions respectively, and $I_x := \{i \in [l] : g(x) \geq 0\}$ is the active index set of inequality constraints.*

*Under this setup, the constrained Langevin dynamics (3 or 4) satisfy the following constraint satisfaction property almost surely:*

$$h_i(X_t) = h_i(X_0)e^{-\alpha S(t)}, \quad t \geq 0$$

*and*

$$\begin{cases} g_j(X_t) = -\epsilon + (g_j(X_0) + \epsilon)e^{-\alpha S(t)}, & t \leq \tau_{j,\epsilon} \\ g_j(X_t) \leq 0, & t \geq \tau_{j,\epsilon}, \end{cases}$$

*where $S(t) := \int_0^t \sigma(s)^2 ds$ and $\tau_{j,\epsilon}$ is defined to be*

$$\tau_{j,\epsilon} := \inf\left\{t \geq 0 \mid S(t) \geq \frac{1}{\alpha}\ln\left(\frac{g_j(X_0) + \epsilon}{\epsilon}\right)\right\}, \quad \forall j \in I_{x_0}.$$

*Proof.* From the Stratonovich chain rule, it holds almost surely that

$$dJ(X_t) = \nabla J(X_t) \circ dX_t = -\alpha\sigma(t)^2 J(X_t)dt.$$

For each equality constraint $h_i$, the component is active for $\forall t \geq 0$. Therefore, we have:

$$dh_i(X_t) = -\alpha\sigma(t)^2 h_i(X_t)dt$$

and solving this ODE yields:

$$h_i(X_t) = h_i(X_0)e^{-\alpha S(t)}, \quad t \geq 0.$$

For the inequality constraints, we fix $j \in I_{X_0} := \{j \in [l] \mid g(X_0) \geq 0\}$. While the $j$-th inequality is active, we have $J_{m+j} = g_j + \epsilon$ in the constraint vector, and the same chain rule gives

$$d(g_j(X_t) + \epsilon) = -\alpha\sigma(t)^2 (g_j(X_t) + \epsilon) \, dt.$$

Hence, before time $t \leq \tau_{j,\epsilon} := \inf \left\{ t \geq 0 \mid S(t) \geq \frac{1}{\alpha} \ln \left( \frac{g_j(X_0) + \epsilon}{\epsilon} \right) \right\}$, we have

$$g_j(X_t) = -\epsilon + (g_j(X_0) + \epsilon) \, e^{-\alpha s(t)}.$$

After $t \geq \tau_{j,\epsilon}$, the particle $X_t$ is instantaneously repelled into the interior of $\Sigma$ whenever it hits the boundary $\partial\Sigma$. Therefore, $g_j(X_t) \leq 0$ holds for $t \geq \tau_{j,\epsilon}$. $\qquad\square$

**Backward process on manifold**     Now, we discuss how to construct the backward process of the proposed constrained Langevin dynamics. Define $\mathcal{M} \subset \mathbb{R}^D$ to be a smooth, compact, embedded Riemannian manifold endowed with an induced metric, and we restrict the choice of $\mathcal{M}$ to be either $\mathcal{M} = \Sigma$ $(D = d)$ or $\mathcal{M} = T^*\Sigma$ $(D = 2d)$. For $x \in \mathcal{M}$, let $\Pi$ be the tangential projection map on $\mathcal{M}$. We consider the stochastic process $X_t \sim q_t$ driven by the following Stratonovich SDE on $\mathcal{M}$ with $X_0 \sim q_0 = q_{\text{data}}$:

$$dX_t = \Pi(X_t)b(X_t, t)dt + \kappa(t)\Pi(X_t) \circ dW_t, \quad t \in [0, T] \tag{6}$$

where $b(y, t) : \mathcal{M} \times \mathbb{R} \to \mathbb{R}^D$, $\kappa(t) : \mathbb{R} \to \mathbb{R}^{D \times D}$ are the drift vector and diagonal diffusion matrix for the corresponding SDE. In practice, $\kappa(t) : \mathbb{R} \to \mathbb{R}^{D \times D}$ is chosen to satisfy $X_T \sim q_T \approx p_{\text{prior}}$ with $p_{\text{prior}}$ being the easy-to-sample prior distribution on $\mathcal{M}$ and $T$ the terminal time for sampling. Then, Lemma B.5 shows that the stochastic process $\overleftarrow{X}_t \sim p_t$ driven by the following Stratonovich SDE becomes the backward process of $X_t$:

$$d\overleftarrow{X}_t = \Pi(\overleftarrow{X}_t) \left[ -b(\overleftarrow{X}_t, T - t) + \kappa(T - t)^2 \nabla \ln q_{T-t}(\overleftarrow{X}_t) \right] dt + \kappa(T - t)\Pi(\overleftarrow{X}_t) \circ d\bar{W}_t, \quad t \in [0, T]. \tag{7}$$

It is assumed that the forward process (6) and the backward process (7) have the same boundary conditions, so that $q_t = p_{T-t}$ holds for $t \in [0, T]$.

> **Lemma B.5** (Backward process verification)**.** *Let $X_t, \overleftarrow{X}_t \in \mathcal{M}$ be the stochastic process driven by the forward process* (6) *and the backward process* (7)*, respectively. If $q_T = p_0$ and the boundary conditions of $q_t$ and $p_{T-t}$ appearing in the Fokker-Planck equation are the same, then the following relation holds:*
>
> $$q_t(x) = p_{T-t}(x), \quad x \in \mathcal{M}, \ t \in [0, T],$$
>
> *where $q_t, p_t$ are the probability densities of each $X_t$ and $\overleftarrow{X}_t$.*

*Proof.* Let $\nabla_{\mathcal{M}} := \Pi\nabla$ and $\mathsf{div}_{\mathcal{M}}$ be the intrinsic gradient and divergence on $\mathcal{M}$, and let $\Delta_{\mathcal{M}} := \mathsf{div}_{\mathcal{M}}(\nabla_{\mathcal{M}} \cdot)$ be the intrinsic Laplacian operator on $\mathcal{M}$. From Theorem B.1 with $V_0(y, t) = \Pi(y)b(y, t), V_k(y, t) = \kappa(t)\Pi(x)e_k$, and $e_k$ being $k$th standard basis of $\mathbb{R}^d$, the Fokker-Planck equation of (6) is given by

$$\partial_t q_t = -\mathsf{div}_{\mathcal{M}}(q_t V_0) + \frac{1}{2}\sum_{k=1}^{D} \mathsf{div}_{\mathcal{M}}(\mathsf{div}_{\mathcal{M}}(q_t V_k)V_k) = -\mathsf{div}_{\mathcal{M}}(q_t \Pi b) + \frac{1}{2}\kappa(t)^2 \Delta_{\mathcal{M}} q_t,$$

where we used the property $\sum_{k=1}^{D} \mathsf{div}_{\mathcal{M}}(\mathsf{div}_{\mathcal{M}}(q_t \Pi e_k)(\Pi e_k)) = \Delta_{\mathcal{M}} p_t$.

Now observe that the process $\overleftarrow{X}_t$ driven by (7) has the following drift and diffusion term:

$$\tilde{V}_0(y, t) = \Pi(y) \left[ -b(y, T - t) + \kappa(T - t)^2 \nabla_{\mathcal{M}} \ln q_{T-t}(x) \right], \quad \tilde{V}_k(y, t) = g(T - t)\Pi(y)e_k.$$

Therefore, Theorem B.1 again implies the following Fokker-Planck equation:

$$\partial_t p_{T-t} = -\partial_s p_s \mid_{s=T-t} = \mathsf{div}_{\mathcal{M}}(p_{T-t}\tilde{V}_0) - \frac{1}{2}\sum_{k=1}^{d}\mathsf{div}_{\mathcal{M}}(\mathsf{div}_{\mathcal{M}}(p_{T-t}\tilde{V}_k)\tilde{V}_k)$$

$$= -\mathsf{div}_{\mathcal{M}}(p_{T-t}\Pi b) + \kappa(t)^2\mathsf{div}_{\mathcal{M}}(q_{T-t}\nabla_{\mathcal{M}}\ln p_{T-t}) - \frac{1}{2}\kappa(t)^2\Delta_{\mathcal{M}}p_{T-t}$$

$$= -\mathsf{div}_{\mathcal{M}}(p_{T-t}\Pi b) + \frac{1}{2}\kappa(t)^2\Delta_{\mathcal{M}}p_{T-t}.$$

In the case of manifold with boundary, it is assumed that the boundary condition on the Fokker-Planck equation of $q_t$ and $p_{T-t}$ are the same and $q_T = p_0$. This implies that $\overleftarrow{X}_t$ achieves the desired property as stated in the theorem. $\square$

**Backward process of Constrained Langevin with landing** From Proposition B.1, we note that the forward process $X_t \sim q_t$ of OLLA is given as follows in the interior of $\Sigma$:

$$dX_t = -\frac{1}{2}\sigma(t)^2\Pi(X_t)\nabla f(X_t)dt + \sigma(t)\Pi(X_t)\circ dW_t.$$

Therefore, applying Lemma B.5 on the interior of $\Sigma$, the backward process of the OLLA is given as:

$$d\overleftarrow{X}_t = \frac{1}{2}\sigma(T-t)^2\Pi(\overleftarrow{X}_t)\left[\nabla f(\overleftarrow{X}_t) + 2\nabla\ln q_{T-t}(\overleftarrow{X}_t)\right]dt + \sigma(t)\Pi(\overleftarrow{X}_t)\circ d\bar{W}_t.$$

Now, we note that the similar construction and proof provided in Proposition B.1 and Lemma B.1 can demonstrate that adding the landing term $-\alpha\sigma^2\nabla J^T G^\dagger J$ to the previous SDE enforces it to have the same boundary condition ($\langle J_t, n\rangle = 0$ a.e on $\partial\Sigma$) imposed on the forward process. Therefore, the backward process $\overleftarrow{X}_t$ is given as:

$$d\overleftarrow{X}_t = \frac{1}{2}\sigma(T-t)^2\Pi(\overleftarrow{X}_t)\left[\nabla f(\overleftarrow{X}_t) + 2\nabla\ln q_{T-t}(\overleftarrow{X}_t)\right]dt - \alpha\sigma(T-t)^2\nabla J(\overleftarrow{X}_t)^T G^\dagger(\overleftarrow{X}_t)J(\overleftarrow{X}_t)dt$$

$$+ \frac{\sigma(t)^2}{2}\mathcal{H}(\overleftarrow{X}_t)dt + \sigma(t)\Pi(\overleftarrow{X}_t)\circ d\bar{W}_t, \qquad\text{(Stratonovich-sense)}$$

which is equal to the following Itô version SDE involving the Itô-Stratonovich correction term $\mathcal{H}$:

$$d\overleftarrow{X}_t = \frac{1}{2}\sigma(T-t)^2\Pi(\overleftarrow{X}_t)\left[\nabla f(\overleftarrow{X}_t) + 2\nabla\ln q_{T-t}(\overleftarrow{X}_t)\right]dt + \frac{1}{2}\sigma(T-t)^2\mathcal{H}(\overleftarrow{X}_t)dt$$

$$\underbrace{-\alpha\sigma(T-t)^2\nabla J(\overleftarrow{X}_t)^T G^\dagger(\overleftarrow{X}_t)J(\overleftarrow{X}_t)}_{\text{Landing term}}dt + \sigma(T-t)\Pi(\overleftarrow{X}_t)\circ d\bar{W}_t. \qquad\text{(Itô-sense)}$$

Similarly, from Proposition B.2, the forward process $[X_t, P_t]^T \sim \overleftarrow{P}_t$ of ULLA is given as below in the interior of $\Sigma$:

$$\begin{cases} dX_t = \sigma(t)^2 P_t dt \\ dP_t = \Pi(X_t)\left[\sigma(t)^2\left[-\nabla f(X_t) - \gamma P_t\right]dt + \sigma(t)\sqrt{2\gamma}\circ dW_t\right] \\ \qquad - \sigma(t)^2\nabla J(X_t)^T G^\dagger(X_t)\mathcal{H}_1(X_t, P_t)dt. \end{cases}$$

Therefore, Lemma B.5 again implies that the following stochastic process $[\overleftarrow{X}_t, \overleftarrow{P}_t]^T \sim p_t$ becomes the backward process of $X_t$ on the interior of $\Sigma$:

$$\begin{cases} d\overleftarrow{X}_t = -\sigma(T-t)^2\overleftarrow{P}_t dt \\ d\overleftarrow{P}_t = \Pi(\overleftarrow{X}_t)\left[\sigma(T-t)^2\left[\nabla f(\overleftarrow{X}_t)dt + \gamma\overleftarrow{P}_t dt + 2\gamma\nabla_p\ln q_{T-t}(\overleftarrow{X}_t, \overleftarrow{P}_t)\right]dt + \sigma(T-t)\sqrt{2\gamma}\circ d\bar{W}_t\right] \\ \qquad + \sigma(T-t)^2\nabla J(\overleftarrow{X}_t)^T G^\dagger(\overleftarrow{X}_t)\mathcal{H}_1(\overleftarrow{X}_t, \overleftarrow{P}_t)dt. \end{cases}$$

Also, the similar construction and proof in Proposition B.2 and Lemma B.3 show that adding both the landing term $-\alpha\sigma^2\nabla J^T G^\dagger J$ and the corresponding landing correction term $+\alpha\sigma^2\nabla J^T G^\dagger \mathcal{H}_2$ to the previous SDE imposes the same boundary condition as in the forward process. Hence, the backward process $[\overleftarrow{X}_t, \overleftarrow{P}_t]^T$ is provided as:

$$
\begin{cases}
d\overleftarrow{X}_t = -\sigma(T-t)^2\overleftarrow{P}_t dt \underbrace{-\alpha\sigma(T-t)^2\nabla J(\overleftarrow{X}_t)G^\dagger(\overleftarrow{X}_t)J(\overleftarrow{X}_t)dt}_{\text{Landing term}} \\[2mm]
d\overleftarrow{P}_t = \Pi(\overleftarrow{X}_t)\left[\sigma(T-t)^2\left[\nabla f(\overleftarrow{X}_t)dt + \gamma\overleftarrow{P}_t dt + 2\gamma\nabla_p\ln q_{T-t}(\overleftarrow{X}_t, \overleftarrow{P}_t)\right]dt + \sigma(T-t)\sqrt{2\gamma}\circ d\bar{W}_t\right] \\[2mm]
\qquad + \sigma(T-t)^2\nabla J(\overleftarrow{X}_t)^T G^\dagger(\overleftarrow{X}_t)\left[\mathcal{H}_1(\overleftarrow{X}_t, \overleftarrow{P}_t) + \underbrace{\alpha\mathcal{H}_2(\overleftarrow{X}_t, \overleftarrow{P}_t)}_{\text{Landing correction term}}\right]dt.
\end{cases}
$$

Because the Itô-Stratonovich correction term vanishes in the underdamped case Proposition B.2, the Itô version SDE can be recovered from the Stratonovich version SDE by chaining $\circ$ to $\cdot$ in the Brownian motion term $d\bar{W}_t$.

### B.4. Discretization of Constrained Langevin Dynamics

In the discretization setup, we use a uniform grid $t_k = k\Delta t$ for $k = 0,.., N$ with terminal time $T = N\Delta t$. The forward trajectory uses state notations $X_k := X_{t_k}$ (or $[X_k, P_k]^T := [X_{t_k}, P_{t_k}]^T$) and updates in ascending index as $X_k \leftarrow X_{k+1} + \dots$. The backward trajectory is written in descending index, using states $X'_k := \overleftarrow{X}_{T-t_k}$ (or $[X'_k, P'_k]^T := [\overleftarrow{X}_{T-t_k}, \overleftarrow{P}_{T-t_k}]^T$) and updating as $X'_k \leftarrow X'_{k+1} + \dots$ for $k \in \{N-1, \dots 0\}$. We define the noise schedule in continuous time and evaluate the schedule on the discrete grid. This yields

$$
\sigma(t) := \sigma_{\min} + \frac{t}{T}(\sigma_{\max} - \sigma_{\min}), \quad \sigma_k := \sigma(t_k) = \sigma_{\min} + \frac{k}{N}(\sigma_{\max} - \sigma_{\min}).
$$

We also denote $q_k, p_k$ to be the probability densities of $X_k$ (or $[X_k, P_k]^T$) and $X'_k$ (or $[X'_k, P'_k]^T$) so that $q_k = q_{t_k}$ and $p_k = p_{T-t_k}$.

**Discretization of OLLA**  For the discretization of OLLA, we use straightforward Euler-Maruyama (EM) discretization.

**Discretization of Forward OLLA**  Recall that the Itô version of the forward process SDE is provided as follows:

$$
dX_t = -\frac{\sigma(t)^2}{2}\Pi(X_t)\nabla f(X_t)dt - \alpha\sigma(t)^2\nabla J(X_t)^T G^\dagger(X_t)J(X_t)dt + \frac{\sigma(t)^2}{2}\mathcal{H}(X_t)dt + \sigma(t)\Pi(X_t)dW_t.
$$

Therefore, the EM discretization of the forward process SDE becomes:

$$
X_{k+1} = X_k - \frac{\sigma_k^2}{2}\Pi(X_k)\nabla f(X_k)\Delta t - \alpha\sigma_k^2\nabla J(X_k)^T G^\dagger(X_k)J(X_k)\Delta t + \frac{\sigma_k^2}{2}\mathcal{H}(X_k)\Delta t + \sigma_k\sqrt{\Delta t}\Pi(X_k)\zeta_k,
$$

$$
\text{(Forward-OLLA)}
$$

where $\zeta_k \sim \mathcal{N}(0, I)$ is the standard Gaussian noise.

**Discretization of Backward OLLA**  Similarly, we note that following the backward SDE

$$
d\overleftarrow{X}_t = \frac{1}{2}\sigma(T-t)^2\Pi(\overleftarrow{X}_t)\left[\nabla f(\overleftarrow{X}_t) + 2\nabla\ln q_{T-t}(\overleftarrow{X}_t)\right]\Delta t + \frac{1}{2}\sigma(T-t)^2\mathcal{H}(\overleftarrow{X}_t)dt
$$

$$
- \alpha\sigma(T-t)^2\nabla J(\overleftarrow{X}_t)^T G^\dagger(\overleftarrow{X}_t)J(\overleftarrow{X}_t)dt + \sigma(T-t)\Pi(\overleftarrow{X}_t)\circ d\bar{W}_t
$$

can be discretized as follows:

$$
\overleftarrow{X}_{t_{k+1}} = \overleftarrow{X}_{t_k} + \frac{1}{2}\sigma(T-t_k)^2\Pi(\overleftarrow{X}_{t_k})\left[\nabla f(\overleftarrow{X}_{t_k}) + 2\nabla\ln q_{T-t_k}(\overleftarrow{X}_{t_k})\right]\Delta t + \frac{1}{2}\sigma(T-t_k)^2\mathcal{H}(\overleftarrow{X}_{t_k})\Delta t
$$

$$
- \alpha\sigma(T-t_k)^2\nabla J(\overleftarrow{X}_{t_k})^T G^\dagger(\overleftarrow{X}_{t_k})J(\overleftarrow{X}_{t_k})\Delta t + \sigma(t_k)\sqrt{\Delta t}\Pi(\overleftarrow{X}_{t_k})\bar{\zeta}_k.
$$

In our notation, this is equivalent to

$$
\begin{aligned}
X'_k =& X'_{k+1} + \frac{1}{2}\sigma_{k+1}^2\Pi(X'_{k+1})\left[\nabla f(X'_{k+1}) + 2\nabla \ln q_{k+1}(X'_{k+1})\right]\Delta t + \frac{1}{2}\sigma_{k+1}^2\mathcal{H}(X'_{k+1})\Delta t \\
&- \alpha\sigma_{k+1}^2\nabla J(X'_{k+1})^T G^\dagger(X'_{k+1})J(X'_{k+1})\Delta t + \sigma_{k+1}\sqrt{\Delta t}\Pi(X'_{k+1})\zeta'_{k+1} \qquad \text{(Backward-OLLA)}
\end{aligned}
$$

by changing $k \leftarrow N - (k+1)$.

**Discretized algorithm of constrained overdamped via Lagrangian multiplier**    When we are available to use Newton's method, we replace the explicit normal drifts $\mathcal{H}, -\alpha\nabla J^T G^\dagger J$ terms by a position projection via Lagrangian multipliers $\lambda$ at each step so that $J(x) = 0$ is satisfied. Under this way, we recover the following discretization of constrained overdamped Langevin dynamics using Lagrangian multiplier:

$$
X_{k+1} = X_k - \frac{\sigma_k^2}{2}\Pi(X_k)\nabla f(X_k)\Delta t + \sigma_k\sqrt{\Delta t}\Pi(X_k)\zeta_k + \nabla J(X_k)^T\lambda_k \qquad \text{(Forward-OLLA-P)}
$$

with $\lambda_k$ such that $J(X_{k+1}) = 0$. Similarly, the backward process can be obtained in a similar way as follows:

$$
\begin{aligned}
X'_k =& X'_{k+1} + \frac{1}{2}\sigma_{k+1}^2\Pi(X'_{k+1})\left[\nabla f(X'_{k+1}) + 2\nabla \ln q_{k+1}(X'_{k+1})\right]\Delta t + \sigma_{k+1}\sqrt{\Delta t}\Pi(X'_{k+1})\zeta'_{k+1} \\
&+ \nabla J(X'_{k+1})^T\lambda_{k+1} \qquad \text{(Backward-OLLA-P)}
\end{aligned}
$$

with $\lambda_{k+1}$ such that $J(X'_k) = 0$.

**Discretization of ULLA**    For the discretization of ULLA, we use a 1st order $O\tilde{B}A$ splitting scheme which uses an approximated $B$ process to compute the curvature correction term at $P_k$ with $\mathcal{O}(\Delta t)$ error. When discretizing, we use a collapsing technique to remove the momentum update rule. Because this collapsing technique requires saving only the position variables $X_k$, it is beneficial for saving memory, especially when saving forward trajectories is necessary.

**Discretization of Forward ULLA**    From the previous subsection, we recall the following Itô version of the forward process SDE:

$$
\begin{cases}
dX_t =& \sigma(t)^2 P_t dt - \alpha\sigma(t)^2\nabla J(X_t)^T G^\dagger(X_t)J(X_t)dt \\
dP_t =& \Pi(X_t)\left[\sigma(t)^2\left[-\nabla f(X_t) - \gamma P_t\right]dt + \sigma(t)\sqrt{2\gamma}dW_t\right] \\
& - \sigma(t)^2\nabla J(X_t)^T G^\dagger(X_t)\left[\mathcal{H}_1(X_t, P_t) - \alpha\mathcal{H}_2(X_t, P_t)\right]dt.
\end{cases}
$$

Under $O\tilde{B}A$ scheme, this can be split into three parts $O$, $\tilde{B}$, $A$ as follows:

$$
O: \begin{cases} dX_t =& 0 \\ dP_t =& -\sigma(t)^2\Pi(X_t)\gamma P_t dt + \sigma(t)\Pi(X_t)\sqrt{2\gamma}dW_t \end{cases} +
$$

$$
\tilde{B}: \begin{cases} dX_t =& 0 \\ dP_t =& -\sigma(t)^2\left(\Pi(X_t)\nabla f(X_t)dt + \nabla J(X_t)^T G^\dagger(X_t)\left[\mathcal{H}_1(X_t, P_t) - \alpha\mathcal{H}_2(X_t, P_t)\right]dt\right) \end{cases} +
$$

$$
A: \begin{cases} dX_t =& \sigma(t)^2 P_t dt - \alpha\sigma(t)^2\nabla J(X_t)^T G^\dagger(X_t)J(X_t)dt \\ dP_t =& 0. \end{cases}
$$

First, we integrate $O$ step from $t = t_k$ to $t = t_{k+1}$ with $X_t$ and $\sigma(t)$ frozen at $t = t_k$. Then, it gives

$$
\begin{cases}
X_{k+1}^O =& X_k \\
P_{k+1}^O =& (1 - \Pi(X_k))P_k + a_k\Pi(X_k)P_k + \sqrt{1 - a_k^2}\Pi(X_k)\zeta_k = a_k\Pi(X_k)P_k + \sqrt{1 - a_k^2}\Pi(X_k)\zeta_k
\end{cases}
$$

where $a_k := e^{-\gamma\sigma_k^2\Delta t}$ and we used the assumption that $P_k \in T_{X_k}\Sigma$, while will be guaranteed on the collapsing technique. Next, we integrate $\tilde{B}$ step with same integration domain with $X_t, \sigma(t)$ frozen at $t = t_k$. Then, we have:

$$
\begin{cases}
X_{k+1}^{O\tilde{B}} =& X_{k+1}^O + 0 = X_k \\
P_{k+1}^{O\tilde{B}} =& P_{k+1}^O - \sigma_k^2\Pi(X_k)\nabla f(X_k)\Delta t - \sigma_k^2\nabla J(X_k)^T G^\dagger(X_k)\left[\mathcal{H}_1(X_k, P_k) - \alpha\mathcal{H}_2(X_k, P_k)\right]\Delta t.
\end{cases}
$$

Similarly, integrating $A$ step with frozen $X_t, \sigma(t)$ gives :

$$\begin{cases} X_{k+1} = X_{k+1}^{O\tilde{B}} + \sigma_k^2 P_k^{O\tilde{B}} \Delta t - \alpha \sigma_k^2 \nabla J(X_k)^T G(X_k)^\dagger J(X_k) \Delta t \\ P_{k+1} = P_{k+1}^{O\tilde{B}} + 0 = P_{k+1}^{O\tilde{B}}, \end{cases}$$

which is equivalent to

$$\begin{cases} P_{k+1} = \Pi(X_k) \left[ a_k P_k - \sigma_k^2 \nabla f(X_k) \Delta t + \sqrt{1 - a_k^2} \zeta_k \right] \\ \qquad - \sigma_k^2 \nabla J(X_k)^T G^\dagger(X_k) \left[ \mathcal{H}_1(X_k, P_k) - \alpha \mathcal{H}_2(X_k, P_k) \right] \Delta t \\ X_{k+1} = X_k + \sigma_k^2 P_{k+1} \Delta t - \alpha \sigma_k^2 \nabla J(X_k)^T G(X_k)^\dagger J(X_k) \Delta t. \end{cases}$$

We note that this can be collapsed into one update rule with respect to $X$ state as follows:

$$X_{k+1} = X_k + \sigma_k^2 \Delta t \Pi(X_k) \left[ a_k P_k - \sigma_k^2 \nabla f(X_k) \Delta t + \sqrt{1 - a_k^2} \zeta_k \right]$$
$$- \alpha \sigma_k^2 \nabla J(X_k)^T G(X_k)^\dagger J(X_k) \Delta t - \sigma_k^4 \Delta t^2 \nabla J(X_k)^T G(X_k)^\dagger \left[ \mathcal{H}_1(X_k, P_k) - \alpha \mathcal{H}_2(X_k, P_k) \right].$$

Also, we observe that $P_k$ can be recovered as $P_k = \Pi(X_{k-1}) \left( \frac{X_k - X_{k-1}}{\sigma_{k-1}^2 \Delta t} \right)$ from the previous recursion which again can be approximated by $P_k \approx \tilde{P}_k := \Pi(X_k) \left( \frac{X_k - X_{k-1}}{\sigma_{k-1}^2 \Delta t} \right)$ with error $\mathcal{O}(\Delta t)$, guaranteeing the 1st order numerical error and $P_k \in T_{X_k} \Sigma$ assumption on $O$ step. Therefore, the final update rule for the forward process of ULLA becomes:

$$X_{k+1} = X_k + \sigma_k^2 \Delta t \Pi(X_k) \left[ a_k \tilde{P}_k - \sigma_k^2 \nabla f(X_k) \Delta t + \sqrt{1 - a_k^2} \zeta_k \right] \qquad \text{(Forward-ULLA)}$$
$$- \alpha \sigma_k^2 \nabla J(X_k)^T G(X_k)^\dagger J(X_k) \Delta t - \sigma_k^4 \Delta t^2 \nabla J(X_k)^T G^\dagger(X_k) \left[ \mathcal{H}_1(X_k, \tilde{P}_k) - \alpha \mathcal{H}_2(X_k, \tilde{P}_k) \right]$$

with $\tilde{P}_k := \Pi(X_k) \left( \frac{X_k - X_{k-1}}{\sigma_{k-1}^2 \Delta t} \right)$ for $k \in \{1, ..., N-1\}$ and $\tilde{P}_0 = \Pi(X_0) \zeta \in T_{X_0}^\Sigma$ where $\zeta \sim \mathcal{N}(0, I)$.

**Discretization of Backward ULLA**    For the discretization of backward process SDE, we recall the following backward process SDE of ULLA:

$$\begin{cases} d\overleftarrow{X}_t = -\sigma(T-t)^2 \overleftarrow{P}_t dt - \alpha \sigma(T-t)^2 \nabla J(\overleftarrow{X}_t) G^\dagger(\overleftarrow{X}_t) J(\overleftarrow{X}_t) dt \\ d\overleftarrow{P}_t = \Pi(\overleftarrow{X}_t) \left[ \sigma(T-t)^2 \left[ \nabla f(\overleftarrow{X}_t) dt + \gamma \overleftarrow{P}_t dt + 2\gamma \nabla_p \ln q_{T-t}(\overleftarrow{X}_t, \overleftarrow{P}_t) \right] dt + \sigma(T-t)\sqrt{2\gamma} d\bar{W}_t \right] \\ \qquad + \sigma(T-t)^2 \nabla J(\overleftarrow{X}_t)^T G^\dagger(\overleftarrow{X}_t) \left[ \mathcal{H}_1(\overleftarrow{X}_t, \overleftarrow{P}_t) + \alpha \mathcal{H}_2(\overleftarrow{X}_t, \overleftarrow{P}_t) \right] dt. \end{cases}$$

For the backward process of the $O\tilde{B}A$ scheme, the $2\sigma(T-t)^2 \Pi(\overleftarrow{X}_t) \gamma \overleftarrow{P}_t$ is added to $\tilde{B}$ step to guarantee the stable non-exploding OU process at $O$ step. We remark that the similar trick to handle $O$ step was previously used in Dockhorn et al. (2022). Under this technique, each $O, \tilde{B}, A$ steps can be given as follows:

$$O: \begin{cases} d\overleftarrow{X}_t = 0 \\ d\overleftarrow{P}_t = -\sigma(T-t)^2 \Pi(\overleftarrow{X}_t) \gamma \overleftarrow{P}_t dt + \sigma(T-t) \Pi(\overleftarrow{X}_t) \sqrt{2\gamma} d\bar{W}_t \end{cases} \quad +$$

$$\tilde{B}: \begin{cases} d\overleftarrow{X}_t = 0 \\ d\overleftarrow{P}_t = \sigma(T-t)^2 \Pi(\overleftarrow{X}_t) \left( \nabla f(\overleftarrow{X}_t) + 2\gamma \nabla_p \ln q_{T-t}(\overleftarrow{X}_t, \overleftarrow{P}_t) + 2\gamma \overleftarrow{P}_t \right) dt \\ \qquad + \nabla J(\overleftarrow{X}_t)^T G^\dagger(\overleftarrow{X}_t) \left[ \mathcal{H}_1(\overleftarrow{X}_t, \overleftarrow{P}_t) + \alpha \mathcal{H}_2(\overleftarrow{X}_t, \overleftarrow{P}_t) \right] dt \end{cases} \quad +$$

$$A: \begin{cases} d\overleftarrow{X}_t = \sigma(T-t)^2 \left( -\overleftarrow{P}_t dt - \alpha \nabla J(\overleftarrow{X}_t) G^\dagger(\overleftarrow{X}_t) J(\overleftarrow{X}_t) dt \right) \\ d\overleftarrow{P}_t = 0. \end{cases}$$

Integrating $O$ step from $t = t_k$ to $t = t_{k+1}$ with $\overleftarrow{X}_t$ and $\sigma(T-t)$ frozen at $t = t_k$. Then, it gives

$$\begin{cases} \overleftarrow{X}^O_{t_{k+1}} = \overleftarrow{X}_{t_k} \\ \overleftarrow{P}^O_{t_{k+1}} = a_{N-k}\Pi(\overleftarrow{X}_{t_k})\overleftarrow{P}_{t_k} + \sqrt{1 - a^2_{N-k}}\Pi(\overleftarrow{X}_{t_k})\bar{\zeta}_k, \end{cases}$$

where $a_{N-k} = e^{-\gamma\sigma^2_{N-k}\Delta t}$ and we again used the assumption that $\overleftarrow{P}_{t_k} \in T_{\overleftarrow{X}_{t_k}}\Sigma$, which will be guaranteed via the collapsing technique later on. Next, integrating $\tilde{B}$ step with the same integration domain with $\overleftarrow{X}_t, \sigma(t)$ frozen at $t = t_k$ gives:

$$\begin{cases} \overleftarrow{X}^{O\tilde{B}}_{t_{k+1}} = \overleftarrow{X}^O_{t_{k+1}} + 0 = \overleftarrow{X}_{t_k} \\ \overleftarrow{P}^{O\tilde{B}}_{t_{k+1}} = \overleftarrow{P}^O_{t_{k+1}} + \sigma^2_{N-k}\Pi(\overleftarrow{X}_{t_k})\left(\nabla f(\overleftarrow{X}_{t_k}) + 2\gamma\nabla_p\ln q_{T-t_k}(\overleftarrow{X}_{t_k},\overleftarrow{P}_{t_k}) + 2\gamma\overleftarrow{P}_{t_k}\right)\Delta t \\ \qquad\quad + \sigma^2_{N-k}\nabla J(\overleftarrow{X}_{t_k})^T G^\dagger(\overleftarrow{X}_{t_k})\left[\mathcal{H}_1(\overleftarrow{X}_{t_k},\overleftarrow{P}_{t_k}) + \alpha\mathcal{H}_2(\overleftarrow{X}_{t_k},\overleftarrow{P}_{t_k})\right]\Delta t \end{cases}$$

Similarly, integrating $A$ step with frozen $\overleftarrow{X}_t, \sigma(t)$ gives :

$$\begin{cases} \overleftarrow{X}_{t_{k+1}} = \overleftarrow{X}^{O\tilde{B}}_{t_{k+1}} - \sigma^2_{N-k}P^{O\tilde{B}}_k\Delta t - \alpha\sigma^2_{N-k}\nabla J(\overleftarrow{X}_{t_k})^T G(\overleftarrow{X}_{t_k})^\dagger J(\overleftarrow{X}_{t_k})\Delta t \\ \overleftarrow{P}_{t_{k+1}} = \overleftarrow{P}^{O\tilde{B}}_{t_{k+1}} + 0 = \overleftarrow{P}^{O\tilde{B}}_{t_{k+1}} \end{cases}$$

which is equivalent to

$$\begin{cases} \overleftarrow{P}_{t_{k+1}} = \Pi(\overleftarrow{X}_{t_k})\left[a_{N-k}\overleftarrow{P}_{t_k} + \sigma^2_{N-k}\left(\nabla f(\overleftarrow{X}_{t_k}) + 2\gamma\nabla_p\ln q_{T-t_k}(\overleftarrow{X}_{t_k},\overleftarrow{P}_{t_k}) + 2\gamma\overleftarrow{P}_{t_k}\right)\Delta t\right] \\ \quad + \sqrt{1 - a^2_{N-k}}\Pi(\overleftarrow{X}_{t_k})\bar{\zeta}_k + \sigma^2_{N-k}\nabla J(\overleftarrow{X}_{t_k})^T G^\dagger(\overleftarrow{X}_{t_k})\left[\mathcal{H}_1(\overleftarrow{X}_{t_k},\overleftarrow{P}_{t_k}) + \alpha\mathcal{H}_2(\overleftarrow{X}_{t_k},\overleftarrow{P}_{t_k})\right]\Delta t \\ \overleftarrow{X}_{t_{k+1}} = \overleftarrow{X}_{t_k} - \sigma^2_{N-k}\overleftarrow{P}_{t_{k+1}}\Delta t - \alpha\sigma^2_{N-k}\nabla J(\overleftarrow{X}_{t_k})^T G(\overleftarrow{X}_{t_k})^\dagger J(\overleftarrow{X}_{t_k})\Delta t \end{cases}$$

Similarly as before, we note that $\overleftarrow{P}_{t_{k+1}}$ can be recovered as $\overleftarrow{P}_{t_{k+1}} = \Pi(\overleftarrow{X}_{t_{k-1}})\left(\frac{\overleftarrow{X}_{t_{k-1}} - \overleftarrow{X}_{t_k}}{\sigma^2_{N-k+1}\Delta t}\right)$ from the previous recursion,

and it can be approximated by $\overleftarrow{\tilde{P}}_{t_{k+1}} \approx \Pi(\overleftarrow{X}_{t_k})\left(\frac{\overleftarrow{X}_{t_{k-1}} - \overleftarrow{X}_{t_k}}{\sigma^2_{N-k+1}\Delta t}\right)$ with $\mathcal{O}(\Delta t)$ error. Therefore, we can collapse these two

position-momentum updates into one single position update with approximated $\overleftarrow{\tilde{P}}_{t_{k+1}}$ as follows:

$$\overleftarrow{X}_{t_{k+1}} = \overleftarrow{X}_{t_k} + \sigma^2_{N-k}\Delta t\sqrt{1 - a^2_{N-k}}\Pi(\overleftarrow{X}_{t_k})\bar{\zeta}_k - \alpha\sigma^2_{N-k}\nabla J(\overleftarrow{X}_{t_k})^T G(\overleftarrow{X}_{t_k})^\dagger J(\overleftarrow{X}_{t_k})\Delta t$$
$$- \sigma^2_{N-k}\Delta t\Pi(\overleftarrow{X}_{t_k})\left[a_{N-k}\overleftarrow{\tilde{P}}_{t_k} + \sigma^2_{N-k}\left(\nabla f(\overleftarrow{X}_{t_k}) + 2\gamma\left(\nabla_p\ln q_{T-t_k}(\overleftarrow{X}_{t_k},\overleftarrow{\tilde{P}}_{t_k}) + \overleftarrow{\tilde{P}}_{t_k}\right)\right)\Delta t\right]$$
$$- \sigma^4_{N-k}\Delta t^2\nabla J(\overleftarrow{X}_{t_k})^T G^\dagger(\overleftarrow{X}_{t_k})\left[\mathcal{H}_1(\overleftarrow{X}_{t_k},\overleftarrow{\tilde{P}}_{t_k}) + \alpha\mathcal{H}_2(\overleftarrow{X}_{t_k},\overleftarrow{\tilde{P}}_{t_k})\right].$$

Finally, we recover the following discretization of the backward SDE of ULLA by changing index $k \leftarrow N - (k+1)$:

$$X'_k = X'_{k+1} + \sigma^2_{k+1}\Delta t\sqrt{1 - a^2_{k+1}}\Pi(X'_{k+1})\zeta'_{k+1} - \alpha\sigma^2_{k+1}\nabla J(X'_{k+1})^T G(X'_{k+1})^\dagger J(X'_{k+1})\Delta t -$$
$$\sigma^2_{k+1}\Delta t\Pi(X'_{k+1})\left[a_{k+1}\tilde{P}'_{k+1} + \sigma^2_{k+1}\left(\nabla f(X'_{k+1}) + 2\gamma\left(\nabla_p\ln q_{k+1}(X'_{k+1},\tilde{P}'_{k+1}) + \tilde{P}'_{k+1}\right)\right)\Delta t\right]$$
$$- \sigma^4_{k+1}\Delta t^2\nabla J(X'_{k+1})^T G^\dagger(X'_{k+1})\left[\mathcal{H}_1(X'_{k+1},\tilde{P}'_{k+1}) + \alpha\mathcal{H}_2(X'_{k+1},\tilde{P}'_{k+1})\right] \qquad \text{(Backward-ULLA)}$$

with $\tilde{P}'_{k+1} := \Pi(X'_{k+1}) \left( \frac{X'_{k+2}-X'_{k+1}}{\sigma^2_{k+2}\Delta t} \right)$ for $k \in \{0, ..., N-2\}$ and $\tilde{P}'_N = \Pi(X'_N)\zeta \in T^\Sigma_{X'_N}$ where $\zeta \sim \mathcal{N}(0, I)$.

**Discretized algorithm of constrained underdamped via Lagrangian multiplier**     Similar to constrained overdamped with Lagrangian multiplier, we replace the explicit normal drifts $-\alpha \nabla J^T G^\dagger J, \nabla J^T G^\dagger \mathcal{H}_1, \nabla J^T G^\dagger \mathcal{H}_2$ terms by a position projection via Lagrangian multipliers $\lambda$ at each step so that $J(x) = 0$ is satisfied. In the collapsed underdamped case, we also used the approximation to re-tangent the momentum $P$ by $\tilde{P}$. So, no separate momentum projection is required. Under this idea, we recover the following discretization of constrained underdamped Langevin dynamics via Lagrangian multiplier:

$$X_{k+1} = X_k + \sigma^2_k \Delta t \Pi(X_k) \left[ a_k \tilde{P}_k - \sigma^2_k \nabla f(X_k)\Delta t + \sqrt{1 - a^2_k} \zeta_k \right] + \nabla J(X_k)^T \lambda_k \qquad \text{(Forward-ULLA-P)}$$

with $\lambda_k$ such that $J(X_{k+1}) = 0$. Similarly, we obtain the following backward discretization:

$$X'_k = X'_{k+1} + \sigma^2_{k+1}\Delta t \sqrt{1 - a^2_{k+1}}\Pi(X'_{k+1})\zeta'_{k+1} + \nabla J(X'_{k+1})\lambda_{k+1} -$$
$$\sigma^2_{k+1}\Delta t \Pi(X'_{k+1}) \left[ a_{k+1}\tilde{P}'_{k+1} + \sigma^2_{k+1} \left( \nabla f(X'_{k+1}) + 2\gamma \left( \nabla_p \ln q_{k+1}(X'_{k+1}, \tilde{P}'_{k+1}) + \tilde{P}'_{k+1} \right) \right) \Delta t \right]$$
$$\text{(Backward-ULLA-P)}$$

with $\lambda_{k+1}$ such that $J(X'_k) = 0$.

## C. DT-ELBO on Riemannian Manifold

In this section, we derive the DT-ELBO variational bounds of KL-divergence between the initial data distribution and the generated data distribution on the Riemannian manifold $\Sigma := \{x \in \mathbb{R}^d \mid h(x) = 0, g(x) \leq 0\}$.

Let $\{x_0, \ldots, x_N\} \subset \Sigma$ be the forward position trajectory produced by $N$ steps of a discretized sampler, and let $q(x_{k+1} \mid c_k)$ and $p_\theta(x_k \mid d_k)$ denote the forward and backward transition densities, respectively. The context $c_k, d_k$ depends on the history $\{x_0, \ldots, x_k\}$: for the overdamped case we take $c_k = \{x_k\}, d_k = \{x_{k+1}\}$, while for the (collapsed) underdamped case we use $c_k = \{x_k, x_{k-1}\}$ (with $c_0 = \{x_0, p_0\}$) and $d_k = \{x_{k+1}, x_{k+2}\}$ (with $d_{N-1} = \{x_N, p_N\}$). We set $q_0(x) = q_{\text{data}}(x)$ as the data distribution and $p(x) = p_N(x)$ as the prior. In what follows, all densities are understood with respect to the surface measure $d\sigma_\Sigma$ on $\Sigma$, and, for clarity of exposition, we will assume that Lagrangian multiplier methods (OLLA-P, ULLA-P) are used each iterate to satisfy $x_k \in \Sigma$; see the remark below for when this may fail under discretization.

**Remark 6** (On constraint enforcement and well-definedness of the DT-ELBO).     The proposed discretized constrained Langevin dynamics with landing does *not* automatically guarantee $x_k \in \Sigma$ at every step. By contrast, the RDDPM formulation (Liu et al., 2026) enforces feasibility at each time by solving the Lagrange multiplier system via Newton's method (see Proposition B.1, Proposition B.2) so that

$$J(X_t) = 0 \quad \text{with} \quad \nabla J(X_t)^T P_t = 0 \quad \text{(if underdamped)}$$

ensuring $X_t \in \Sigma$ and $P_t \in T_{X_t}\Sigma$ exactly. Throughout our DT-ELBO derivation we adopt the same *feasible-trajectory assumption*: we assume the multiplier solve succeeds by Newton's method so that the sampled forward path lies on $\Sigma$, i.e., $x_k \in \Sigma$ for all $k \in \{1, ..., N\}$.

This assumption is not merely cosmetic. If some $x_k \notin \Sigma$, the conditional densities that appear in the ELBO may be undefined or degenerate (e.g., in the overdamped case $p_\theta(x_k \mid x_{k+1})/q(x_{k+1} \mid x_k)$, and in the underdamped case $p_\theta(x_{k-1} \mid x_k, x_{k+1})/q(x_{k+1} \mid x_k, x_{k-1})$). Such off-manifold iterates can induce singularities in the ELBO and render the induced NLL numerically unstable even under small constraint violations. For this reason, in our experiments we do not report test NLL; instead, we evaluate with task-specific metrics which reflect downstream performance.

To solidify our framework, we later introduce the Conditional Wasserstein Path Matching (CWPM) framework in Appendix D, which drops the feasibility-trajectory assumption. Its training loss has the same DT-ELBO form up to the choice of training loss weights, justifying that the CWPM framework shares the same principle with DT-ELBO framework in the constrained Langevin dynamics with landing as the step size $\Delta t \to 0$.

**DT-ELBO for overdamped Langevin**     From the Markov property, the densities $q(x_0, x_{1:N}), p_\theta(x_{1:N}|x_0)$ are given by

$$q(x_1, ...x_N|x_0) = \prod_{k=0}^{N-1} q(x_{k+1}|x_k), \quad p_\theta(x_0, x_{1:N}) = p(x_N) \prod_{k=0}^{N-1} p_\theta(x_k|x_{k+1})$$

The common goal suggested in DDPM (Ho et al., 2020) (Euclidean space) or RDDPM (Liu et al., 2026) (Riemannian manifold) is to minimize $\mathsf{KL}^\Sigma(q_0||p_\theta)$. For this, we observe that

$$\mathsf{KL}^\Sigma(q_0(x_0)||p_\theta(x_0)) = \int q_0(x_0) \ln q_0(x_0) d\sigma_\Sigma(x_0)$$
$$- \underbrace{\int q_0(x_0) \ln \left( \int p_\theta(x_{0:N}) d\sigma_\Sigma(x_{1:N}) \right) d\sigma_\Sigma(x_0)}_{\text{Term (1)}}$$

and

$$\text{Term (1)} = \int q_0(x_0) \ln \left( \int \frac{p_\theta(x_{0:N})}{q(x_{1:N}|x_0)} q(x_{1:N}|x_0) d\sigma_\Sigma(x_{1:N}) \right) d\sigma_\Sigma(x_0)$$
$$\overset{\text{Jensen}}{\geq} \int q_0(x_0) q(x_{1:N}|x_0) \ln \left( \frac{p_\theta(x_{0:N})}{q(x_{1:N}|x_0)} \right) d\sigma_\Sigma(x_{1:N})$$
$$= \int q(x_{0:N}) \left( \sum_{k=0}^{N-1} \ln \frac{p_\theta(x_k|x_{k+1})}{q(x_{k+1}|x_k)} + \ln p(x_N) \right) d\sigma_\Sigma(x_{1:N})$$

Because we want to minimize $\mathsf{KL}(q_0(x_0)||p_\theta(x_0))$, the training loss $L^{\text{over}}(\theta)$ can be set to

$$L^{\text{over}}(\theta) = -\int q(x_{0:N}) \sum_{k=0}^{N-1} \ln p_\theta(x_k|x_{k+1}) d\sigma_\Sigma(x_{1:N}) = -\mathbb{E}_{q(x_{0:N})} \left[ \sum_{k=0}^{N-1} \ln p_\theta(x_k|x_{k+1}) \right]$$

where the product surface measure is defined to be $d\sigma_\Sigma(x_1, ..x_N) := \prod_{k=1}^{N} d\sigma_\Sigma(x_k)$.

**Lemma C.1** (Backward transition density – overdamped (Liu et al., 2026)). *Suppose $x_{k+1} \in int(\Sigma)$ and assume the followings hold:*

1. *There exists a measurable set $\mathcal{F}_{x_{k+1}} \subset T_{x_{k+1}}\Sigma$ such that, for every $\eta \in \mathcal{F}_{x_{k+1}}$, the Newton's method returns a unique pair $(x, \lambda), x \in int(\Sigma)$ solving*

$$x = x_{k+1} + \mu^o_{k+1}(x_{k+1}) + \sigma_{k+1}\sqrt{\Delta t}\eta + \nabla J(x_{k+1})\lambda, \quad \text{with } \lambda \text{ s.t. } J(x) = 0$$

   *with the minimal-displacement normal correction and it fails for $\eta \notin \mathcal{F}_{x_{k+1}}$*

2. *The solver success probability is $1 - \epsilon_{x_{k+1}} := \mathbb{P}(\eta \in \mathcal{F}_{x_{k+1}}) \in (0, 1]$.*

3. *The map $\Phi_{k+1} : \mathcal{F}_{x_{k+1}} \to \Sigma_{x_{k+1}} := \Phi_{k+1}(\mathcal{F}_{x_{k+1}}) \subset int(\Sigma)$ with $\Phi_{k+1}(\eta) = x$ is a $C^1$ bijection.*

*Then, the backward transition density of OLLA-P with respect to surface measure $d\sigma_\Sigma$ is given as $p_\theta(x_k|x_{k+1})$ :*

$$p_\theta(x_k|x_{k+1}) = \frac{|\det(U(x_{k+1})^T U(x_k))|}{(2\pi\sigma^2_{k+1}\Delta t)^{\frac{d-m}{2}}(1 - \epsilon_{x_{k+1}})} \exp \left( -\frac{\|\Pi(x_{k+1})(x_k - \mu^o_{k+1}(x_{k+1}))\|^2}{2\sigma^2_{k+1}\Delta t} \right)$$

*for $x_k \in \Sigma_{x_{k+1}}$, and $p_\theta(x_k|x_{k+1}) = 0$ outside of $\Sigma_{x_{k+1}}$, where*

$$\mu^o_{k+1}(x_{k+1}) := x_{k+1} + \frac{1}{2}\sigma^2_{k+1}\Delta t\Pi(x_{k+1}) \left[ \nabla f(x_{k+1}) + s^{(k+1)}_\theta(x_{k+1}) \right]$$

*is the backward mean vector, and $U(x)$ is the orthonormal matrix whose column vectors form an orthonormal basis of $T_x\Sigma$ so that $U(x)U(x)^T = \Pi(x)$*

*Proof.* Recall that the backward discretization of OLLA-P is given as below:

$$x_k = x_{k+1} + \frac{1}{2}\sigma^2_{k+1}\Pi(x_{k+1}) \left[ \nabla f(x_{k+1}) + 2s^{(k+1)}_\theta(x_{k+1}) \right] \Delta t + \sigma_{k+1}\sqrt{\Delta t}\Pi(x_{k+1})\zeta_{k+1} + \nabla J(x_{k+1})\lambda_{k+1}$$

with $\lambda_{k+1}$ such that $J(x_k) = 0$.

Let $\eta \sim \mathcal{N}(0, I_{d-m})$ on $T_{x_{k+1}}\Sigma$. Conditioning on the success event $\{\eta \in \mathcal{F}_{x_{k+1}}\}$, the conditional density of $\eta$ is given by

$$\phi(\zeta) = \frac{1}{(2\pi)^{\frac{d-m}{2}}(1 - \epsilon_{x_{k+1}})} \exp\left(-\frac{1}{2}\|\eta\|^2\right) \mathbb{1}_{\eta \in \mathcal{F}_{x_{k+1}}}$$

From the assumption, for each $\eta \in \mathcal{F}_{x_{k+1}}$, there is a unique $x = \Phi_{k+1}(\eta) \in \Sigma$ solving

$$x = \mu^o_{k+1}(x_{k+1}) + \sigma_{k+1}\sqrt{\Delta t}\eta + \nabla J(x_{k+1})\lambda$$

with $\lambda$ such that $J(x) = 0$. Because $\Phi_{k+1}$ is bijection from $\mathcal{F}_{x_{k+1}} \subset T_{x_{k+1}}\Sigma$ onto $\Sigma_{x_{k+1}} := \Phi_{k+1}(\mathcal{F}_{x_{k+1}}) \subset \Sigma$, we can define $G_{x_{k+1}} := \Phi_{k+1}^{-1}$ and its Jacobian is given by $DG_{x_{k+1}}(x) = (\sigma_{k+1}\sqrt{\Delta t})^{-1}U(x_{k+1})^T U(x)$ for $x \in \Sigma_{x_{k+1}}$. Hence, we have

$$\left|\det(DG_{x_{k+1}}(x))\right| = (\sigma_{k+1}\sqrt{\Delta t})^{-(d-m)}\left|\det(U(x_{k+1})^T U(x))\right|$$

Lastly, we observe that, for $x \in \Sigma_{x_{k+1}}$, the pushforward of the conditional density $\phi$ by $\Phi_{k+1}$ yields the following density with respect to $d\sigma_\Sigma$:

$$p_\theta(x_k|x_{k+1}) = \frac{\left|\det(U(x_{k+1})^T U(x_k))\right|}{(2\pi\sigma^2_{k+1}\Delta t)^{\frac{d-m}{2}}(1 - \epsilon_{x_{k+1}})} \exp\left(-\frac{\|\Pi(x_{k+1})^T(x_k - \mu^o_{k+1}(x_{k+1}))\|^2}{2\sigma^2_{k+1}\Delta t}\right)$$

which becomes zero outside $\Sigma_{x_{k+1}}$ (equivalently, when projection failure happens). $\qquad\square$

Therefore, assuming the forward path trajectory $\{x_0, ..., x_N\} \subset \text{int}(\Sigma)$, the training loss has the following upper bound:

$$L^{\text{over}}(\theta) = -\mathbb{E}_{q(x_{0:N})}\left[\sum_{k=0}^{N-1}\ln p_\theta(x_k|x_{k+1})\right] \le \mathbb{E}_{q(x_{0:N})}\left[\sum_{k=0}^{N-1}\left(\frac{\|\Pi(x_{k+1})^T(x_k - \mu^o_{k+1}(x_{k+1}))\|^2}{2\sigma^2_{k+1}\Delta t}\right)\right] + C$$

$$\text{(Training loss-OLLA-P)}$$

where $C := \sum_{k=0}^{N-1}\left[-\ln\left(\frac{\left|\det(U(x_{k+1})^T U(x_k))\right|}{(2\pi\sigma^2_{k+1}\Delta t)^{\frac{d-m}{2}}}\right)\right]$ is constant with respect to $\theta$ and the last inequality is obtained using $\epsilon_{x_{k+1}} \le 1$.

**DT-ELBO for underdamped Langevin** In the collapsed underdamped setting, we keep only positions, which makes the forward chain a second-order Markov chain in $\{x_0, ...x_N\}$. Similarly, applying the same Jensen inequality argument used for the overdamped case gives the ELBO:

$$\text{Term (1)} = \int q_0(x_0) \ln\left(\int \frac{p_\theta(x_{0:N})}{q(x_{1:N}|x_0, p_0)}q(p_0|x_0)q(x_{1:N}|x_0, p_0)d\sigma_{T_{x_0}\Sigma}(p_0)d\sigma_\Sigma(x_{1:N})\right)d\sigma_\Sigma(x_0)$$

$$\overset{\text{Jensen}}{\ge} \int q(x_0, p_0)q(x_{1:N}|x_0, p_0)\ln\left(\frac{p_\theta(x_{0:N})}{q(x_{1:N}|x_0, p_0)}\right)d\sigma_{T_{x_0}\Sigma}(p_0)d\sigma_\Sigma(x_{0:N})$$

$$\overset{\text{Jensen}}{\ge} \mathbb{E}_{q(x_{0:N}, p_0)}\mathbb{E}_{p(p_N|x_N)}\left[\sum_{k=1}^{N-1}\ln\left(\frac{p_\theta(x_{k-1}|x_k, x_{k+1})}{q(x_{k+1}|x_k, x_{k-1})}\right) + \ln\left(\frac{p_\theta(x_{N-1}|x_N, p_N)}{q(x_1|x_0, p_0)}\right) + \ln p(x_N)\right]$$

where the last inequality holds because $p_\theta(x_{0:N})$ has the following form:

$$p_\theta(x_{0:N}) = p(x_N)\int p(p_N|x_N)p_\theta(x_{N-1}|x_N, p_N)\prod_{k=1}^{N-1}p_\theta(x_{k-1}|x_k, x_{k+1})d\sigma_{T_{x_N}\Sigma}(p_N)$$

Therefore, the training loss $L^{\text{under}}(\theta)$ is naturally given as follows to minimize $\mathsf{KL}^\Sigma(q_0(x_0)\|p_\theta(x_0))$:

$$L^{\text{under}}(\theta) = -\mathbb{E}_{q(x_{0:N}, p_0)}\mathbb{E}_{\rho_N(p_N|x_N)}\left[\sum_{k=0}^{N-2}\ln p_\theta(x_k|x_{k+1}, x_{k+2}) + \ln p_\theta(x_{N-1}|x_N, p_N)\right]$$

$$= -\mathbb{E}_{q(x_{0:N}, p_0)}\mathbb{E}_{\rho_N(p_N|x_N)}\left[\sum_{k=0}^{N-1}\ln p_\theta(x_k|x_{k+1}, x_{k+2})\right]$$

where $\rho_N(p_N|x_N) \propto \exp\left(-\frac{1}{2}\|\Pi(x_N)p_N\|_2^2\right)$ is the density of the momentum prior density, and we used the notation abusing $x_{N+1} := p_N$ for notational convenience.

---

**Lemma C.2** (Backward transition density – underdamped). *Suppose $x_{k+1} \in int(\Sigma)$ and assume the followings hold:*

1. *There exists a measurable set $\mathcal{F}_{x_{k+1}} \subset T_{x_{k+1}}\Sigma$ such that, for every $\eta \in \mathcal{F}_{x_{k+1}}$, the Newton's method returns a unique pair $(x, \lambda), x \in int(\Sigma)$ solving*

$$x = \mu_{k+1}^u(x_{k+1}, x_{k+2}) + \sigma_{k+1}^2 \Delta t \sqrt{1 - a_{k+1}^2}\, \eta + \nabla J(x_{k+1})\lambda, \text{ with } \lambda \text{ s.t. } J(x) = 0$$

   *with the minimal-displacement normal correction and it fails for $\eta \notin \mathcal{F}_{x_{k+1}}$*

2. *The solver success probability is $1 - \epsilon_{x_{k+1}} := \mathbb{P}(\eta \in \mathcal{F}_{x_{k+1}}) \in (0, 1]$.*

3. *The map $\Phi_{k+1} : \mathcal{F}_{x_{k+1}} \to \Sigma_{x_{k+1}} := \Phi_{k+1}(\mathcal{F}_{x_{k+1}}) \subset int(\Sigma)$ with $\Phi_{k+1}(\eta) = x$ is a $C^1$ bijection.*

*Then, the backward transition density of ULLA-P with respect to surface measure $d\sigma_\Sigma$ is given as $p_\theta(x_k|x_{k+1})$ :*

$$p_\theta(x_k|x_{k+1}) = \frac{\left|\det(U(x_{k+1})^T U(x_k))\right|}{(2\pi\sigma_{k+1}^4 \Delta t^2(1 - a_{k+1}^2))^{\frac{d-m}{2}}(1 - \epsilon_{x_{k+1}})} \exp\left(-\frac{\|\Pi(x_{k+1})^T(x_k - \mu_{k+1}^u(x_{k+1}, x_{k+2}))\|^2}{2\sigma_{k+1}^4 \Delta t^2(1 - a_{k+1}^2)}\right)$$

*for $x_k \in \Sigma_{x_{k+1}}$, and $p_\theta(x_k|x_{k+1}) = 0$ outside of $\Sigma_{x_{k+1}}$, where*

$$\mu_{k+1}^u = x_{k+1} + \sigma_{k+1}^2 \Delta t \Pi(x_{k+1})\left[a_{k+1}\tilde{p}_{k+1} + \sigma_{k+1}^2 \Delta t\left(\nabla f(x_{k+1}) + s_{k+1}^\theta(x_{k+1}, \tilde{p}_{k+1})\right)\right]$$

*is the backward mean vector.*

---

*Proof.* Recall that the backward discretization of ULLA-P is given as below:

$$\begin{aligned}
x_k =&\, x_{k+1} + \sigma_{k+1}^2 \Delta t \sqrt{1 - a_{k+1}^2}\, \Pi(x_{k+1})\zeta_{k+1} + \nabla J(x_{k+1})\lambda_{k+1} + \\
&\, \sigma_{k+1}^2 \Delta t \Pi(x_{k+1})\left[a_{k+1}\tilde{p}_{k+1} + \sigma_{k+1}^2\left(\nabla f(x_{k+1}) + s_{k+1}^\theta(x_{k+1}, \tilde{p}_{k+1})\right)\Delta t\right]
\end{aligned}$$

with $\lambda_{k+1}$ such that $J(x_k) = 0$. Using the same proof as the overdamped case, we observe that $\Phi_{k+1}$ is bijection from $\mathcal{F}_{x_{k+1}} \subset T_{x_{k+1}}\Sigma$ onto $\Sigma_{x_{k+1}} := \Phi_{k+1}(\mathcal{F}_{x_{k+1}}) \subset \Sigma$, and we can define $G_{x_{k+1}} := \Phi_{k+1}^{-1}$ whose Jacobian is given by $DG_{x_{k+1}}(x) = (\sigma_{k+1}^2 \Delta t \sqrt{1 - a_{k+1}^2})^{-1} U(x_{k+1})^T U(x)$ for $x \in \Sigma_{x_{k+1}}$. Therefore, the determinant of this is:

$$\left|\det(DG_{x_{k+1}}(x))\right| = (\sigma_{k+1}^2 \Delta t \sqrt{1 - a_{k+1}^2})^{-(d-m)}\left|\det(U(x_{k+1})^T U(x))\right|$$

Lastly, we observe that, for $x \in \Sigma_{x_{k+1}}$, the pushforward of the conditional density of $\eta$ (defined on the overdamped case proof) by $\Phi_{k+1}$ yields the following density with respect to $d\sigma_\Sigma$:

$$p_\theta(x_k|x_{k+1}) = \frac{\left|\det(U(x_{k+1})^T U(x_k))\right|}{(2\pi\sigma_{k+1}^4 \Delta t^2(1 - a_{k+1}^2))^{\frac{d-m}{2}}(1 - \epsilon_{x_{k+1}})} \exp\left(-\frac{\|\Pi(x_{k+1})^T(x_k - \mu_{k+1}^u(x_{k+1}, x_{k+2}))\|^2}{2\sigma_{k+1}^4 \Delta t^2(1 - a_{k+1}^2)}\right)$$

which becomes zero outside $\Sigma_{x_{k+1}}$ (equivalently, when projection failure happens). $\square$

---

Therefore, assuming the forward path trajectory $\{x_0, ..., x_N\} \subset int(\Sigma)$, the training loss has the following upper bound (with notation $x_{N+1} := p_N$):

$$L^{\text{under}}(\theta) = -\mathbb{E}_{q(x_{0:N}, p_0)}\mathbb{E}_{\rho_N(p_N|x_N)}\left[\sum_{k=0}^{N-1} \ln p_\theta(x_k|x_{k+1}, x_{k+2})\right] \qquad \text{(Training loss-ULLA-P)}$$

$$\leq \mathbb{E}_{q(x_{0:N})}\mathbb{E}_{\rho_N(p_N|x_N)}\left[\sum_{k=0}^{N-1}\left(\frac{\|\Pi(x_{k+1})^T(x_k - \mu_{k+1}^u(x_{k+1}, x_{k+2}))\|^2}{2\sigma_{k+1}^4 \Delta t^2(1 - a_{k+1}^2)}\right)\right] + C$$

where $C := \sum_{k=0}^{N-1} \left[ -\ln\left( \frac{|\det(U(x_{k+1})^T U(x_k))|}{(2\pi\sigma_{k+1}^4 \Delta t^2 (1-a_{k+1}^2))^{\frac{d-m}{2}}} \right) \right]$ is constant with respect to $\theta$ and the last inequality is again obtained using $\epsilon_{x_{k+1}} \leq 1$.

## D. Conditional Wasserstein Path Matching (CWPM)

Let $q_0, q_1, ..., q_N$ be the marginal forward probability densities at each step, evolved by discrete forward transition kernels $Q_k = q(x_{k+1}|c_k)$. Similarly, define $p_N, p_{N-1}^\theta, ..., p_0^\theta$ to be the marginal backward probability densities driven by parameterized backward transition kernels $T_{k+1}^\theta = p^\theta(x_k|d_k)$. Similarly as in DT-ELBO, the context vector is fixed to be $c_k = \{x_k\}, d_k = \{x_{k+1}\}$ for the overdamped case, while for the (collapsed) underdamped case we use $c_k = \{x_k, x_{k-1}\}$ (with $c_0 = \{x_0, p_0\}$) and $d_k = \{x_{k+1}, x_{k+2}\}$ (with $d_{N-1} = \{x_N, p_N\}$).

Our goal is to minimize $W_2(q_0, p_0^\theta)$ so that the Wasserstein-2 distance between data distribution and generated data distribution becomes close to each other.

**CWPM framework for the overdamped**    We first define **circuitous** density at step $k$ as

$$\sigma_k := q_k T_k^\theta T_{k-1}^\theta \cdots T_1^\theta \quad \text{for } k \in \{1, ..., N\} \qquad \sigma_0 := q_0.$$

We assume that for any $\mu, \nu \in \mathcal{P}_2(\mathbb{R}^d)$, there exists $K_{k+1} > 0$ such that

$$W_2(\mu T_{k+1}^\theta, \nu T_{k+1}^\theta) \leq K_{k+1} W_2(\mu, \nu) + \mathcal{O}(\sqrt{\Delta t}) \qquad \text{(stepwise-Lipschitz)}$$

Under this assumption, we can choose $\Lambda_{k+1} > 0$ such that

$$W_2(\sigma_k, \sigma_{k+1}) \leq \Lambda_{k+1} W_2(q_k, q_{k+1} T_{k+1}^\theta) + \mathcal{O}(\sqrt{\Delta t})$$

for $k \in \{0, ..., N-1\}$. We note that Lemma D.1 implies that such $\Lambda_k$ exists without stepwise-Lipschitz assumption when the function class of score function and the constraint functions are sufficiently regular.

Under this setup, from the triangular inequality of $W_2$, it holds that

$$W_2(q_0, p_0^\theta) \leq W_2(q_0, \sigma_N) + W_2(\sigma_N, p_0^\theta) \leq \sum_{k=0}^{N-1} W_2(\sigma_k, \sigma_{k+1}) + W_2(\sigma_N, p_0^\theta)$$

$$\leq \sum_{k=0}^{N-1} \Lambda_{k+1} W_2(q_k, q_{k+1} T_{k+1}^\theta) + \Lambda_{N+1} W_2(q_N, p_N) + \mathcal{O}(\sqrt{\Delta t})$$

$$\leq \sum_{k=0}^{N-1} \Lambda_{k+1} \left( \mathbb{E}_{x_{k+1} \sim q_{k+1}} W_2^2(q_{k|k+1}(\cdot|x_{k+1}), T_{k+1}^\theta(\cdot|x_{k+1})) \right)^{1/2} + \Lambda_{N+1} W_2(q_N, p_N) + \mathcal{O}(\sqrt{\Delta t}).$$

The last inequality follows by disintegrating the forward joint law as

$$q(x_k, x_{k+1}) = q_{k+1}(x_{k+1}) q_{k|k+1}(x_k|x_{k+1})$$

and coupling each conditional pair optimally:

$$W_2^2(q_k, q_{k+1} T_{k+1}^\theta) \leq \mathbb{E}_{x_{k+1} \sim q_{k+1}} W_2^2(q_{k|k+1}(\cdot|x_{k+1}), T_{k+1}^\theta(\cdot|x_{k+1})).$$

Now, we know that

$$T_{k+1}^\theta(\cdot|x_{k+1}) = \mathcal{N}(\mu_{k+1}^o(x_{k+1}), \sigma_{k+1}^2 \Delta t\, \Pi(x_{k+1})).$$

Let $(\mu_{k|k+1}, \Sigma_{k|k+1})$ be the conditional mean and covariance of $q_{k|k+1}(\cdot|x_{k+1})$. Then, from the relation between Gelbrich distance and 2-Wasserstein distance (Gelbrich, 1990; Borelle et al., 2024), with a nonnegative gap $\Delta_{k+1}(x_{k+1})$ independent

of $\theta$, we have:

$$
\begin{aligned}
&\mathbb{E}_{x_{k+1}\sim q_{k+1}} W_2(q_{k|k+1}, T_{k+1}^\theta(\cdot|x_{k+1}))^2 = \mathbb{E}_{x_{k+1}\sim q_{k+1}} \|\mu_{k|k+1}(x_{k+1}) - \mu_{k+1}^o(x_{k+1})\|^2 \\
&+ \mathbb{E}_{x_{k+1}\sim q_{k+1}} \left[ \mathsf{Tr}\left(\Sigma_{k|k+1}(x_{k+1})\right) + \mathsf{Tr}\left(\sigma_{k+1}^2 \Delta t \Pi(x_{k+1})\right) \right] + \mathbb{E}_{x_{k+1}\sim q_{k+1}} \Delta_{k+1}(x_{k+1}) \\
&- 2\mathbb{E}_{x_{k+1}\sim q_{k+1}} \left[ \mathsf{Tr}\left( \left(\Sigma_{k|k+1}^{1/2}(x_{k+1})\sigma_{k+1}^2 \Delta t \Pi(x_{k+1})\Sigma_{k|k+1}^{1/2}(x_{k+1})\right)^{1/2} \right) \right] \\
&= \mathbb{E}_{(x_k, x_{k+1})\sim q(x_k, x_{k+1})} \|x_k - \mu_{k+1}^o(x_{k+1})\|^2 + \mathbb{E}_{x_{k+1}\sim q_{k+1}} \mathsf{Tr}\left(\sigma_{k+1}^2 \Delta t \Pi(x_{k+1})\right) + \mathbb{E}_{x_{k+1}\sim q_{k+1}} \Delta_{k+1}(x_{k+1}) \\
&- 2\mathbb{E}_{x_{k+1}\sim q_{k+1}} \left[ \mathsf{Tr}\left( \left(\Sigma_{k|k+1}^{1/2}(x_{k+1})\sigma_{k+1}^2 \Delta t \Pi(x_{k+1})\Sigma_{k|k+1}^{1/2}(x_{k+1})\right)^{1/2} \right) \right]
\end{aligned}
$$

where we used the law of total variance for the last equality, by observing that $x_k = \mu_{k|k+1}(x_{k+1}) + \epsilon \sim q_k$ with $\mathbb{E}[\epsilon|x_{k+1}] = 0$ and $\mathrm{Cov}(\epsilon|x_{k+1}) = \Sigma_{k|k+1}$. The gap is independent of $\theta$ because $T_{k+1}^\theta$ has a fixed covariance and $\theta$ only changes its mean. Note that $\Delta_{k+1}(x_{k+1}) \geq 0$ is the distance gap between Gelbrich distance and 2-Wasserstein distance (independent of $\theta$) so that it becomes zero if the true conditional $x_k|x_{k+1}$ follows the Gaussian distribution as

$$
x_k|x_{k+1} \sim \mathcal{N}(\mu_{k|k+1}(x_{k+1}), \Sigma_{k|k+1}(x_{k+1}))
$$

**Training loss (overdamped) from CWPM**    We note that from the above bound, minimizing $\mathbb{E}_{(x_k, x_{k+1})\sim q(x_k, x_{k+1})} \|x_k - \mu_{k+1}^o(x_{k+1})\|^2$ is sufficient for closing the Wasserstein distance between $q_0$ and $p_0^\theta$. Because $\|x_k - \mu_{k+1}^o(x_{k+1})\|^2$ can be decomposed into

$$
\|x_k - \mu_{k+1}^o(x_{k+1})\|^2 = \|\Pi(x_{k+1})(x_k - \mu_{k+1}^o(x_{k+1}))\|^2 + \underbrace{\|(I - \Pi(x_{k+1}))(x_k - \mu_{k+1}^o(x_{k+1}))\|^2}_{\text{constant w.r.t } \theta}
$$

and $\mu_{k+1}^o$ does not have $\theta$ dependency on normal (second) term, the natural choice of loss (leveraging the saved forward trajectories) is

$$
L_{\mathrm{CWPM}}^{\mathrm{over}}(\theta) = \mathbb{E}_{q(x_{0:N})} \left[ \sum_{k=0}^{N-1} \lambda(k)\|\Pi(x_{k+1})\left(x_k - \mu_{k+1}^o(x_{k+1})\right)\|^2 \right] = \mathbb{E}_{q(x_{0:N})} \left[ \sum_{k=0}^{N-1} \frac{\|\Pi(x_{k+1})\left(x_k - \mu_{k+1}^o(x_{k+1})\right)\|^2}{2\sigma_{k+1}^2 \Delta t} \right]
$$

with some training loss weight $\lambda(k)$. We note that other seminal works (Ho et al., 2020; Wang et al., 2024; Karras et al., 2022) in diffusion model choose the weight proportional to the inverse of variance of corresponding term, which, in our case, becomes $\lambda(k) = \frac{1}{2\sigma_{k+1}^2 \Delta t}$ with proportional constant $1/2$. And, notably, this leads to the exactly the same training loss provided in DT-ELBO, (Lemma C.1) without the requirement $x_k \in \Sigma$.

---

**Lemma D.1** (Sufficient condition for $\Lambda_k < \infty$ – Overdamped). *Let the one-step landing backward update of OLLA (subsection Backward-OLLA) be*

$$
F_\theta^k(y, \zeta) = y + \Pi(y)b_\theta^k(y)\Delta t + \sigma_k\sqrt{\Delta t}\Pi(y)\zeta + \sigma_k^2\phi(y)\Delta t, \quad \zeta \sim \mathcal{N}(0, I_d)
$$

*where $b_\theta^k(y) := \frac{\sigma_k^2}{2}\left[\nabla f(y) + 2s_\theta^k(y)\right]$ is the drift term and $\phi(y)$ is the normal term. Assume:*

1. *(Regularity of constraint functions)    There exists constants $c_\phi^0, c_\phi^1 < \infty$ such that for any square-integrable $Y$,*

$$
\mathbb{E}\|\phi(Y)\|^2 \leq c_\phi^0 + c_\phi^1 \mathbb{E}\|Y\|^2
$$

2. *(Regularity of function class)    There exists $L_s, B_s < \infty$ independent of $\theta$ with*

$$
\|s_\theta^k(x) - s_\theta^k(y)\| \leq L_s\|x - y\|, \quad \|s_\theta^k(x)\| \leq B_s + L_s\|x\|
$$

*for all $k \in \{1, ..., N\}$ and assume $\nabla f$ is Lipschitz with constant $L_f$ so that*

$$
\|b_\theta^k(x) - b_\theta^k(y)\| \leq L_b\|x - y\|, \quad \|b_\theta^k(x)\| \leq C_b(1 + \|x\|)
$$

*for some constant $L_b, C_b$, $k \in \{1, ..., N\}$*

*Let $T_k^\theta$ be the associated Markov kernel to $F_\theta^k$. That is, $T_k^\theta(\cdot|y) = Law(F_\theta^k(y, \zeta))$. Then, for any $\mu, \nu \in \mathcal{P}_2(\mathbb{R}^d)$, we have*

$$W_2(\mu T_k^\theta, \nu T_k^\theta) \le K_k W_2(\mu, \nu) + \mathcal{O}\left(\sqrt{\Delta t} + \Delta t \left(1 + \sqrt{\mathbb{E}_{Y \sim \mu}\|Y\|^2} + \sqrt{\mathbb{E}_{Y' \sim \nu}\|Y'\|^2}\right)\right)$$

*for some constant $K_k$. Also, if $\mu$ is given as $\mu = \rho T_i^\theta T_{i-1}^\theta \ldots T_j^\theta$ for $i \ge j$ and some density $\rho \in \mathcal{P}_2(\mathbb{R}^d)$ independent of $\theta$, the supremum of second moment of $\mu$ is finite under $\theta \in \Theta$:*

$$\sup_{\theta \in \Theta} \mathbb{E}_{Y_\mu \sim \mu}\|Y_\mu\|^2 < \infty$$

*Combining these two, we can guarantee the existence of $\Lambda_k > 0$ such that*

$$W_2(\sigma_{k-1}, \sigma_k) \le \Lambda_k W_2(q_{k-1}, q_k T_k^\theta) + \mathcal{O}(\sqrt{\Delta t})$$

*for $k \in \{1, \ldots N\}$.*

*Proof.* Let $(Y, Y')$ be the synchronous coupling for $W_2(\mu, \nu)$ with shared noise $\zeta \sim \mathcal{N}(0, I)$. Set

$$\Delta F_\theta^k := F_\theta^k(Y, \zeta) - F_\theta^k(Y', \zeta) = (Y - Y') + \underbrace{\Delta t \left(\Pi(Y) b_\theta^k(Y) - \Pi(Y') b_\theta^k(Y')\right)}_{:=B}$$

$$+ \underbrace{\sigma_k \sqrt{\Delta t}(\Pi(Y) - \Pi(Y'))\zeta}_{:=N} + \underbrace{\sigma_k^2 \Delta t \left(\phi(Y) - \phi(Y')\right)}_{:=L}$$

Then, using $\|a + b + c + d\|^2 \le 4(\|a\|^2 + \|b\|^2 + \|c\|^2 + \|d\|^2)$, we have:

$$\mathbb{E}\|\Delta F_\theta^k\|^2 \le 4\mathbb{E}\|Y - Y'\|^2 + 4\mathbb{E}\|B\|^2 + 4\mathbb{E}\|N\|^2 + 4\mathbb{E}\|L\|^2$$

Now, we note that

$$B = \Delta t \left(\Pi(Y)(b_\theta^k(Y) - b_\theta^k(Y')) + (\Pi(Y) - \Pi(Y')) b_\theta^k(Y')\right)$$

and it implies

$$\|B\| \le \Delta t \left(L_b\|Y - Y'\| + 2\|b_\theta^k(Y')\|\right)$$

Therefore, using the growth bound $\|b_\theta^k(x)\| \le C_b(1 + \|x\|)$, we have

$$\mathbb{E}\|B\|^2 \le 2L_b^2 \Delta t^2 \mathbb{E}\|Y - Y'\|^2 + 8\Delta t^2 \mathbb{E}\|b_\theta^k(Y')\|^2$$
$$\le 2L_b^2 \Delta t^2 \mathbb{E}\|Y - Y'\|^2 + C_1 \Delta t^2 \left(1 + \mathbb{E}\|Y'\|^2\right)$$
$$\le 2L_b^2 \Delta t \mathbb{E}\|Y - Y'\|^2 + C_1 \Delta t^2 \left(1 + \mathbb{E}\|Y'\|^2 + \mathbb{E}\|Y\|^2\right),$$

where the last inequality uses $\Delta t \le 1$ and comes by swapping $Y, Y'$ and taking average. Here $C_1$ is a finite constant depending on $C_b$. Also, we note that, for some constant $C_2 > 0$,

$$\mathbb{E}\|N\|^2 = \sigma_k^2 \Delta t \, \mathbb{E}\left[\|(\Pi(Y) - \Pi(Y'))\zeta\|^2\right] \le C_2 \Delta t,$$

where we used $\|\Pi\| \le 1$ and $\mathbb{E}\|\zeta\|^2 = d$. Similarly, using $(a - b)^2 \le 2a^2 + 2b^2$ and Assumption 1, we observe that

$$\mathbb{E}\|L\|^2 \le 2\sigma_k^4 \Delta t^2 \left(\mathbb{E}\|\phi(Y)\|^2 + \mathbb{E}\|\phi(Y')\|^2\right) \le C_3 \Delta t^2 \left(1 + \mathbb{E}\|Y\|^2 + \mathbb{E}\|Y'\|^2\right),$$

for some constant $C_3 > 0$. By collecting all terms, we obtain

$$\mathbb{E}\|\Delta F_\theta^k\|^2 \le C_4(1 + L_b^2 \Delta t)\mathbb{E}\|Y - Y'\|^2 + C_5 \Delta t + C_6 \Delta t^2 \left(1 + \mathbb{E}\|Y\|^2 + \mathbb{E}\|Y'\|^2\right),$$

for finite constants $C_4, C_5, C_6 > 0$ independent of $\theta$ and $\Delta t$. By taking square-root and using $\sqrt{u + v + w} \le \sqrt{u} + \sqrt{v} + \sqrt{w}$, we get

$$W_2(\mu T_k^\theta, \nu T_k^\theta) \le K_k W_2(\mu, \nu) + \mathcal{O}\left(\sqrt{\Delta t} + \Delta t \left(1 + \sqrt{\mathbb{E}_{Y \sim \mu}\|Y\|^2} + \sqrt{\mathbb{E}_{Y' \sim \nu}\|Y'\|^2}\right)\right),$$

for some constant $K_k$ independent of $\theta$. Also, following the same linear-growth estimates, one can show that

$$\mathbb{E}\|F_\theta^k(Y,\zeta)\|^2 \leq \left(1 + C_{7,k}\Delta t + C_{8,k}\Delta t^2\right)\mathbb{E}\|Y\|^2 + C_{9,k}\Delta t + C_{10,k}\Delta t^2,$$

for finite constants $C_{7,k}, C_{8,k}, C_{9,k}, C_{10,k} > 0$ independent of $\theta$. Indeed, writing

$$F_\theta^k(Y,\zeta) = Y + \Delta t\,\Pi(Y)b_\theta^k(Y) + \sigma_k\sqrt{\Delta t}\Pi(Y)\zeta + \sigma_k^2\Delta t\,\phi(Y),$$

the cross term with $\zeta$ vanishes after conditioning on $Y$, and the remaining terms are controlled by $\|b_\theta^k(Y)\| \leq C_b(1 + \|Y\|)$ and $\mathbb{E}\|\phi(Y)\|^2 \leq c_\phi^0 + c_\phi^1\mathbb{E}\|Y\|^2$.

So, once $\mu$ is given as

$$\mu = \rho T_i^\theta T_{i-1}^\theta \ldots T_j^\theta$$

for $i \geq j$ and some density $\rho$ independent of $\theta$ with finite second moment, applying the recursive inequality above gives

$$\mathbb{E}_{Y_\mu \sim \mu}\|Y_\mu\|^2 \leq \prod_{\ell=j}^i \left(1 + C_{7,\ell}\Delta t + C_{8,\ell}\Delta t^2\right)\mathbb{E}_{Y_\rho \sim \rho}\|Y_\rho\|^2 + C_{\mathrm{hor}}\left(1 + \mathbb{E}_{Y_\rho \sim \rho}\|Y_\rho\|^2\right),$$

for some finite-horizon constant $C_{\mathrm{hor}}$ independent of $\theta$. Taking the supremum over $\theta$ implies

$$\sup_{\theta \in \Theta}\mathbb{E}_{Y_\mu \sim \mu}\|Y_\mu\|^2 < \infty,$$

because all constants and the density $\rho$ are independent of $\theta$. $\qquad\square$

**CWPM framework for the underdamped**    Let $y_k := (x_k, x_{k+1}) \in \mathbb{R}^{2d}$ with $x_k \sim q_k, x_{k+1} \sim q_{k+1}$ and let $\bar{q}_k$ be the law of $y_k$. The forward pair-kernel $\bar{Q}_k$ and backward pair-kernel $\bar{T}_{k+1}^\theta$ are

$$\bar{Q}_k\left(x_{k+1}, x_{k+2}|x_k, x_{k+1}\right) = \delta_{x_{k+1}} \otimes Q_k(x_{k+2}|x_k, x_{k+1})$$
$$\bar{T}_{k+1}^\theta\left(x_k, x_{k+1}|x_{k+1}, x_{k+2}\right) = T_{k+1}^\theta(x_k|x_{k+1}, x_{k+2}) \otimes \delta_{x_{k+1}}$$

Now, we similarly define the **circuitous** densities on pairs as

$$\bar{\sigma}_0 := \bar{q}_0, \qquad \bar{\sigma}_k := \bar{q}_k \bar{T}_k^\theta \cdots \bar{T}_1^\theta, \quad \text{for } k \in \{1, ..., N-1\}$$

Assume the stepwise Lipschitz inequality on pairs holds such that there exists $\bar{K}_{k+1} > 0$

$$W_2(\mu \bar{T}_{k+1}^\theta, \nu \bar{T}_{k+1}^\theta) \leq \bar{K}_{k+1} W_2(\mu, \nu) + \mathcal{O}(\Delta t)$$

for pair laws $\mu, \nu$ on $\mathbb{R}^{2d}$ with finite scaled pair moments. Then there exists finite $\bar{\Lambda}_{k+1} > 0$ such that

$$W_2(\bar{\sigma}_k, \bar{\sigma}_{k+1}) \leq \bar{\Lambda}_{k+1} W_2(\bar{q}_k, \bar{q}_{k+1}\bar{T}_{k+1}^\theta) + \mathcal{O}(\Delta t)$$

for $k \in \{0, ..., N-2\}$. (As in Lemma D.2) such $\Lambda_k$ exists without stepwise-Lipschitz assumption under some regularity of the score function class and constraints.

For the prior on pair chain setup, we let $\bar{p}_{N-1}^\theta$ be a terminal pair prior on $(X_{N-1}, X_N)$ induced by sampling $X_N \sim p_N, P_N \sim \Pi(X_N)\zeta$ with $\zeta \sim \mathcal{N}(0, I_d)$ so that $X_{N-1} \sim \bar{T}_N^\theta(\cdot|X_N, P_N)$. Then, we propagate backward by

$$\bar{p}_0^\theta := \bar{p}_{N-1}^\theta \bar{T}_{N-1}^\theta \cdots \bar{T}_1^\theta, \quad p_0^\theta := (\pi_1)_\# \bar{p}_0^\theta$$

where $\pi_1(x_0, x_1) = x_0$ is the projection map onto first coordinate. Since $\pi_1$ is 1-Lipschitz, $W_2(q_0, p_0^\theta) \leq W_2(\bar{q}_0, \bar{p}_0^\theta)$ holds and, from the triangle inequality for $W_2$, we have

$$\begin{aligned}
W_2(q_0, p_0^\theta) \leq W_2(\bar{q}_0, \bar{p}_0^\theta) &\leq W_2(\bar{q}_0, \bar{\sigma}_{N-1}) + W_2(\bar{\sigma}_{N-1}, \bar{p}_0^\theta)\\
&\leq \sum_{k=0}^{N-2} W_2(\bar{\sigma}_k, \bar{\sigma}_{k+1}) + W_2(\bar{\sigma}_{N-1}, \bar{p}_0^\theta)\\
&\leq \sum_{k=0}^{N-2} \bar{\Lambda}_{k+1} W_2(\bar{q}_k, \bar{q}_{k+1}\bar{T}_{k+1}^\theta) + \bar{\Lambda}_N W_2(\bar{q}_{N-1}, \bar{p}_{N-1}^\theta) + \mathcal{O}(\Delta t).
\end{aligned}$$

Because in pair conditionals the second coordinate is a Dirac mass, conditioning on the shared second coordinate gives

$$W_2^2(\bar{q}_k, \bar{q}_{k+1}\bar{T}_{k+1}^\theta) \leq \mathbb{E}_{(x_{k+1}, x_{k+2}) \sim \bar{q}_{k+1}} W_2^2(q_{k|k+1, k+2}(\cdot|x_{k+1}, x_{k+2}), T_{k+1}^\theta(\cdot|x_{k+1}, x_{k+2})).$$

Thus, as in the overdamped case, the expected squared conditional mismatch controls the corresponding pair-level Wasserstein term up to a square root and an additive constant independent of $\theta$. Also, the following decomposition holds by triangle inequality:

$$W_2(\bar{q}_{N-1}, \bar{p}_{N-1}^\theta) \leq W_2((q_N \otimes \rho_N)S_N, (p_N \otimes \rho_N)S_N) + W_2((p_N \otimes \rho_N)S_N, (p_N \otimes \rho_N)S_N^\theta)$$

where $p_N$ is the prior of position, $\rho_N(\cdot|X_N)$ is the prior of momentum defined by the law of $\Pi(x_N)\zeta$ with $\zeta \sim \mathcal{N}(0, I), x_N \sim p_N$, and each $S_N$ and $S_N^\theta$ are defined by

$$S_N(x_N, p_N) := \bar{q}_{N-1|N}(\cdot|x_N, p_N) \otimes \delta_{x_N}, \quad S_N^\theta(x_N, p_N) := \bar{T}_N^\theta(\cdot|x_N, p_N) \otimes \delta_{x_N}$$

The first term is independent of $\theta$. For the second term, using the common base coupling $(x_N, p_N) \sim p_N \otimes \rho_N$ gives

$$W_2^2((p_N \otimes \rho_N)S_N, (p_N \otimes \rho_N)S_N^\theta) \leq \mathbb{E}_{(x_N, p_N) \sim p_N \otimes \rho_N} W_2^2(\bar{q}_{N-1|N}(\cdot|x_N, p_N), \bar{T}_N^\theta(\cdot|x_N, p_N)).$$

Therefore, we have the following bound:

$$W_2(q_0, p_0^\theta) \leq \sum_{k=0}^{N-2} \bar{\Lambda}_{k+1} \left(\mathbb{E}_{\bar{q}_{k+1}} W_2^2(q_{k|k+1, k+2}, T_{k+1}^\theta)\right)^{1/2} + \bar{\Lambda}_N \left(\mathbb{E}_{(x_N, p_N) \sim p_N \otimes \rho_N} W_2^2(\bar{q}_{N-1|N}, \bar{T}_N^\theta)\right)^{1/2}$$
$$+ \bar{\Lambda}_N W_2((q_N \otimes \rho_N)S_N, (p_N \otimes \rho_N)S_N) + \mathcal{O}(\Delta t).$$

Now, we recall that

$$T_{k+1}^\theta(\cdot|x_{k+1}, x_{k+2}) = \mathcal{N}(\mu_{k+1}^u(x_{k+1}, x_{k+2}), \sigma_{k+1}^4 \Delta t^2(1 - a_{k+1}^2)\Pi(x_{k+1})).$$

Let $(\mu_{k|k+1, k+2}, \Sigma_{k|k+1, k+2})$ be the conditional mean and covariance of $q_{k|k+1, k+2}(\cdot|x_{k+1}, x_{k+2})$. Then, from the relation between Gelbrich distance and 2-Wasserstein distance, with a nonnegative gap $\Delta_{k+1}(x_{k+1}, x_{k+2})$ independent of $\theta$, we have:

$$\mathbb{E}_{\bar{q}_{k+1}} W_2^2(q_{k|k+1, k+2}, T_{k+1}^\theta(\cdot|x_{k+1}, x_{k+2}))$$
$$= \mathbb{E}_{\bar{q}_{k+1}} \|\mu_{k|k+1, k+2}(x_{k+1}, x_{k+2}) - \mu_{k+1}^u(x_{k+1}, x_{k+2})\|^2$$
$$+ \mathbb{E}_{\bar{q}_{k+1}} \left[\mathsf{Tr}\left(\Sigma_{k|k+1, k+2}(x_{k+1}, x_{k+2})\right) + \mathsf{Tr}\left(\sigma_{k+1}^4 \Delta t^2(1 - a_{k+1}^2)\Pi(x_{k+1})\right)\right]$$
$$- 2\mathbb{E}_{\bar{q}_{k+1}} \left[\mathsf{Tr}\left(\left(\Sigma_{k|k+1, k+2}^{1/2}(x_{k+1}, x_{k+2})\sigma_{k+1}^4 \Delta t^2(1 - a_{k+1}^2)\Pi(x_{k+1})\Sigma_{k|k+1, k+2}^{1/2}(x_{k+1}, x_{k+2})\right)^{1/2}\right)\right]$$
$$+ \mathbb{E}_{\bar{q}_{k+1}} \Delta_{k+1}(x_{k+1}, x_{k+2})$$
$$= \mathbb{E}_{(x_k, x_{k+1}, x_{k+2})} \|x_k - \mu_{k+1}^u(x_{k+1}, x_{k+2})\|^2 + \mathbb{E}_{\bar{q}_{k+1}} \mathsf{Tr}\left(\sigma_{k+1}^4 \Delta t^2(1 - a_{k+1}^2)\Pi(x_{k+1})\right)$$
$$- 2\mathbb{E}_{\bar{q}_{k+1}} \left[\mathsf{Tr}\left(\left(\Sigma_{k|k+1, k+2}^{1/2}(x_{k+1}, x_{k+2})\sigma_{k+1}^4 \Delta t^2(1 - a_{k+1}^2)\Pi(x_{k+1})\Sigma_{k|k+1, k+2}^{1/2}(x_{k+1}, x_{k+2})\right)^{1/2}\right)\right]$$
$$+ \mathbb{E}_{\bar{q}_{k+1}} \Delta_{k+1}(x_{k+1}, x_{k+2}).$$

where the expectation $\mathbb{E}_{(x_k, x_{k+1}, x_{k+2})}$ is over the forward joint triple. We used the law of total variance for the last equality, by observing that $x_k = \mu_{k|k+1, k+2}(x_{k+1}, x_{k+2}) + \epsilon \sim q_k$ with $\mathbb{E}[\epsilon|x_{k+1}, x_{k+2}] = 0$ and $\mathrm{Cov}(\epsilon|x_{k+1}, x_{k+2}) = \Sigma_{k|k+1, k+2}$. Similar to overdamped case, $\Delta_{k+1}(x_{k+1}, x_{k+2}) \geq 0$ is the distance gap between Gelbrich distance and 2-Wasserstein distance (independent of $\theta$) so that it becomes zero if the true conditional $x_k|x_{k+1}, x_{k+2}$ follows the Gaussian distribution as

$$x_k|x_{k+1}, x_{k+2} \sim \mathcal{N}(\mu_{k|k+1, k+2}(x_{k+1}, x_{k+2}), \Sigma_{k|k+1, k+2}(x_{k+1}, x_{k+2}))$$

**Training loss (underdamped) from CWPM**    Because $\|x_k - \mu_{k+1}^u(x_{k+1}, x_{k+2})\|^2$ can be decomposed into

$$\|x_k - \mu_{k+1}^u(x_{k+1}, x_{k+2})\|^2 = \|\Pi(x_{k+1})(x_k - \mu_{k+1}^u(x_{k+1}, x_{k+2}))\|^2$$
$$+ \underbrace{\|(I - \Pi(x_{k+1}))(x_k - \mu_{k+1}^u(x_{k+1}, x_{k+2}))\|^2}_{\text{constant w.r.t } \theta}$$

where $\mu_{k+1}^u$ does not have $\theta$ dependency on normal (second) term. Therefore, by abusing notation to set $p_N = x_{N+1}$, the choice of training loss becomes

$$L_{\text{CWPM}}^{\text{under}}(\theta) = \mathbb{E}_{q(x_{0:N})}\mathbb{E}_{\rho_N(p_N|x_N)}\left[\sum_{k=0}^{N-1}\lambda(k)\|\Pi(x_{k+1})\left(x_k - \mu_{k+1}^u(x_{k+1}, x_{k+2})\right)\|^2\right]$$

$$= \mathbb{E}_{q(x_{0:N})}\mathbb{E}_{\rho_N(p_N|x_N)}\left[\sum_{k=0}^{N-1}\frac{\|\Pi(x_{k+1})\left(x_k - \mu_{k+1}^u(x_{k+1}, x_{k+2})\right)\|^2}{2\sigma_{k+1}^4\Delta t^2(1 - a_{k+1}^2)}\right]$$

with some training loss weight $\lambda(k)$. In our case, the training loss weight proportional to the inverse of variance can be chosen by $\lambda(k) = \frac{1}{2\sigma_{k+1}^4\Delta t^2(1-a_{k+1}^2)}$ with proportional constant $1/2$. And, notably, this leads to the exactly the same training loss provided in DT-ELBO, (Lemma C.2) without the requirement $x_k \in \Sigma$.

**Lemma D.2** (Sufficient condition for $\Lambda_k < \infty$ – Underdamped). *Let the one-step landing backward update of ULLA (subsection Backward-ULLA) be*

$$\bar{F}_{k+1}^\theta(x_+, x_{++}, \zeta) = x_+ - \sigma_{k+1}^2\Delta t\Pi(x_+)\left[a_{k+1}\tilde{p} + \sigma_{k+1}^2\Delta t\, b_{k+1}^\theta(x_+, \tilde{p})\right]$$
$$+ \sigma_{k+1}^2\Delta t\sqrt{1 - a_{k+1}^2}\Pi(x_+)\zeta + \sigma_{k+1}^2\Delta t\phi(x_+, \tilde{p})$$

*with pseudo-momentum*

$$\tilde{p}(x_+, x_{++}) := \Pi(x_+)\left(\frac{x_{++} - x_+}{\sigma_{k+2}^2\Delta t}\right) \in T_{x_+}\Sigma.$$

*Here $\phi(x_+, \tilde{p})$ is the normal term and $b_{k+1}^\theta(x_+, \tilde{p})$ is the drift term. We use the standing finite-grid bounds*

$$0 < \underline{\sigma} \le \sigma_j \le \overline{\sigma} < \infty, \qquad a_j = e^{-\gamma\sigma_j^2\Delta t}, \qquad 1 - a_j^2 = \mathcal{O}(\Delta t).$$

*Assume the following regularity:*

1. *(Regularity of constraint functions)    For every $r \ge 1$, there exist constants $c_{\phi,r}^0, c_{\phi,r}^1, c_{\phi,r}^2 < \infty$ such that, for any pair $(X, P)$ with finite $4r$-th moment in $P$,*

$$\mathbb{E}\|\phi(X, P)\|^{2r} \le c_{\phi,r}^0 + c_{\phi,r}^1\mathbb{E}\left(\|X\|^{2r} + \|P\|^{2r}\right) + c_{\phi,r}^2\mathbb{E}\|P\|^{4r}.$$

2. *(Regularity of function class)    There exist constants $L_g, C_b < \infty$, independent of $\theta$, such that*

$$\|b_{k+1}^\theta(x, p) - b_{k+1}^\theta(x', p')\| \le L_g\left(\|x - x'\| + \|p - p'\|\right), \qquad \|b_{k+1}^\theta(x, p)\| \le C_b\left(1 + \|x\| + \|p\|\right).$$

*Let $\bar{T}_{k+1}^\theta$ be the associated pair-kernel to*

$$\bar{G}_{k+1}^\theta(x_+, x_{++}, \zeta) := (\bar{F}_{k+1}^\theta(x_+, x_{++}, \zeta), x_+).$$

*For a pair law $\eta$ on $\mathbb{R}^{2d}$, define its augmented scaled pair moment by*

$$\mathcal{M}_{k+1}(\eta)^2 := \mathbb{E}_{(X_+, X_{++})\sim\eta}\left(\|X_+\|^2 + \|X_{++}\|^2 + \|\tilde{p}(X_+, X_{++})\|^2 + \|\tilde{p}(X_+, X_{++})\|^4\right).$$

*For the finite-horizon moment propagation below, we also write, for $r \ge 1$,*

$$\mathcal{A}_{k+1}^{(r)}(\eta) := \mathbb{E}_{(X_+, X_{++})\sim\eta}\left(\|X_+\|^{2r} + \|X_{++}\|^{2r} + \|\tilde{p}(X_+, X_{++})\|^{2r}\right).$$

*Then, for any probability measures $\mu, \nu$ on $\mathbb{R}^{2d}$ satisfying $\mathcal{M}_{k+1}(\mu) + \mathcal{M}_{k+1}(\nu) < \infty$, we have*

$$W_2(\mu\bar{T}_{k+1}^\theta, \nu\bar{T}_{k+1}^\theta) \le K_{k+1}W_2(\mu, \nu) + \bar{C}_{k+1}\Delta t\left(1 + \mathcal{M}_{k+1}(\mu) + \mathcal{M}_{k+1}(\nu)\right)$$

*for some constants $K_{k+1}, \bar{C}_{k+1} < \infty$ independent of $\theta$.*

*Also, if $\mu$ is given as*

$$\mu = \rho \bar{T}_i^\theta \bar{T}_{i-1}^\theta \ldots \bar{T}_j^\theta$$

*for $i \geq j$ and some source pair law $\rho$ satisfying, for every $r \geq 1$,*

$$\sup_{\theta \in \Theta} \mathcal{A}_i^{(r)}(\rho) < \infty,$$

*then the augmented scaled pair moment of $\mu$ is uniformly finite under $\theta \in \Theta$:*

$$\sup_{\theta \in \Theta} \mathcal{M}_{j-1}(\mu) < \infty.$$

*Moreover, assuming the initial data/momentum and terminal position/momentum priors have finite moments of all orders,*

$$\mathbb{E}\|X_0\|^{2r} + \mathbb{E}\|P_0\|^{2r} < \infty, \qquad \mathbb{E}_{X_N \sim p_N}\|X_N\|^{2r} + \mathbb{E}_{P_N \sim \rho_N(\cdot|X_N)}\|P_N\|^{2r} < \infty \quad \text{for every } r \geq 1,$$

*and assuming the forward collapsed ULLA drift and normal terms satisfy the same polynomial-growth bounds as above, we can guarantee the existence of $\Lambda_k > 0$ such that*

$$W_2(\bar{\sigma}_{k-1}, \bar{\sigma}_k) \leq \Lambda_k W_2(\bar{q}_{k-1}, \bar{q}_k \bar{T}_k^\theta) + \mathcal{O}(\Delta t)$$

*for $k \in \{1, \ldots, N-1\}$ and*

$$W_2(\bar{\sigma}_{N-1}, \bar{p}_0^\theta) \leq \Lambda_N W_2(\bar{q}_{N-1}, \bar{p}_{N-1}^\theta) + \mathcal{O}(\Delta t).$$

*Proof.* Throughout the proof, $C_1, C_2, C_3, \ldots$ denote finite positive constants independent of $\theta$ and $\Delta t$, and their values may change from line to line; $C_r$ may also depend on $r$. Let $Y = (X_+, X_{++})$ and $Y' = (X'_+, X'_{++})$ be an optimal coupling of $\mu$ and $\nu$, and use the same Gaussian noise $\zeta \sim \mathcal{N}(0, I_d)$ for the two transitions. Set

$$\tau_{k+1} := \sigma_{k+1}^2 \Delta t, \qquad a := a_{k+1},$$

and write

$$\Delta_+ := X_+ - X'_+, \qquad \tilde{p} := \tilde{p}(X_+, X_{++}), \qquad \tilde{p}' := \tilde{p}(X'_+, X'_{++}).$$

Also, we use notations

$$\Pi_+ := \Pi(X_+), \qquad \Pi'_+ := \Pi(X'_+), \qquad b := b_{k+1}^\theta(X_+, \tilde{p}), \qquad b' := b_{k+1}^\theta(X'_+, \tilde{p}'),$$

and

$$\phi := \phi(X_+, \tilde{p}), \qquad \phi' := \phi(X'_+, \tilde{p}').$$

The pair transition outputs are

$$\bar{G}_{k+1}^\theta(Y, \zeta) = (\bar{F}_{k+1}^\theta(Y, \zeta), X_+), \qquad \bar{G}_{k+1}^\theta(Y', \zeta) = (\bar{F}_{k+1}^\theta(Y', \zeta), X'_+).$$

Hence, by the synchronous coupling,

$$W_2(\mu \bar{T}_{k+1}^\theta, \nu \bar{T}_{k+1}^\theta) \leq \left(\mathbb{E}\|\bar{G}_{k+1}^\theta(Y, \zeta) - \bar{G}_{k+1}^\theta(Y', \zeta)\|^2\right)^{1/2}.$$

Since the second coordinate of $\bar{G}_{k+1}^\theta$ is $X_+$, we have

$$\left(\mathbb{E}\|\bar{G}_{k+1}^\theta(Y, \zeta) - \bar{G}_{k+1}^\theta(Y', \zeta)\|^2\right)^{1/2} \leq \left(\mathbb{E}\|\bar{F}_{k+1}^\theta(Y, \zeta) - \bar{F}_{k+1}^\theta(Y', \zeta)\|^2\right)^{1/2} + \left(\mathbb{E}\|\Delta_+\|^2\right)^{1/2}.$$

Now decompose the first-coordinate difference as

$$\bar{F}_{k+1}^\theta(Y, \zeta) - \bar{F}_{k+1}^\theta(Y', \zeta) = \Delta_+ + R_1 + R_2 + R_3 + R_4,$$

where

$$R_1 := -\tau_{k+1} a (\Pi_+ \tilde{p} - \Pi'_+ \tilde{p}'), \quad R_2 := -\tau_{k+1}^2 (\Pi_+ b - \Pi'_+ b'), \quad R_3 := \tau_{k+1}\sqrt{1-a^2}(\Pi_+ - \Pi'_+)\zeta, \quad R_4 := \tau_{k+1}(\phi - \phi').$$

Therefore, by Minkowski's inequality,

$$\left(\mathbb{E}\|\bar{G}_{k+1}^{\theta}(Y,\zeta)-\bar{G}_{k+1}^{\theta}(Y',\zeta)\|^2\right)^{1/2}\leq 2\left(\mathbb{E}\|\Delta_+\|^2\right)^{1/2}+\sum_{i=1}^{4}\left(\mathbb{E}\|R_i\|^2\right)^{1/2}.$$

Since $(Y,Y')$ is an optimal coupling,

$$\left(\mathbb{E}\|\Delta_+\|^2\right)^{1/2}\leq\left(\mathbb{E}\|Y-Y'\|^2\right)^{1/2}=W_2(\mu,\nu).$$

It remains to control the four remainder terms.

First, using $\|\Pi(x)\|\leq 1$ and $0<a\leq 1$,
$$\|R_1\|\leq\tau_{k+1}\left(\|\tilde{p}\|+\|\tilde{p}'\|\right).$$

Thus,

$$\left(\mathbb{E}\|R_1\|^2\right)^{1/2}\leq C_1\Delta t\left(\left(\mathbb{E}\|\tilde{p}\|^2\right)^{1/2}+\left(\mathbb{E}\|\tilde{p}'\|^2\right)^{1/2}\right).$$

Second, using $\|\Pi(x)\|\leq 1$ and the linear-growth part of Assumption 2,

$$\|R_2\|\leq\tau_{k+1}^2\left(\|b\|+\|b'\|\right)\leq C_2\Delta t^2\left(1+\|X_+\|+\|X_+'\|+\|\tilde{p}\|+\|\tilde{p}'\|\right).$$

Therefore,

$$\left(\mathbb{E}\|R_2\|^2\right)^{1/2}\leq C_2\Delta t^2\left(1+\left(\mathbb{E}\|X_+\|^2\right)^{1/2}+\left(\mathbb{E}\|X_+'\|^2\right)^{1/2}+\left(\mathbb{E}\|\tilde{p}\|^2\right)^{1/2}+\left(\mathbb{E}\|\tilde{p}'\|^2\right)^{1/2}\right).$$

Third, since $1-a^2=\mathcal{O}(\Delta t)$ and $\|\Pi_+-\Pi_+'\|\leq 2$,

$$\left(\mathbb{E}\|R_3\|^2\right)^{1/2}\leq C_3\Delta t\sqrt{1-a^2}\left(\mathbb{E}\|\zeta\|^2\right)^{1/2}\leq C_4\Delta t^{3/2}\leq C_4\Delta t,$$

where the last inequality uses $\Delta t\leq 1$.

Fourth, by Assumption 1 with $r=1$,

$$\begin{aligned}\left(\mathbb{E}\|R_4\|^2\right)^{1/2}&\leq\tau_{k+1}\left(\left(\mathbb{E}\|\phi\|^2\right)^{1/2}+\left(\mathbb{E}\|\phi'\|^2\right)^{1/2}\right)\\ &\leq C_5\Delta t\left(1+\left(\mathbb{E}\|X_+\|^2\right)^{1/2}+\left(\mathbb{E}\|X_+'\|^2\right)^{1/2}+\left(\mathbb{E}\|\tilde{p}\|^2\right)^{1/2}+\left(\mathbb{E}\|\tilde{p}'\|^2\right)^{1/2}\right)\\ &\quad+C_6\Delta t\left(\left(\mathbb{E}\|\tilde{p}\|^4\right)^{1/2}+\left(\mathbb{E}\|\tilde{p}'\|^4\right)^{1/2}\right)\\ &\leq C_7\Delta t\left(1+\mathcal{M}_{k+1}(\mu)+\mathcal{M}_{k+1}(\nu)\right),\end{aligned}$$

where the last inequality uses the definition of the augmented scaled pair moment. Combining these four bounds gives

$$W_2(\mu\bar{T}_{k+1}^{\theta},\nu\bar{T}_{k+1}^{\theta})\leq K_{k+1}W_2(\mu,\nu)+\bar{C}_{k+1}\Delta t\left(1+\mathcal{M}_{k+1}(\mu)+\mathcal{M}_{k+1}(\nu)\right),$$

for constants $K_{k+1},\bar{C}_{k+1}<\infty$ independent of $\theta$.

Next, we show that the augmented scaled pair moment remains finite under finitely many reverse pair-kernel applications, provided the source law has finite scaled moments of all orders. Let

$$Y^-=(X_-,X_+):=\bar{G}_{k+1}^{\theta}(Y,\zeta),\qquad X_-:=\bar{F}_{k+1}^{\theta}(Y,\zeta).$$

The scaled pseudo-momentum of the new pair is

$$\tilde{p}^-:=\Pi(X_-)\frac{X_+-X_-}{\sigma_{k+1}^2\Delta t}.$$

Since $\sigma_{k+1}$ is bounded below and $\|\Pi(X_-)\|\leq 1$,

$$\|\tilde{p}^-\|\leq C_8\frac{\|X_+-X_-\|}{\Delta t}.$$

By the definition of $X_-$,

$$X_+ - X_- = \tau_{k+1}\Pi_+ \left(a\tilde{p} + \tau_{k+1}b\right) - \tau_{k+1}\sqrt{1-a^2}\Pi_+\zeta - \tau_{k+1}\phi.$$

Therefore,

$$\|\tilde{p}^-\| \leq C_9 \left(\|\tilde{p}\| + \Delta t\|b\| + \sqrt{1-a^2}\|\zeta\| + \|\phi\|\right).$$

More generally, for every $r \geq 1$, using $(u_1 + \cdots + u_4)^{2r} \leq \tilde{C}_r \sum_{m=1}^{4} u_m^{2r}$, the linear growth of $b$, the polynomial-growth bound on $\phi$, and the finiteness of Gaussian moments, we obtain

$$\mathbb{E}\|\tilde{p}^-\|^{2r} \leq \tilde{C}_r \left(1 + \mathbb{E}\|X_+\|^{2r} + \mathbb{E}\|\tilde{p}\|^{2r} + \mathbb{E}\|\tilde{p}\|^{4r}\right).$$

Hence,

$$\mathbb{E}\|\tilde{p}^-\|^{2r} \leq \tilde{C}_r \left(1 + \mathcal{A}_{k+1}^{(2r)}(\eta)\right).$$

Similarly, the position update gives

$$\mathbb{E}\|X_-\|^{2r} \leq \tilde{C}_r \left(1 + \mathcal{A}_{k+1}^{(2r)}(\eta)\right).$$

Since the second coordinate of the new pair is $X_+$, we obtain the all-order finite-horizon moment recursion

$$\mathcal{A}_k^{(r)}(\eta\bar{T}_{k+1}^\theta) \leq C_r' \left(1 + \mathcal{A}_{k+1}^{(2r)}(\eta)\right).$$

Iterating this estimate over finitely many steps gives

$$\sup_{\theta\in\Theta} \mathcal{M}_{j-1}\left(\rho\bar{T}_i^\theta\bar{T}_{i-1}^\theta\cdots\bar{T}_j^\theta\right) < \infty,$$

whenever the source pair law $\rho$ satisfies

$$\sup_{\theta\in\Theta} \mathcal{A}_i^{(r)}(\rho) < \infty \qquad \text{for every } r \geq 1.$$

We now verify the required source moment bounds. For the forward pair laws $\bar{q}_\ell = \mathrm{Law}(X_\ell, X_{\ell+1})$, assume that the initial data and initial momentum have finite moments of all orders:

$$\mathbb{E}\|X_0\|^{2r} + \mathbb{E}\|P_0\|^{2r} < \infty \qquad \text{for every } r \geq 1.$$

Define the approximate forward momentum used in the collapsed ULLA update by

$$\widehat{p}_0^q := \Pi(X_0)P_0, \qquad \widehat{p}_\ell^q := \Pi(X_\ell)\frac{X_\ell - X_{\ell-1}}{\sigma_{\ell-1}^2\Delta t}, \quad \ell \geq 1.$$

The forward collapsed ULLA update can be written in the generic form

$$X_{\ell+1} = X_\ell + \sigma_\ell^2\Delta t\,\Pi(X_\ell)\left(a_\ell\widehat{p}_\ell^q + \sigma_\ell^2\Delta t\,b_\ell^{\mathrm{fwd}}(X_\ell,\widehat{p}_\ell^q)\right) + \sigma_\ell^2\Delta t\sqrt{1-a_\ell^2}\,\Pi(X_\ell)\zeta_\ell + \sigma_\ell^2\Delta t\,\phi_\ell^{\mathrm{fwd}}(X_\ell,\widehat{p}_\ell^q),$$

where the signs of the deterministic drift and landing terms are absorbed into $b_\ell^{\mathrm{fwd}}$ and $\phi_\ell^{\mathrm{fwd}}$. We assume that the forward drift and normal terms satisfy the same growth bounds uniformly over the finite grid: for every $r \geq 1$,

$$\|b_\ell^{\mathrm{fwd}}(x,p)\| \leq C_{\mathrm{f},1}\left(1 + \|x\| + \|p\|\right),$$

and

$$\mathbb{E}\|\phi_\ell^{\mathrm{fwd}}(X,P)\|^{2r} \leq C_{\mathrm{f},r}\left(1 + \mathbb{E}\|X\|^{2r} + \mathbb{E}\|P\|^{2r} + \mathbb{E}\|P\|^{4r}\right).$$

The scaled pseudo-momentum appearing in the pair law $\bar{q}_\ell$ is

$$\tilde{p}_\ell^q := \Pi(X_\ell)\frac{X_{\ell+1} - X_\ell}{\sigma_{\ell+1}^2\Delta t}.$$

Substituting the forward update into this definition and using $\|\Pi(x)\| \leq 1$ gives

$$\|\tilde{p}_\ell^q\| \leq C_{\sigma,1}\left(\|\widehat{p}_\ell^q\| + \Delta t\|b_\ell^{\text{fwd}}(X_\ell, \widehat{p}_\ell^q)\| + \sqrt{1-a_\ell^2}\|\zeta_\ell\| + \|\phi_\ell^{\text{fwd}}(X_\ell, \widehat{p}_\ell^q)\|\right),$$

where $C_{\sigma,1} := \sup_\ell \sigma_\ell^2/\sigma_{\ell+1}^2 < \infty$. Therefore, for every $r \geq 1$, the same polynomial-growth argument gives

$$\mathbb{E}\|\tilde{p}_\ell^q\|^{2r} \leq C_r\left(1 + \mathbb{E}\|X_\ell\|^{2r} + \mathbb{E}\|\widehat{p}_\ell^q\|^{2r} + \mathbb{E}\|\widehat{p}_\ell^q\|^{4r}\right).$$

For $\ell = 0$, $\widehat{p}_0^q = \Pi(X_0)P_0$ has finite moments of all orders. For $\ell \geq 1$, using the previous forward update for $X_\ell - X_{\ell-1}$ gives, for every $r \geq 1$,

$$\mathbb{E}\|\widehat{p}_\ell^q\|^{2r} \leq C_r\left(1 + \mathbb{E}\|X_{\ell-1}\|^{2r} + \mathbb{E}\|\widehat{p}_{\ell-1}^q\|^{2r} + \mathbb{E}\|\widehat{p}_{\ell-1}^q\|^{4r}\right).$$

Similarly, the forward position update gives

$$\mathbb{E}\|X_{\ell+1}\|^{2r} \leq C_r\left(1 + \mathbb{E}\|X_\ell\|^{2r} + \mathbb{E}\|\widehat{p}_\ell^q\|^{2r} + \mathbb{E}\|\widehat{p}_\ell^q\|^{4r}\right).$$

Because the grid is finite and the initial data/momentum have finite moments of all orders, these recursions imply by finite-horizon induction that

$$\sup_{0 \leq \ell \leq N} \mathbb{E}\|X_\ell\|^{2r} + \sup_{0 \leq \ell \leq N-1} \mathbb{E}\|\widehat{p}_\ell^q\|^{2r} < \infty \qquad \text{for every } r \geq 1.$$

In particular, the augmented scaled pair moments of the forward source laws are finite:

$$\sup_{0 \leq \ell \leq N-1} \mathcal{M}_\ell(\bar{q}_\ell) < \infty.$$

Similarly, for the terminal source law, assume

$$\mathbb{E}_{X_N \sim p_N}\|X_N\|^{2r} + \mathbb{E}_{P_N \sim \rho_N(\cdot|X_N)}\|P_N\|^{2r} < \infty \qquad \text{for every } r \geq 1.$$

In the collapsed notation, set

$$X_{N+1} := X_N + \sigma_{N+1}^2 \Delta t\, P_N,$$

where $\sigma_{N+1}$ is any bounded terminal extension of the schedule, e.g. $\sigma_{N+1} = \sigma_N$. Then

$$\Pi(X_N)\frac{X_{N+1} - X_N}{\sigma_{N+1}^2 \Delta t} = \Pi(X_N)P_N,$$

and hence the base terminal pair $(X_N, X_{N+1})$ has finite scaled moments of all orders. The terminal pair law $\bar{p}_{N-1}^\theta$ is obtained from this base pair by one application of the backward pair-kernel $\bar{T}_N^\theta$. The all-order finite-horizon moment recursion above therefore yields

$$\sup_{\theta \in \Theta} \mathcal{M}_{N-1}(\bar{p}_{N-1}^\theta) < \infty.$$

Finally, define

$$\bar{T}_{a:b}^\theta := \bar{T}_a^\theta \bar{T}_{a-1}^\theta \cdots \bar{T}_b^\theta \qquad (a \geq b),$$

and interpret $\bar{T}_{a:b}^\theta$ as the identity kernel when $a < b$. For $k = 1, \ldots, N-1$,

$$\bar{\sigma}_{k-1} = \bar{q}_{k-1}\bar{T}_{k-1:1}^\theta, \qquad \bar{\sigma}_k = (\bar{q}_k\bar{T}_k^\theta)\bar{T}_{k-1:1}^\theta.$$

Applying the one-step stability estimate successively to $\bar{T}_{k-1}^\theta, \ldots, \bar{T}_1^\theta$, and using the finite-horizon augmented moment bounds above, gives

$$W_2(\bar{\sigma}_{k-1}, \bar{\sigma}_k) \leq \Lambda_k W_2(\bar{q}_{k-1}, \bar{q}_k\bar{T}_k^\theta) + \mathcal{O}(\Delta t),$$

for some finite $\Lambda_k$ independent of $\theta$.

The terminal estimate follows in the same way. Since

$$\bar{\sigma}_{N-1} = \bar{q}_{N-1}\bar{T}_{N-1:1}^\theta, \qquad \bar{p}_0^\theta = \bar{p}_{N-1}^\theta \bar{T}_{N-1:1}^\theta,$$

repeated application of the same one-step estimate gives

$$W_2(\bar{\sigma}_{N-1}, \bar{p}_0^\theta) \leq \Lambda_N W_2(\bar{q}_{N-1}, \bar{p}_{N-1}^\theta) + \mathcal{O}(\Delta t).$$

This completes the proof. $\qquad\qquad\qquad\qquad\qquad\qquad\qquad\qquad\qquad\qquad\qquad\qquad\qquad\qquad\square$

# E. Experiment Settings and Supplementary Results

**Settings.** All experiments were implemented in Python using the PyTorch framework (Paszke et al., 2019) and run in a Linux (Ubuntu) environment. The computational hardware was tailored to the specific experimental group. We utilized an NVIDIA L40S GPU with 48GB of VRAM for the Earth and climate science datasets, and an NVIDIA H100 GPU with 80GB of VRAM for the 3D mesh data experiments. All other tasks, including the $SO(10)$ manifold, Alanine dipeptide, and the 7-DOF robot arm, were conducted on an NVIDIA H200 GPU with 141GB of VRAM.

## E.1. Description of Baseline Algorithms

**Riemannian Flow Matching (RFM).** RFM (Chen & Lipman, 2024) is a framework for training Continuous Normalizing Flows (CNF) (Chen et al., 2018) on Riemannian manifold by regressing a vector field $v_t$ to a conditional target vector field $u_t(x|x_1)$ for $t \in [0, 1]$ defined via a user-specified premetric $d(\cdot, \cdot)$ (e.g., geodesics, spectral distances). The model minimizes the Riemannian Conditional Flow Matching objective given as :

$$\mathcal{L}_{\text{RCFM}} = \mathbb{E}_{t \sim \mathcal{U}(0,1), x_1 \sim q_{\text{data}}, x_0 \sim p_{\text{prior}}} \left[ \|v_t(x_t) - u_t(x_t \mid x_1)\|_g^2 \right]$$

where $x_t$ is as conditional flow sample interpolation between prior samples $x_0$ and the data point $x_1$, and $\|\cdot\|_g$ is the norm defined in the corresponding Riemannian manifold.

The computational requirements for $x_t$ may depend on the manifold's geometry. On simple manifold (e.g., spheres, tori), the geodesic distance can be used as the premetric, allowing $x_t$ to be computed in closed form via the exponential map, thus making the algorithm simulation-free. In contrast, on general geometries (e.g., triangular meshes) where exact geodesics are intractable, spectral distances such as the biharmonic distance are employed as the premetric. In this case, computing $x_t$ requires solving an ODE during the training process.

**Remark 7** (Implementation details on RFM). For RFM, we used the default configuration from the official code from authors.[1] For Earth & Climate datasets, training iterations were reduced to $1/10$, whereas Mesh data experiments were conducted using the unaltered default configuration.

**Riemannian Denoising Diffusion Probabilistic Models (RDDPM).** RDDPM (Liu et al., 2026) is a constrained diffusion model framework that adapts Denoising Diffusion Probabilistic Models (DDPMs) (Ho et al., 2020) to Riemannian manifold $\Sigma := \{x \in \mathbb{R}^d \mid h(x) = 0\}$ setup by incorporating a Newton's method projection step into the diffusion process.

The method constructs forward and backward Markov chains that alternate between diffusion steps along tangential direction of $\Sigma$ and projecting the resulting sample back onto $\Sigma$ via Newton's method. While this guarantees feasibility at every step, the iterative nature of the projection leads to higher computational costs and potentially result in forward trajectory resampling due to projection failures. We remark that the projected version of OLLA (OLLA-P) corresponds to RDDPM under the equality-only scenario.

**Euclidean Forward with Backward Variants.** These baselines employ a standard unconstrained Euclidean diffusion process for the forward process, and distinguish themselves by the mechanism used to enforce constraints $h(x) = 0, g(x) \leq 0$ during the backward process. In the forward process, it follows the update rule below:

$$x_{k+1} = x_k - \frac{\sigma_k^2 \Delta t}{2} \nabla f(x_k) + \sigma_k \sqrt{\Delta t} \zeta_k$$

with corresponding training loss

$$\mathcal{L}_{\text{Euclidean}}^{\text{over}}(\theta) = \mathbb{E}_{q(x_{0:N})} \left[ \sum_{k=0}^{N-1} \frac{\|x_k - \mu_{k+1}^o(x_{k+1})\|^2}{2\sigma_{k+1}^2 \Delta t} \right]$$

and $\mu_{k+1}^o := x_{k+1} + \frac{\sigma_{k+1}^2 \Delta t}{2} \left[ \nabla f(x_{k+1}) + s_\theta^{k+1}(x_{k+1}) \right]$.

1. **Euclidean**: This method performs sampling using the standard Euclidean backward without any constraint enforcement. The backward update rule is given as:

$$x_k = x_{k+1} + \frac{\sigma_{k+1}^2 \Delta t}{2} \left[ \nabla f(x_{k+1}) + s_{k+1}^\theta(x_{k+1}) \right] + \sigma_{k+1} \sqrt{\Delta t} \zeta_{k+1}$$

---

[1] https://github.com/facebookresearch/riemannian-fm

This approach offers no guarantee that the generated samples lie on $\Sigma$.

2. **Projected**: This variant strictly enforces equality constraints by projecting the sample onto $\Sigma$ immediately after each Euclidean backward step. Let $\tilde{x}_k$ be the proposal from the Euclidean backward step. Then, the final state is obtained via $x_k = \mathcal{P}_\Sigma(\tilde{x}_k)$, where $\mathcal{P}_\Sigma$ finds the root of $h(y) = 0, g(y) \leq 0$ close to $\tilde{x}_k$ using the interior point method (Wächter & Biegler, 2006; Christopher et al., 2024). We remark that our implementation uses log-barrier for $g(x) < 0$ and quadratic penalty for $g(x) \geq 0$.

3. **Lagrangian**: This method formulates the sampling step as a constrained optimization problem using the Augmented Lagrangian Method (ALM). At each timestep, the proposal $\tilde{x}_k$ is refined by minimizing an augmented Lagrangian objective:

$$\mathcal{L}(x, \lambda, \mu) = \lambda^T h(x) + \frac{\rho}{2}\|h(x)\|^2 + \frac{1}{2\rho}\left(\|\text{ReLU}(\mu + \rho g(x))\|^2 - \|\mu\|^2\right)$$

The inequality term follows the Powell-Hestenes-Rockafellar (PHR) formulation. This specific form is derived by introducing a non-negative slack variable $s \geq 0$ to convert the inequality constraint $g(x) \leq 0$ into an equality $g(x) + s = 0$. By constructing the standard augmented Lagrangian for this equality and analytically minimizing it with respect to $s$, the slack variable is eliminated, resulting in the closed-form $\text{ReLU}(\mu + \rho g(x))$ term. This ensures that penalties are applied correctly only when constraints are violated or multipliers are active. The multipliers $\lambda$ and $\mu$ are updated iteratively via dual ascent. We note that this approach is also introduced in Liang et al. (2025).

4. **Guided**: This approach utilizes constraint guidance during sampling. The standard drift term of the backward process is modified by adding a guidance term derived from the gradient of a constraint violation energy potential. This potential is defined as $V(x) = \frac{1}{2}\|h(x)\|^2 + \frac{1}{2}\|\text{ReLU}(g(x))\|^2$, where the first term penalizes deviations from equality constraints and the second term penalizes violations of inequality constraints. Consequently, the backward update rule naturally incorporates a gradient descent step on this potential, which steers the generated trajectory towards the feasible set $\Sigma$ by actively minimizing the constraint violation at each step.

### E.2. Experiment Settings and Descriptions

**Earth and Climate Science Datasets** $S^2$. This benchmark (National Geophysical Data Center / World Data Service (NGDC/WDS), 2026b;a; Mathieu & Nickel, 2020; Brakenridge, 2017; NASA Earth Observing System Data and Information System (EOSDIS), 2020) evaluates the model's ability to learn geographical distributions on the Earth's surface, which is modeled as the 2-sphere, $S^2$. The datasets represent the locations of phenomena such as volcanoes, earthquakes, floods, and fires.

*Mathematical formulation.* A sample $x$ represents a point in 3D Euclidean space lying on the surface of a unit sphere. Thus, $x \in \mathbb{R}^d$ with $d = 3$. The manifold is defined by a single, simple equality constraint $h(x) = \|x\|_2 - 1 = 0$.

*Prior distribution.* As this is a compact manifold, the prior distribution $p_N$ is set to be the uniform distribution over the surface of the sphere $S^2$.

**3D Mesh Data on a Learned Manifolds.** The objective is to learn a probability distribution over the surface of a complex 3D shape, such as the Stanford Bunny (Turk & Levoy, 1994) and Spot the Cow (Crane et al., 2013). The manifold is implicitly defined as the zero-level set of a Signed Distance Function (SDF) that is itself represented by a pre-trained neural network $h_{NN}(x)$ as performed in Rozen et al. (2021); Gropp et al. (2020).

*Mathematical Formulation.* A sample $x$ represents a point in 3D Euclidean space, thus $x \in \mathbb{R}^d$ with $d = 3$. The manifold is defined by a single equality constraint requiring any valid point to lie on the zero-level set of $h_{NN}(x) = 0$.

*Prior distribution.* The prior distribution $p_N$ is chosen to be uniform distribution over the learned manifold surface $\Sigma$ due to its compactness.

**High-Dimensional Special Orthogonal Group** ($SO(10)$). This experiment tests the model's ability to learn a multimodal distribution on the high-dimensional Lie group $SO(10)$. This is a challenging task due to the high dimensionality and non-trivial geometric structure of the manifold.

*Mathematical Formulation.* A sample is a $10 \times 10$ matrix, which is vectorized into $x \in \mathbb{R}^{100}$. The constraints enforce the defining properties of a special orthogonal matrix. For the equality constraints, we impose

$$h_{ij}(X) = (X^T X - I)_{ij} = 0 \quad \text{for } 1 \leq i \leq j \leq 10$$

and the determinant condition $\det(X) = 1$ is handled by via rejection when it is violated.

*Prior distribution.* Similarly, the manifold is compact and we choose uniform distribution over $SO(10)$ as our prior distribution.

**Alanine Dipeptide** This task involves generating valid 3D conformations of Alanine dipeptide, a model system in biophysics. The goal is to learn the distribution of structures subject to constraints on specific internal coordinates, including a mixed equality and inequality setup. Following the same approach in Liu et al. (2026), we generated the dataset by running a 1ns constrained molecular dynamics simulation of alanine dipeptide in water using GROMACS (Abraham et al., 2015) with a 1 fs timestep. A harmonic bias was applied through the COLVARS module (Fiorin et al., 2013), where the chosen collective variable was dihedral angle $\phi$. The harmonic restraint was centered at $\phi = -70°$ with a force constant 5.0. Other simulation settings follow closely those reported in Lelièvre et al. (2024). In total, $10^4$ configurations were collected by saving a snapshot every 100 simulation steps. Hydrogen atoms were removed, leaving the coordinates of the 10 heavy atoms for further analysis.

*Mathematical Formulation.* The state $x$ consists of the 3D coordinates of the 10 non-hydrogen atoms, so $x \in \mathbb{R}^{30}$. The constraints are placed on two of the molecule's primary dihedral angles, $\phi$ and $\psi$. For the equality constraints, the dihedral angle $\phi$ is fixed to a specific value:

$$h(x) = \phi(x) - (-70°)_{\text{rad}} = 0$$

and we impose an inequality constraint so that another adjacent dihedral angle $\psi$ is constrained to lie within the range $[130°, 170°]$. This is formulated as a single inequality:

$$g(x) = \max\left\{\psi(x) - 170°_{\text{rad}}, 130°_{\text{rad}} - \psi(x)\right\} \leq 0.$$

*Prior distribution.* Instead of introducing a potential-based drift term to induce a specific unimodal prior as in Liu et al. (2026), we employ an empirical prior strategy. We first generate a large set of forward trajectories using the corresponding constrained dynamics (OLLA/ULLA) by running them to approximate the terminal prior distribution $q_N$ on the feasible set. The terminal states $x_N$ of these trajectories are collected, and the backward sampling process is initiated by drawing starting points uniformly from this pre-computed set, serving as a discrete approximation of the prior. Furthermore, to ensure the generated conformations respect physical symmetries, the score network for this task is designed to be $SE(3)$-invariant as proposed in Liu et al. (2026).

**7-DOF Robot Arm Trajectory** This experiment focuses on learning a complex, bimodal distribution of trajectories for a 7-DOF Franka Emika Panda robot arm. The model is trained on a dataset of 400 valid paths (200 for S-shaped, 200 for reverse S-shaped paths) generated by the Rapidly-exploring Random Tree (RRT) algorithm. The primary task is to generate trajectories that trace both S-shaped and reverse S-shaped paths between fixed start and end points. Throughout the motion, the generated trajectories must satisfy several critical constraints: the robot arm must navigate around two spherical obstacles, and its end-effector must maintain a constant height of $z = z_{\text{target}} = 0.1$.

*Mathematical Formulation.* The fundamental state of the robot arm is its configuration in joint space, represented by a vector of 7 joint angles, $\theta \in \mathbb{R}^7$. A trajectory is a time-discretized sequence of these configurations, $(\theta_l)_{l=1}^{L}$. To avoid the periodicity issue of raw angles, which poses challenges for neural networks, we represent each joint angle $\theta_{l,j}$ as a 2D vector on the unit circle $(\cos(\theta_{l,j}), \sin(\theta_{l,j}))$. Consequently, the state at a single time step $l$ is a vector $x_l \in \mathbb{R}^{14}$. The full trajectory is flattened into a single vector $x = [x_1, \ldots, x_L] \in \mathbb{R}^d$. For a trajectory with $L \in \{10, 20, 30, 40\}$ as in our setup, the ambient space dimension is $d \in \{140, 280, 420, 560\}$.

The constraints on the robot's behavior, such as end-effector position and obstacle avoidance, are defined in 3D Cartesian space. We bridge the joint space representation and the Cartesian space constraints using the forward kinematics function, FK : $\mathbb{R}^7 \to \mathbb{R}^{3 \times K}$, which maps a set of joint angles $\theta_l$ to the 3D positions of the $K = 7$ links of the robot arm. To handle the large number of resulting constraints efficiently, we employ a "summation trick" to combine multiple constraint violations into a single function for both equalities and inequalities. In particular, multiple geometric and kinematic conditions are aggregated into a single sum-of-squares function:

$$h(x) = \sum_{i=1}^{m} h_i(x)^2 = 0.$$

The individual components $h_i(x)$ enforce: (1) the validity of the joint representation, $h_{\text{rep}}(x_{l,j}) = \cos^2(\theta_{l,j}) + \sin^2(\theta_{l,j}) -$

$1 = 0$, for each joint $j$ at each time step $l$, (2) fixed start and end points for the trajectory

$$h_{\text{end}}(x_L) = \|\text{FK}(\theta_L)_{\text{end-effector}} - p_{\text{end}}\|^2 = 0, \quad h_{\text{start}}(x_1) = \|\text{FK}(\theta_1)_{\text{end-effector}} - p_{\text{start}}\|^2 = 0$$

with $p_{\text{start}}$ and $p_{\text{end}}$ being the target start and end positions, and (3) a fixed $z$-height for the end effector throughout the trajectory, $h_z(x_l) = [\text{FK}(\theta_l)_{\text{end-effector}}]_z - z_{\text{target}} = 0$.

For the inequality constraint, the robot arm must avoid two spherical obstacles. For each relevant robot link $k \in [K]$ and obstacle $o \in \{1, 2\} := N_{\text{obs}}$, the distance between them must exceed a safety margin. These conditions are combined into a single function by summing the rectified violations:

$$g(x) = \sum_{l=1}^{L} \sum_{k=1}^{K} \sum_{o=1}^{N_{\text{obs}}} \text{ReLU}\left((r_{\text{obs},o} + r_{\text{safety}}) - \|\text{FK}(\theta_l)_{\text{link},k} - p_{\text{obs},o}\|\right) \leq 0.$$

with $r_{\text{obs},o}, p_{\text{obs},o}$ being the radius and position of obstacles. This function is non-positive if and only if all links maintain the required minimum distance $r_{\text{safety}}$ from all obstacles throughout the entire trajectory.

*Prior distribution.* Similar to the Alanine Dipeptide task, we employ an empirical prior strategy. We first generate a large set of forward trajectories using the corresponding constrained dynamics (OLLA/ULLA) by running them to approximate the target prior distribution $q_N$ on the feasible set. The terminal states $x_N$ of these trajectories are collected, and the backward sampling process is initiated by drawing starting points uniformly from this pre-computed set, serving as a discrete approximation of the prior.

*Table 9.* **Summary of constrained feasible set dimensions and constraint specifications.** Below table represent the ambient dimension $d$, the intrinsic manifold dimension, and the number of equality ($m$) and inequality ($l$) constraints. For the Robot Arm task, $L$ denotes the number of time steps (e.g., $L \in \{10, \ldots, 40\}$), and the constraint counts $m$ and $l$ are reported before applying the summation trick.

| Dataset / Task | Ambient Dim. ($d$) | Intrinsic Dim. | Equality ($m$) | Inequality ($l$) |
|---|---|---|---|---|
| Earth & Climate ($S^2$) | 3 | 2 | 1 | 0 |
| 3D Mesh (Bunny / Spot) | 3 | 2 | 1 | 0 |
| Lie Group $SO(10)$ | 100 | 45 | 55 | 0 |
| Alanine Dipeptide | 30 | 29 | 1 | 2 |
| 7-DOF Robot Arm | $14L$ | $7L$ | $8L + 2$ | $14L$ |

*Table 10.* Detailed hyperparameters for all datasets, specified per algorithm. Here, $l_f$ denotes the frequency of forward trajectory generation (once every $l_f$ epochs), $B$ is the batch size, $N_{\text{hidden}}$ represents the hidden dimension of the MLP, and $N_{\text{layer}}$ is the number of layers in the MLP. For OLLA and ULLA, terminal projection is applied in all tasks except the 7-DOF robot arm. The symbol $\triangle$ in the $\alpha$ column denotes implicit landing with $\alpha = 1/(\sigma_k^2 \Delta t)$.

| Dataset | Algorithm | $\gamma$ | $\sigma_{\min}$ | $\sigma_{\max}$ | $N$ | $T$ | $l_f$ | $N_{\text{epoch}}$ | $B$ | $N_{\text{hidden}}$ | $N_{\text{layer}}$ | $\alpha$ | $\epsilon$ |
|---|---|---|---|---|---|---|---|---|---|---|---|---|---|
| Volcano | OLLA | - | 0.01 | 1.0 | 100 | 4.0 | 1 | 20000 | 128 | 512 | 5 | $\triangle$ | - |
| | ULLA | 5 | 0.1 | 2.0 | 50 | 2.0 | 1 | 20000 | 128 | 512 | 5 | $\triangle$ | - |
| | ULLA-P | 5 | 0.1 | 2.0 | 100 | 2.0 | 1 | 20000 | 128 | 512 | 5 | - | - |
| Earthquake | OLLA | - | 0.01 | 1.0 | 100 | 4.0 | 1 | 20000 | 512 | 512 | 5 | $\triangle$ | - |
| | ULLA | 5 | 0.1 | 2.0 | 50 | 2.0 | 1 | 20000 | 512 | 512 | 5 | $\triangle$ | - |
| | ULLA-P | 5 | 0.1 | 2.0 | 100 | 2.0 | 1 | 20000 | 512 | 512 | 5 | - | - |
| Flood | OLLA | - | 0.01 | 1.0 | 100 | 4.0 | 1 | 20000 | 512 | 512 | 5 | $\triangle$ | - |
| | ULLA | 5 | 0.1 | 2.0 | 50 | 2.0 | 1 | 20000 | 512 | 512 | 5 | $\triangle$ | - |
| | ULLA-P | 5 | 0.1 | 2.0 | 100 | 2.0 | 1 | 20000 | 512 | 512 | 5 | - | - |
| Fire | OLLA | - | 0.01 | 1.0 | 100 | 4.0 | 1 | 20000 | 512 | 512 | 5 | $\triangle$ | - |
| | ULLA | 5 | 0.1 | 2.0 | 50 | 2.0 | 1 | 20000 | 512 | 512 | 5 | $\triangle$ | - |
| | ULLA-P | 5 | 0.1 | 2.0 | 100 | 2.0 | 1 | 20000 | 512 | 512 | 5 | - | - |
| Bunny ($k = 50$) | OLLA | - | 0.07 | 0.07 | 100 | 8.0 | 100 | 2000 | 2048 | 256 | 5 | $\triangle$ | - |
| | ULLA | 20 | 0.2 | 0.6 | 30 | 3.0 | 100 | 2000 | 2048 | 256 | 5 | 25 | - |
| | ULLA-P | 20 | 0.2 | 0.6 | 50 | 3.0 | 100 | 2000 | 2048 | 256 | 5 | - | - |
| Bunny ($k = 100$) | OLLA | - | 0.07 | 0.07 | 100 | 5.0 | 100 | 2000 | 2048 | 256 | 5 | $\triangle$ | - |
| | ULLA | 20 | 0.2 | 0.6 | 30 | 3.0 | 100 | 2000 | 2048 | 256 | 5 | 25 | - |
| | ULLA-P | 20 | 0.2 | 0.6 | 50 | 3.0 | 100 | 2000 | 2048 | 256 | 5 | - | - |
| Spot ($k = 50$) | OLLA | - | 0.1 | 0.1 | 100 | 5.0 | 100 | 2000 | 2048 | 256 | 5 | $\triangle$ | - |
| | ULLA | 20 | 0.2 | 0.5 | 30 | 3.0 | 100 | 2000 | 2048 | 256 | 5 | 25 | - |
| | ULLA-P | 20 | 0.2 | 0.5 | 50 | 3.0 | 100 | 2000 | 2048 | 256 | 5 | - | - |
| Spot ($k = 100$) | OLLA | - | 0.1 | 0.1 | 100 | 3.0 | 100 | 2000 | 2048 | 256 | 5 | $\triangle$ | - |
| | ULLA | 20 | 0.2 | 0.5 | 30 | 3.0 | 100 | 2000 | 2048 | 256 | 5 | 25 | - |
| | ULLA-P | 20 | 0.2 | 0.5 | 50 | 3.0 | 100 | 2000 | 2048 | 256 | 5 | - | - |
| SO(10) ($m = 3$) | OLLA | - | 0.2 | 2.0 | 100 | 1.0 | 5 | 2000 | 128 | 1024 | 5 | 50 | - |
| | ULLA | 10 | 0.2 | 2.0 | 50 | 2.5 | 5 | 2000 | 128 | 1024 | 5 | 15 | - |
| | ULLA-P | 10 | 0.2 | 2.0 | 50 | 2.5 | 5 | 2000 | 128 | 1024 | 5 | - | - |
| SO(10) ($m = 5$) | OLLA | - | 0.2 | 2.0 | 100 | 1.0 | 5 | 2000 | 512 | 1024 | 5 | 50 | - |
| | ULLA | 10 | 0.2 | 2.0 | 50 | 2.5 | 5 | 2000 | 512 | 1024 | 5 | 15 | - |
| | ULLA-P | 5 | 0.2 | 2.0 | 50 | 2.5 | 5 | 2000 | 512 | 1024 | 5 | - | - |
| Alanine dipeptide | ULLA | 10 | 0.5 | 2.0 | 100 | 0.2 | 10 | 2000 | 1024 | 1024 | 5 | 50 | 0.05 |
| 7-DOF robot arm | ULLA | 1 | 0.1 | 2.0 | 100 | 0.2 | 50 | 10000 | 160 | 1024 | 5 | 200 | 0.001 |

### E.3. Generated Samples from Tasks

In this subsection, we demonstrate the effectiveness of our proposed methods by comparing the generated samples in various tasks to the baseline RDDPM (Liu et al., 2026).

**Earth and Climate Science Datasets $S^2$ –Volcano.**

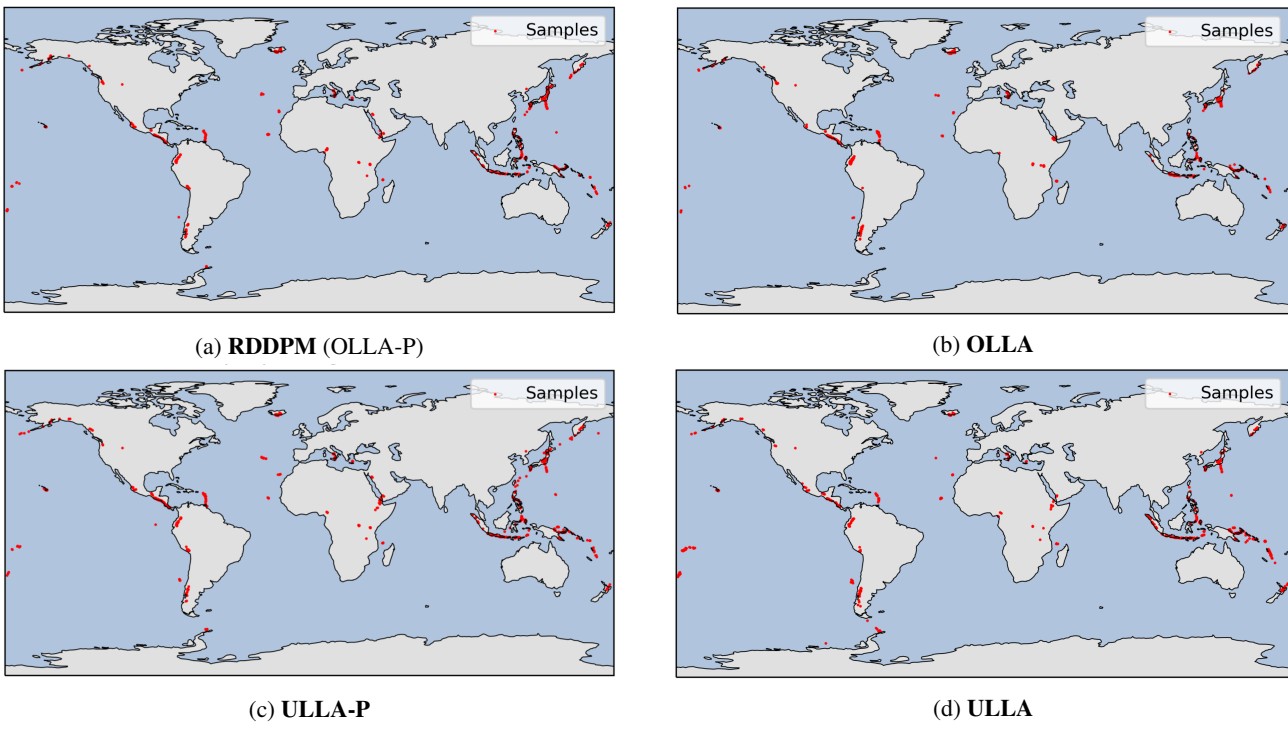

(a) **RDDPM** (OLLA-P)

(b) **OLLA**

(c) **ULLA-P**

(d) **ULLA**

*Figure 6.* Comparison of generated distributions across different algorithms-Volcano dataset

**3D Mesh data on learned manifold – Spot the Cow** $(k = 100)$**.**

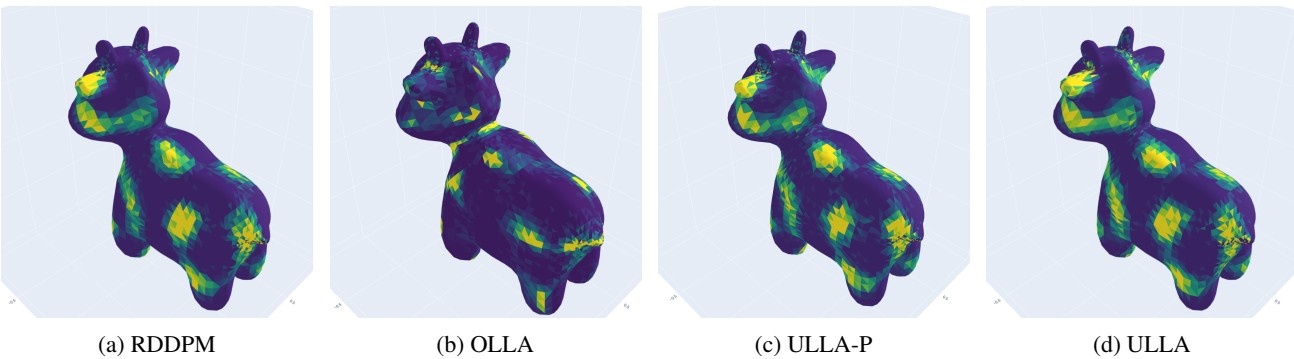

(a) RDDPM

(b) OLLA

(c) ULLA-P

(d) ULLA

*Figure 7.* Comparison of generated distributions across different algorithms - Spot the Cow $k = 100$

**SO(10) manifold with $m = 3$.**

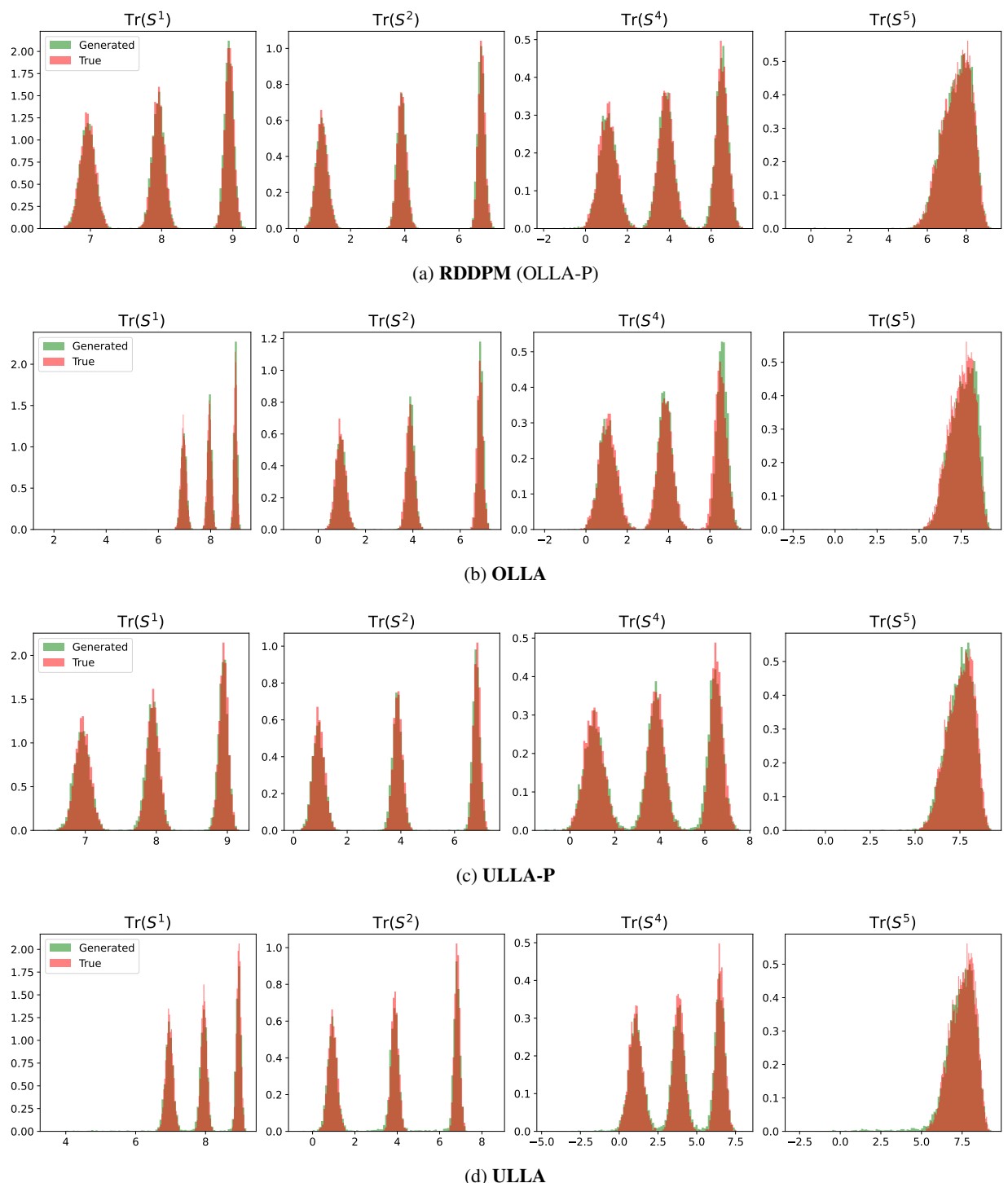

(a) **RDDPM** (OLLA-P)

(b) **OLLA**

(c) **ULLA-P**

(d) **ULLA**

*Figure 8.* Comparison of generated distributions across different algorithms- $SO(10)$ with $m = 3$

## E.4. Supplementary Results: Effect of Hyperparameters $\alpha$ and $\epsilon$

Table 11 summarizes how the landing rate $\alpha$ and boundary repulsion rate $\epsilon$ affect in the dipeptide experiment. Increasing $\alpha$ strengthens contraction toward the equality manifold and improves the JSD up to a moderate range, but overly large $\alpha$ can introduce discretization error and numerical stiffness. For the inequality parameter, too small $\epsilon$ leads to boundary stickiness, while too large $\epsilon$ pushes samples too aggressively into the feasible interior and distorts the generated distribution.

**Tuning of $\alpha$ and $\epsilon$.** The continuous-time landing property in Lemma 1 suggests a simple discretized tuning rule. For equality constraints, the discretized residual approximately follows

$$\mathbb{E}[h(X_{k+1})] = (1 - \alpha\sigma_k^2\Delta t)\mathbb{E}[h(X_k)] + O(\Delta t).$$

Thus $0 < \alpha\sigma_k^2\Delta t < 2$ gives stable contraction, while $0 < \alpha\sigma_k^2\Delta t \leq 1$ gives monotone decay. We initialize $\alpha$ so that $\alpha\sigma_k^2\Delta t \leq 1$ for all $k$, then increase it while monitoring violation and stability. For inequalities, $\epsilon$ is chosen as the smallest value that prevents boundary sticking without causing overshoot.

*Table 11.* Effect of hyperparameters $\alpha$ (with fixed $\epsilon = 0.05$) and $\epsilon$ (with fixed $\alpha = 50$) on JSD metrics and constraint violations for Alanine Dipeptide using ULLA without last step projection.

| Parameter | JSD ($\psi$ angle) | JSD (RMSD) | $\mathbb{E}[|h(x)|]$ | $\mathbb{E}[g(x)^+]$ |
|---|---|---|---|---|
| *Effect of $\alpha$ (with $\epsilon = 0.05$)* | | | | |
| $\alpha = 0.1$ | $0.419_{\pm.000}$ | $0.045_{\pm.002}$ | $5.40 \times 10^{-3}$ | $3.79 \times 10^{-3}$ |
| $\alpha = 0.5$ | $0.247_{\pm.000}$ | $0.036_{\pm.001}$ | $5.39 \times 10^{-3}$ | $2.82 \times 10^{-3}$ |
| $\alpha = 1.0$ | $0.134_{\pm.004}$ | $0.034_{\pm.003}$ | $5.51 \times 10^{-3}$ | $1.60 \times 10^{-3}$ |
| $\alpha = 5.0$ | $0.060_{\pm.002}$ | $0.033_{\pm.001}$ | $2.97 \times 10^{-3}$ | $3.42 \times 10^{-4}$ |
| $\alpha = 10.0$ | $0.053_{\pm.004}$ | $0.034_{\pm.002}$ | $1.02 \times 10^{-3}$ | $1.46 \times 10^{-4}$ |
| $\alpha = 20.0$ | $0.043_{\pm.001}$ | $0.035_{\pm.002}$ | $2.09 \times 10^{-4}$ | $4.28 \times 10^{-7}$ |
| $\alpha = 50.0$ | $0.033_{\pm.002}$ | $0.034_{\pm.002}$ | $7.20 \times 10^{-5}$ | $7.95 \times 10^{-8}$ |
| $\alpha = 100.0$ | $0.051_{\pm.003}$ | $0.035_{\pm.002}$ | $3.70 \times 10^{-5}$ | $6.92 \times 10^{-7}$ |
| $\alpha = 200.0$ | $0.059_{\pm.008}$ | $0.183_{\pm.066}$ | $2.73 \times 10^{-3}$ | $4.19 \times 10^{-3}$ |
| $\alpha = 400.0$ | $0.235_{\pm.001}$ | NaN | $6.74 \times 10^{-1}$ | $9.95 \times 10^{-1}$ |
| *Effect of $\epsilon$ (with $\alpha = 50$)* | | | | |
| $\epsilon = 0.001$ | $0.084_{\pm.001}$ | $0.036_{\pm.002}$ | $7.00 \times 10^{-5}$ | $3.49 \times 10^{-6}$ |
| $\epsilon = 0.005$ | $0.054_{\pm.003}$ | $0.033_{\pm.002}$ | $7.30 \times 10^{-5}$ | $3.58 \times 10^{-5}$ |
| $\epsilon = 0.01$ | $0.048_{\pm.002}$ | $0.032_{\pm.002}$ | $7.00 \times 10^{-5}$ | $5.49 \times 10^{-7}$ |
| $\epsilon = 0.05$ | $0.033_{\pm.002}$ | $0.034_{\pm.002}$ | $7.20 \times 10^{-5}$ | $7.95 \times 10^{-8}$ |
| $\epsilon = 0.1$ | $0.052_{\pm.007}$ | $0.035_{\pm.002}$ | $2.57 \times 10^{-4}$ | $5.26 \times 10^{-4}$ |
| $\epsilon = 0.5$ | $0.081_{\pm.009}$ | $0.738_{\pm.012}$ | $2.37 \times 10^{-1}$ | $1.98 \times 10^{-1}$ |
| $\epsilon = 1.0$ | $0.107_{\pm.024}$ | $0.788_{\pm.001}$ | $3.01 \times 10^{-1}$ | $1.95 \times 10^{-1}$ |
| $\epsilon = 5.0$ | $0.134_{\pm.002}$ | $0.796_{\pm.002}$ | $7.34 \times 10^{-1}$ | $2.97 \times 10^{-1}$ |
| $\epsilon = 10.0$ | $0.188_{\pm.003}$ | $0.796_{\pm.002}$ | $1.05 \times 10^{0}$ | $1.08 \times 10^{0}$ |

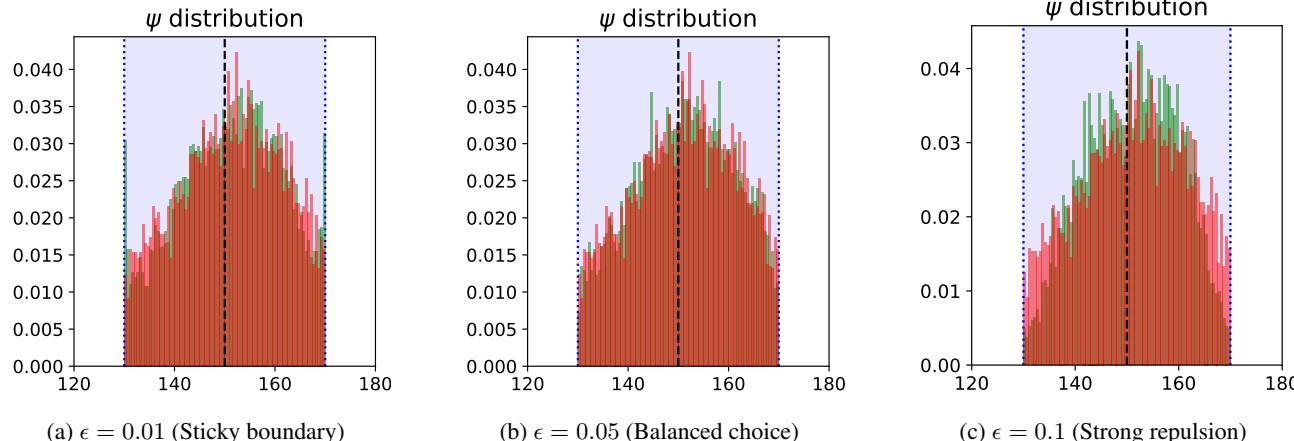

(a) $\epsilon = 0.01$ (Sticky boundary)  (b) $\epsilon = 0.05$ (Balanced choice)  (c) $\epsilon = 0.1$ (Strong repulsion)

*Figure 9.* **Effect of boundary repulsion rate $\epsilon$ on the generated distribution.** When $\epsilon$ is too small (a), trajectories tend to stick to the boundary. Conversely, an excessively large $\epsilon$ (c) aggressively pushes samples away from the boundary, distorting the distribution. A moderate choice (b) balances these effects, yielding the best sampling quality.

