# OpenReview forum: "Efficient Diffusion Models under Nonconvex Equality and Inequality constraints via Landing"
_ICML.cc/2026/Conference — ICML 2026 spotlight_

### Official Review · Reviewer_1DgZ · 2026-03-13

**Soundness:** 2
**Presentation:** 3
**Significance:** 3
**Originality:** 3
**Overall Recommendation:** 4
**Confidence:** 3

**Summary:**

This paper proposes a unified framework for constrained diffusion on nonconvex feasible sets with both equality and inequality constraints. The main contribution is a landing-based construction of overdamped and underdamped constrained Langevin dynamics (OLLA / ULLA), together with a Conditional Wasserstein Path Matching (CWPM) objective for training when discretized trajectories can leave the manifold slightly. The underdamped version is motivated by faster mixing and shorter forward trajectories. Experiments cover equality-only tasks (Earth/climate data, mesh manifolds, SO(10)) and mixed-constraint tasks (alanine dipeptide and 7-DOF robot arm), with strong emphasis on computational efficiency.

**Compliance With Llm Reviewing Policy:**

Affirmed.

**Final Justification:**

The authors have shown enough for me to bring the score up a little.

**Key Questions For Authors:**

1. Can the authors more clearly separate the continuous-time theoretical object from the practical discretized algorithm used in experiments, especially regarding feasibility guarantees and off-manifold samples?

2. How much projection work remains in practice because of the terminal Newton projection steps? A simple timing or failure-rate breakdown would help.

3. How sensitive are the main results to dropping the curvature correction terms, and can the authors quantify the performance / violation trade-off of this approximation?

4. Can the authors report constraint violations along trajectories (not only final averages) and clarify where the practical discretization most often departs from the feasible set?

**Limitations:**

The paper discusses some practical limitations, but it should more explicitly acknowledge that the experiments use an approximate practical variant: intermediate samples can leave the feasible set, curvature corrections are omitted, and terminal projection is still required. I would also like a clearer discussion of failure modes when Jacobians or pseudoinverses become ill-conditioned, and of sensitivity to the landing / repulsion hyperparameters in more difficult tasks.

**Strengths And Weaknesses:**

I think this paper addresses an important and practically relevant problem. Constrained generation on nonconvex feasible sets matters in scientific ML, robotics, and molecular modeling, and the proposed landing mechanism is appealing because it tries to replace expensive repeated projection with a first-order correction built into the dynamics. The underdamped variant is especially compelling from a systems perspective. I also found the empirical section strong overall: the paper quantifies wall-clock savings, evaluates both equality-only and mixed-constraint settings, and shows that ULLA can remain robust in settings where projection-based baselines become unstable or expensive. The 7-DOF robot arm and alanine dipeptide experiments are useful demonstrations of practical relevance.

My main concern is a mismatch between the strongest claims and the actual implementation. The paper is repeatedly framed as avoiding projection / Newton solves, but the algorithms still include terminal projections by Newton’s method in both the forward and backward pipelines. More importantly, the strongest guarantees seem to apply to the continuous-time landed dynamics, whereas the practical discretization can still produce off-manifold samples, and the experiments explicitly drop curvature correction terms to reduce cost. This is also the reason the paper moves to CWPM rather than standard likelihood-based training. I do not think these issues invalidate the method, but they do mean the headline language should be more carefully qualified. I also found parts of the presentation mathematically dense, and some assumptions / notation (e.g., rCRCQ, active sets, off-manifold extension) could be explained more clearly for a general ICML audience. Overall, I think the paper is useful and likely publishable, but only as a weak accept because the strongest claims are stated a bit too aggressively relative to the practical algorithm.

---

> ### Author Rebuttal · Authors · 2026-03-27
>
> Thank you for the helpful comment. We agree that our original wording might be slightly too strong, and we will revise it to distinguish more clearly between the continuous-time landing SDE and the practical discretized algorithms.
>
> **Separation of continuous-time SDE from discretized algorithms.** We agree that the practical algorithms still include an optional terminal Newton projection in both pipelines. Our claim is narrower: our practical method **avoids repeated per-step projection / Newton solves** and uses only an optional **terminal cleanup projection**.
>
> At the continuous-time level, if the process is initialized on $\Sigma$, the landing SDE stays on $\Sigma$ and preserves the constrained target law. Moreover, Lemma 1 shows that even if $X_t$ is off-manifold, the constraint residual decays exponentially fast. At the discrete level, however, intermediate iterates can be slightly off-manifold due to numerical error; curvature-dependent correction terms may be omitted for computational efficiency; and CWPM is used because the practical discrete trajectories are only approximately feasible.
>
> The key point is that the landing mechanism provides a built-in **normal self-correction**. In the equality-only case, one can show that
>
> $$
> \mathbb{E}[h(X_{k+1})] = (1-\alpha \sigma_k^2 \Delta t)\mathbb{E}[h(X_k)] + O(\Delta t).
> $$
>
> so constraint violations in the discretized algorithm are contracted rather than accumulated, under a proper choice of $\alpha$ (and $\epsilon$ for inequality constraints); see our response to **Reviewer rbSf** for the tuning discussion. This is why we can safely skip **intermediate** projections while keeping the trajectory close to $\Sigma$.
>
> **Terminal-projection and projection overhead ablation.** To show the effect of terminal projection, we include a **no-terminal-projection ablation** below (Ala.: $\psi$-JSD; Rob.: JSD at $d=560$).
>
> | Task | Method | Terminal projection | JSD | $\mathbb E\|h(x)\|$ | $\mathbb E\|g(x)^+\|$ |
> |---|:---:|:---:|:---:|:---:|:---:|
> | Alanine | ULLA | off | 0.033 | 7.2e-5 | 8.0e-5 |
> | Alanine | ULLA | on  | 0.031 | 1.4e-7 | 6.0e-10 |
> | 7-DOF | ULLA | off | 0.375 | 5.1e-3 | 3.9e-5 |
> | 7-DOF | ULLA | on  | 0.391 | 1.8e-5 | 1.5e-9 |
>
> After removing the final projection, final constraint violations remain very low while generative performance is nearly unchanged. Thus, terminal projection mainly improves final numerical precision rather than sample quality itself.
>
> We also quantify projection overhead for both ULLA and ULLA-P:
>
> | Task | Method | Inference time (ms/sample) | Projection time (ms/sample) | Overhead ratio (%) | Failure rate (%) |
> |---|:---:|:---:|:---:|:---:|:---:|
> | Alanine | ULLA | 0.13 | 0.01 | 9.0% | 0.0% |
> | Alanine | ULLA-P | 1.13 | 1.03 | 93.4% | 20.0% |
> | 7-DOF | ULLA | 107.50 | 21.99 | 20.5% | 0.0% |
> | 7-DOF | ULLA-P | 824.90 | 792.49 | 96.1% | 12.7% |
>
> Here, inference time is the total per-sample time (model forward + projection), and
>
> $$
> \text{overhead ratio}=\frac{\text{projection time}}{\text{inference time}}, \qquad
> \text{failure rate}=\frac{\text{\\# of failed trajectories}}{\text{\\# of total trajectories}}.
> $$
>
> These results show that, in the projection variants, projection cost dominates the model-forward cost: roughly $10\times$ on Alanine and $24\times$ on 7-DOF. This demonstrates why repeated projection is a critical bottleneck in general constrained diffusion. In addition, when projection fails, the trajectory must be discarded and re-sampled, which creates further computational overhead.
>
> We also report constraint violations throughout backward trajectories (inference starts from $k=100$ with an on-manifold prior and ends at $k=1$):
>
> | Metric (ULLA) | $k=1$ | $k=5$ | $k=10$ | $k=20$ | $k=50$ | $k=80$ | $k=100$ |
> |:---:|:---:|:---:|:---:|:---:|:---:|:---:|:---:|
> | Eq. viol. (Ala.) | 6.2e-5 | 2.0e-4 | 2.7e-4 | 2.7e-4 | 1.4e-4 | 8.2e-5 | 1.4e-6 |
> | Ineq. viol. (Ala.) | 6.5e-5 | 4.2e-5 | 8.0e-5 | 5.4e-5 | 9.9e-6 | 6.2e-7 | 0.0e+0 |
> | Eq. viol. (Rob.) | 8.0e-3 | 5.7e-3 | 3.7e-3 | 2.5e-3 | 9.7e-4 | 3.2e-4 | 4.0e-7 |
> | Ineq. viol. (Rob.) | 4.4e-5 | 2.7e-6 | 8.3e-6 | 0.0e+0 | 1.1e-6 | 0.0e+0 | 0.0e+0 |
>
> These values show that constraint violations remain sufficiently low throughout backward sampling, and directly support the intended picture: the discretized sampler is not exactly pathwise feasible, but the landing dynamics keep it **near-feasible throughout the trajectory**. We will incorporate the no-terminal-projection ablation, the projection-overhead breakdown, and the trajectory-wise diagnostics into the revised manuscript.
>
> **Dropping curvature correction terms and landing / repulsion hyperparameters.** Due to the length limit, we refer to our response to **Reviewer rbSf** for the discussion of correction terms and hyperparameter tuning. For the stability of the pseudoinverse $G(x)^\dagger$ and the stacked Jacobian $\nabla J(x)$, which is closely related to rCRCQ, we refer to our response to **Reviewer 6aS6**.

---

> > ### Author Rebuttal · Reviewer_1DgZ · 2026-04-04
> >
> > Thank you for the clarifications. That satisfies my inquiries.

---

> > > ### Author Response · Authors · 2026-04-05
> > >
> > > Thank you very much for your careful feedback and constructive discussion throughout the rebuttal process. We appreciate your comments on clarifying the distinction between the continuous-time theory and the practical discretized algorithm, as well as your suggestions on projection overhead and trajectory-wise constraint diagnostics. These points helped us identify where the original presentation needed to be more precise.
> > >
> > > We are glad that the additional clarifications and experiments addressed your concerns. If our rebuttal has positively affected your evaluation, we would be very grateful if this could be reflected in the final assessment. Thank you again for your time, thoughtful feedback, and engagement.

---

### Official Review · Reviewer_rbSf · 2026-03-13

**Soundness:** 4
**Presentation:** 2
**Significance:** 3
**Originality:** 3
**Overall Recommendation:** 5
**Confidence:** 3

**Summary:**

The paper proposes to solve equality and inequality-based constrained sampling problems by incorporating landing into diffusion models. By altering the dynamics to incorporate **landing**, a technique which originally appeared in the constrained optimization literature, we can ensure that the magnitude of constraint violation decreases exponentially. It is adapted to both overdamped and underdamped, with some tricks to reduce memory/runtime complexity for the underdamped dynamics. It is then assessed on numerous benchmarks against other methods.

**Compliance With Llm Reviewing Policy:**

Affirmed.

**Final Justification:**

Broadly, I think this paper proposes an interesting framework and contains non-trivial analysis, to the extent possible with current mathematical techniques, for sampling from constrained bodies. I believe this is both a natural problem and also a simple family of techniques for practitioners to investigate in practice. As a result, I believe this paper deserves to be accepted.

The other reviewers raised some concerns about feasibility/applicability of the method, which I think are valid. While I do not believe these are not hard constraints against acceptance, I do believe that the authors have responded well to such criticisms. Hence, I vote for acceptance.

**Key Questions For Authors:**

It is mysterious why only ULLA should succeed in the peptide task. Why is this the case? What about this algorithm makes it uniquely suited to this task?

Does the method still make sense if the Hessian term is omitted in the simulations? I feel like this significantly weakens the case for the algorithm.

What is a reasonable way to set the noise schedule and stepsizes?

**Minor**

It is not clear to me why the authors sometimes switch between Stratonovich and It\^o integrals. It would be ideal if we kept it considered It\^o consistently.

There is an extra space in Line 104, left column. Same in Line 268, left.

Line 255 left: “approximated score” $\gets$ “the approximate score”

Line 296 left: tangential part mean $\gets$ mean of the tangential part

**Limitations:**

Yes.

**Strengths And Weaknesses:**

**Strengths**

The method elegantly incorporates the landing mechanism into the SDE/diffusion model formulation. Although the resulting equations are not necessarily user friendly, a competent user can easily implement them.

The benchmarks are fairly diverse and thorough and contain detailed experiments.

The method appears significantly faster than competitors despite its relatively convoluted formulation.

The theory seems relatively rigorous and succeeds in translating the intuition from optimization. I should stress that this is not easy and the calculations for Theorem 1 and 2 are both quite hairy.

**Weaknesses**

The experimental benchmarks are not 100% convincing for me, and show that the algorithm isn’t always competitive, particularly on the Earth and Climate benchmark.

---

> ### Author Rebuttal · Authors · 2026-03-27
>
> We appreciate the reviewer’s recognition of the paper’s technical rigor and the helpful suggestions.
>
> **On empirical competitiveness.** We clarify that our goal is to retain *comparable* generative performance while improving efficiency by avoiding repeated projection. As shown in our response to **Reviewer 1DgZ**, this bottleneck is significant in both training and inference. On Earth/Climate, exact projection is available, yet we do **not** rely on it to demonstrate the practical value of landing. Even there, our method remains competitive, supporting the intended trade-off: preserving sample quality while removing repeated projection overhead.
>
> **Why ULLA works better on peptide.** We attribute this mainly to underdamped dynamics. Alanine is a **mixed-constraint** problem with a narrow feasible region and active inequality boundaries. In this regime, overdamped dynamics tend to hit the boundary repeatedly: even after repulsion pushes a sample away, the next noisy step can drive it back toward violation. By contrast, underdamped dynamics are more **ballistic** due to momentum, so once repelled, the trajectory tends to keep moving away instead of repeatedly re-entering the violating region.
>
> To make this concrete, we add the OLLA-vs-ULLA comparison:
>
> | Task | Method | JSD(psi) | JSD(RMSD) | Traj. AUC eq. | Traj. AUC ineq. |
> |:---:|:---:|:---:|:---:|:---:|:---:|
> | Alanine | OLLA | 0.0804 | 0.7830 | 3.93e-3 | 2.96e-3 |
> | Alanine | ULLA | 0.0372 | 0.0381 | 1.52e-4 | 2.14e-5 |
>
> Here, Traj. AUC denotes the area under the per-step violation curve, so larger values mean larger cumulative violation. The much smaller AUC values of ULLA show that its momentum keeps the trajectory closer to the feasible set.
>
> **Ablation on correction terms.** Our view is that the method **still makes sense without the correction term** and becomes much more computationally tractable, but it should then be interpreted as a **practical approximation** to the ideal continuous-time SDE rather than a faithful implementation of the exact SDE.
>
> More specifically, the omitted term is a **second-order curvature correction**. Since it is always multiplied by $\nabla J(x)^\top G(x)^\dagger$, it lies in the **manifold-normal direction**. Its main role is therefore not to transport the data distribution along the manifold, but to improve the second-order accuracy of constraint control.
>
> In our setting, the first-order **landing term** already suppresses constraint violation effectively. Therefore, the computationally expensive second-order correction may provide limited benefit in feasibility while increasing numerical stiffness and sensitivity. This is especially relevant when the constraint is parameterized by a neural network (e.g., Mesh data), because the correction depends on the **Hessian**, which can be locally noisy and destabilize the trajectory. Even when an exact analytic Hessian is available (as in alanine), the effect on both generative quality and constraint violation is empirically very small. To support this, we include the following ablation on correction terms (Alanine JSD = $\psi$ angle).
>
> | Task | Method | Correction | JSD | Avg. eq. viol. | Avg. ineq. viol. |
> |:---:|:---:|:---:|:---:|:---:|:---:|
> | Bunny-100 | ULLA | off | 0.034 | 1.59e-7 | NA |
> | Bunny-100 | ULLA | on  | 0.040 | 2.24e-5 | NA |
> | Alanine | ULLA | off | 0.034 | 1.45e-6 | 1.20e-10 |
> | Alanine | ULLA | on  | 0.035 | 1.48e-6 | 3.90e-10 |
>
> **Choice of noise schedule, step size, and hyperparameters.** We use the **linear schedule** as in the baseline implementations. Once the schedule $\{\sigma_k\}_{k=0}^N$ and step size $\Delta t$ are fixed, the main task-specific parameters are the landing rate $\alpha$ and the boundary repulsion rate $\epsilon$.
>
> A useful guideline for choosing $\alpha$ comes from the one-step equality-only residual evolution in the discretized algorithm:
> $$
> \mathbb{E}[h(X_{k+1})] = (1-\alpha \sigma_k^2 \Delta t)\mathbb{E}[h(X_k)] + O(\Delta t).
> $$
> This suggests the stability condition $0 < \alpha \sigma_k^2 \Delta t < 2$ for all $k \in [N]$, under which the residual decays without oscillation or blow-up. After fixing the diffusion schedule and $\Delta t$, we initialize $\alpha_{\mathrm{init}} := \min_{0\le k\le N} 1/(\sigma_k^2 \Delta t)$ so that $\alpha_{\mathrm{init}} \sigma_k^2 \Delta t \le 1$ for all $k$, and then gradually increase $\alpha$ while monitoring training stability and constraint violation.
>
> For inequality constraints, $\epsilon$ controls boundary behavior. If $\epsilon$ is too small, trajectories become **sticky** near the boundary; if it is too large, repulsion becomes overly aggressive and trajectories overshoot into the interior. In practice, we choose the smallest $\epsilon$ that avoids persistent sticking without overshoot. Empirically, $\epsilon$ was less sensitive than $\alpha$ and usually easy to tune. We will incorporate both the correction-term ablation and this tuning discussion into the revised manuscript.

---

> > ### Author Rebuttal · Reviewer_rbSf · 2026-04-04
> >
> > I think this paper proposes an interesting framework which fuses landing with sampling, and this is not trivial to implement in this context. However, I think the relative complexity of this method means that practitioners might prefer simpler methods with fewer parameters to tune. I think it is an interesting contribution, just not necessarily a strong accept. I maintain my score.

---

> > > ### Author Response · Authors · 2026-04-05
> > >
> > > Thank you very much for your thoughtful questions, and constructive engagement throughout the rebuttal process. We especially appreciate your comments on ULLA’s behavior on the peptide task, the role of the correction term, and the discussion on hyperparameters. These points helped us clarify several aspects of the paper and improve the presentation. We are very grateful for your time, feedback, and engagement.

---

### Official Review · Reviewer_6aS6 · 2026-03-13

**Soundness:** 2
**Presentation:** 2
**Significance:** 2
**Originality:** 2
**Overall Recommendation:** 3
**Confidence:** 2

**Summary:**

This paper introduces a landing-based constrained diffusion method for sampling from constrained distributions.

The method is designed for general nonconvex constraint sets and uses a landing mechanism to steer the sampling trajectory toward the feasible set.

Unlike prior projection-based approaches, the proposed method adopts a Lagrangian formulation that governs the entire sampling process.

The paper further shows that this design is more efficient, and reports faster runtimes than Riemannian Denoising Diffusion Probabilistic Models.

**Compliance With Llm Reviewing Policy:**

Affirmed.

**Key Questions For Authors:**

NA

**Strengths And Weaknesses:**

**Strengths:**

- The authors introduce a new method to make the diffusion diffusion models practical when samples must satisfy nontrivial equality and inequality constraints throughout the forward and backward processes.

- The methodological contribution is reasonable. The landing method is an alternative method to projections, and the paper gives a unified SDE formulation for both equality and inequality constraints. The exponential-decay target property for constraint violation is reasonable.

- The paper also has a nice systems angle. ULLA is designed to reduce memory by collapsing the dynamics into a second-order Markov chain over positions, and the experiments suggest strong efficiency gains, including much faster sampling and training in some settings.

- The experimental section is reasonably broad. It includes equality-only and mixed-constraint tasks, reports both quality and constraint-violation metrics.


**Weaknesses:**

- Limited novelty. The core idea of using a landing mechanism is not particularly new. Landing-type methods have been widely studied in constrained optimization, especially for problems with manifold constraints or more general Riemannian geometric structures. From this perspective, the paper mainly adapts an augmented penalty/Lagrangian-style correction into the diffusion sampling process, rather than introducing a fundamentally new conceptual framework.

Moreover, the landing strategy is inherently an infeasible approach: it typically guarantees only that the limit point satisfies the constraints, while intermediate iterates are not necessarily feasible. This is an important distinction, especially when compared with methods that enforce feasibility throughout the trajectory. For constrained generation problems, the fact that intermediate samples may violate the constraints could be a nontrivial drawback.

- The authors assumt that the feasible set $\Sigma$ is assumed to be a **stratified manifold**, and the equality/inequality constraints are assumed to satisfy rCRCQ in a neighborhood of $\Sigma$. These assumptions are not lightweight assumptions for general nonconvex constrained-generation problems. In realistic applications, especially when constraints are learned, noisy, or highly irregular, it is not obvious when these assumptions hold or how one would verify them.

- Theorem 1 assumes the existence of constants $\Lambda_{k+1}$ such that $W_2(\sigma_k,\sigma_{k+1}) \le \Lambda_{k+1} W_2(q_k, q_{k+1}T^\theta_{k+1}) + O(\sqrt{\Delta t})$, stating that this holds under “minor regularity assumptions” on the score and constraint functions. The paper does not make fully clear how restrictive these assumptions are, whether they are realistic for neural score models.

- Experimental evaluation is limited. The empirical study compares against only a narrow set of baselines, primarily Riemannian diffusion models, which makes it difficult to assess the true competitiveness of the proposed method. In addition, many of the experimental settings still resemble relatively controlled or toy-style examples, leaving open the question of how well the method would scale to more realistic constrained generation tasks.

---

> ### Author Rebuttal · Authors · 2026-03-27
>
> We appreciate the reviewer’s helpful comments.
>
> **On novelty.** We agree that landing itself is widely studied in constrained optimization. Our contribution is not landing itself, but its use in diffusion-based generative modeling: (i) a unified framework for *mixed equality and inequality constraints* on general nonconvex feasible sets; (ii) a landed **underdamped** constrained dynamics (ULLA), which, to our knowledge, is new in this setting; and (iii) a CWPM objective tailored to the discretized regime where small off-manifold errors may appear.
>
> This is also different from adding a penalty, guidance, or augmented-Lagrangian term to a Euclidean backward sampler. In our framework, the **forward process is also constrained**, and the backward dynamics are derived as its time-reversal. In the revised experiments, we also include Euclidean-forward + constrained-backward baselines, which show that constrained forward diffusion is more effective than unconstrained forward baselines.
>
> **On off-manifold intermediate samples.** As clarified in our response to **Reviewer 1DgZ**, our goal is to replace expensive per-step projection by a much cheaper landing mechanism while retaining the performance of Riemannian/projection-based constrained diffusion. Our design allows **small intermediate violations** to avoid repeated Newton/projection subproblems at every diffusion step. Unlike generic infeasible methods, the landing term acts as a **self-correcting normal drift** that suppresses residuals and keeps the trajectory close to $\Sigma$, preventing error accumulation. In practice, the per-step violations remain small, and for inequality constraints the landing dynamics can even recover feasibility in finite time. Empirically, Table 1 shows that OLLA is comparable to OLLA-P (RDDPM) and ULLA to ULLA-P, where the (-P) variants use per-step projection. Moreover, in the harder mixed-constraint tasks (7-DOF, alanine), the per-step projection variants fail frequently and were therefore excluded. For further ablations, please refer to Section 5.2 and our response to **Reviewer 1DgZ** on trajectory-wise violations.
>
> **On rCRCQ.** The role of the stratified-manifold and rCRCQ assumptions is to ensure that the geometric operators $\Pi(x)$ and $G(x)^\dagger$ are well-defined and continuous near $\Sigma$. Specifically, under rCRCQ, the rank of the stacked active Jacobian $\nabla J(x)$ is locally constant near $\Sigma$, so the pseudo-inverse and projector remain well behaved. Many practical constraints (polynomial, kinematic, etc.) are smooth analytic functions of the state $x$. When $d \geq m + l$, for such constraints, the rank change of $\nabla J(x)$ occurs only on a lower-dimensional singular set, which has Lebesgue measure zero in $\mathbb{R}^d$ [1]. In our setting, the discretized algorithms (OLLA, ULLA) incur discretization errors and evolve on the entire ambient space $\mathbb{R}^d$. Since this singular set has Lebesgue measure zero, a randomly perturbed iterate lands on it with essentially zero probability. In other words, the discretized algorithms **almost surely** evolve in regions where rCRCQ holds.
>
> **On Theorem 1 assumptions.** We agree that “minor regularity assumptions” was too vague. More precisely, Appendix D.1/D.2 provide explicit sufficient conditions for the existence of $\Lambda_{k+1}$: controlled second-moment growth of the landing term, Lipschitz/linear-growth bounds for the score-induced drift, and bounded second moments under the iterated reverse kernels. Also, we do **not** claim that arbitrary unconstrained neural score networks satisfy these automatically. Rather, Theorem 1 should be read as a **conditional sufficient result for a regularized score family**, whose analogues are standard in diffusion theory [2–3]. In practice, these assumptions are compatible with common controls such as weight regularization or gradient clipping. We will revise the manuscript accordingly and replace “minor regularity assumptions” by these explicit sufficient conditions.
>
> **On experimental breadth.** We respectfully note that the experiments are not limited to Riemannian baselines: besides RFM/RDDPM, we also added Euclidean, Projected, Lagrangian, and Guided baselines. The benchmarks also span both settings where Riemannian methods are natural baselines (e.g., $S^2$ and SO(10)) and settings where projection is expensive or ill-defined in practice (e.g., mesh, alanine, and 7-DOF). Some tasks are diagnostic, but the paper also includes practical settings such as SO(10), alanine dipeptide, and the 7-DOF arm with hundreds of equality/inequality constraints. For more discussion on this, please refer to our response to **Reviewer rbSf**.
>
> **References.**
> [1] Mityagin, *The Zero Set of a Real Analytic Function*, 2015.
> [2] Chen, Lee, Lu, *Improved Analysis of Score-based Generative Modeling*, 2022.
> [3] Wang, Wang, *Wasserstein Bounds for generative diffusion models with Gaussian tail targets*, 2024.

---

> > ### Author Rebuttal · Reviewer_6aS6 · 2026-04-08
> >
> > Thank you for the rebuttal. My concerns are only partially resolved.
> >
> > In my view, the main contribution is to incorporate constraints into Langevin diffusion via a landing-based correction term. Beyond this aspect, much of the remaining analysis seems to follow a strategy very similar to that used in projection-based approaches, especially RDDPM and DT-ELBO. Since landing is already a fairly mature idea in constrained optimization (especially in optimization over Manifolds), this adaptation appears natural, and I am not yet fully convinced that the level of novelty is strong enough.
> >
> > However, diffusion models are not my primary area of expertise, so I may be underestimating the significance of this contribution in this community. For this reason, I remain somewhat uncertain in my assessment.

---

> > > ### Author Response · Authors · 2026-04-08
> > >
> > > Thank you very much for the follow-up and for taking the time to engage with our rebuttal. We appreciate your candid clarification that the remaining concern is primarily about the level of novelty, and we also appreciate your openness regarding the possibility that the significance of this contribution may be easier to assess from within the diffusion model community.
> > >
> > > Just to clarify our intended claim one last time: we do **not** view the novelty as the landing idea in isolation. We agree that landing-type formulations have an established history in constrained optimization. Our claim is that the contribution lies in bringing this idea into the **Langevin/diffusion setting** in a way that is both mathematically well defined and practically useful. In particular, while the closest constrained-sampling paper in our view is [1], that framework is limited to the **overdamped** Langevin setting. By contrast, our paper introduces a landing-based **underdamped** Langevin framework (ULLA), which, to the best of our knowledge, is new.
> > >
> > > We believe this is meaningful for two reasons. First, this extension is not automatic: once one moves to the stochastic underdamped setting, the Brownian component introduces additional geometric correction terms that do not appear in the deterministic optimization setting, so the construction requires a separate and more delicate analysis than the optimization analogues. Second, beyond the formulation itself, we show that ULLA is practically useful in challenging constrained diffusion settings. As discussed in Section 5.2 and in our response to Reviewer rbSf, the ballistic behavior of ULLA leads to substantially more stable forward/backward dynamics on the harder mixed-constraint tasks.
> > >
> > > We would also respectfully disagree with the view that the remaining analysis is essentially the same as RDDPM / DT-ELBO. As shown in Section 4.3, our CWPM formulation is introduced precisely for the landing setting, where one allows small pathwise constraint violations in the discretized regime while still removing the theoretical singularity that would otherwise arise in the training objective. The resulting variational bounds in Theorems 1 and 2, together with the analysis in Appendices C and D, are therefore tailored to this landing off-manifold setting and are substantively different from the standard DT-ELBO analysis.
> > >
> > > So, at a high level, we agree that the overall picture may look natural in hindsight. Our point is that formulating it rigorously in the stochastic sampling setting, extending it to constrained diffusion models, and turning it into a practical algorithm that can skip most projection steps while preserving comparable sample quality and substantially reducing computational cost, are precisely the main contributions of the paper.
> > >
> > > Thank you again for your thoughtful engagement. If our clarifications have positively affected your evaluation, we would be very grateful if this could be reflected in the final score.
> > >
> > > **References.**
> > > [1] Jeon, Muehlebach, Tao, *Fast Non-Log-Concave Sampling under Nonconvex Equality and Inequality Constraints with Landing*, NeurIPS, 2025.

---

### Decision · Program_Chairs · 2026-04-30

**Decision:**

Accept (spotlight)

**Comment:**

The paper proposes a unified constrained diffusion framework that uses a landing mechanism to replace expensive per-step projections onto nonconvex sets. Its primary strength is computational efficiency, significantly reducing the number of function evaluations and memory usage during training and inference. The introduction of underdamped dynamics (ULLA) provides a "ballistic" advantage, keeping trajectories closer to feasible sets in complex tasks like molecular modeling. However, reviewers noted limited novelty, as landing is a mature optimization concept. A key weakness is that the method is inherently infeasible, meaning intermediate samples may violate constraints, unlike projection-based approaches. Additionally, while theoretically rigorous, the practical implementation often omits curvature corrections for speed and requires careful hyperparameter tuning. Despite these trade-offs, the method offers a scalable solution for practical applications where exact projections are ill-defined or too costly.